

# Anomalies of generalized symmetries from solitonic defects

Lakshya Bhardwaj[1], Mathew Bullimore[2],
Andrea E. V. Ferrari[2] and Sakura Schäfer-Nameki[1]

**1** Mathematical Institute, University of Oxford,
Woodstock Road, Oxford, OX2 6GG, United Kingdom
**2** Department of Mathematical Sciences, Durham University, Lower Mountjoy,
Stockton Road, Durham, DH1 3LE, United Kingdom

## Abstract

We propose the general idea that 't Hooft anomalies of generalized global symmetries can be understood in terms of the properties of solitonic defects, which generically are non-topological defects. The defining property of such defects is that they act as sources for background fields of generalized symmetries. 't Hooft anomalies arise when solitonic defects are charged under these generalized symmetries. We illustrate this idea for several kinds of anomalies in various spacetime dimensions. A systematic exploration is performed in 3d for 0-form, 1-form, and 2-group symmetries, whose 't Hooft anomalies are related to two special types of solitonic defects, namely vortex line defects and monopole operators. This analysis is supplemented with detailed computations of such anomalies in a large class of 3d gauge theories. Central to this computation is the determination of the gauge and 0-form charges of a variety of monopole operators: these involve standard gauge monopole operators, but also fractional gauge monopole operators, as well as monopole operators for 0-form symmetries. The charges of these monopole operators mainly receive contributions from Chern-Simons terms and fermions in the matter content. Along the way, we interpret the vanishing of the global gauge and ABJ anomalies, which are anomalies not captured by local anomaly polynomials, as the requirement that gauge monopole operators and mixed monopole operators for 0-form and gauge symmetries have non-fractional integer charges.



# 1  Introduction

The study of generalized global symmetries and their 't Hooft anomalies [1–3] has seen a flurry of activity over the past few years [4–85].

Traditionally, 't Hooft anomalies of generalized symmetries are characterized in terms of non-invariance of the partition function on a compact spacetime manifold under gauge transformations of background fields for such symmetries. In this work, we point out a different but equivalent way of characterizing 't Hooft anomalies in terms of charges under generalized symmetries of certain defects, that we call *solitonic defects*. Their defining property is that they induce backgrounds for generalized symmetries in their vicinity. This point of view on 't Hooft

anomalies has a distinct advantage over the traditional point of view, in that we do not have to concern ourselves with the subtleties of defining the theory on a compact spacetime manifold.

This is particularly relevant for lattice systems studied in condensed matter physics, which cannot be naturally defined on a compact spacetime manifold. On the other hand, it is natural to define defects in such systems. Thus, the solitonic defect point of view should be especially useful in formulating and understanding 't Hooft anomalies of generalized symmetries in condensed matter systems. The usefulness of defect based approaches in studying generalized symmetries of condensed matter systems was also stressed in [86].

The relationship between solitonic defects and 't Hooft anomalies is bi-directional. On the one hand, if we know 't Hooft anomalies, then we learn information about the charges of solitonic defects under generalized symmetries. On the other hand, if we can compute charges of solitonic defects under generalized symmetries, then we can use that information to deduce 't Hooft anomalies. The former can be thought of as providing new physical consequences of 't Hooft anomalies, and the latter can be thought of as providing a novel way for computing 't Hooft anomalies.

In this paper, we use the computational approach opened up by solitonic defects to compute 't Hooft anomalies of 0-form, 1-form and 2-group symmetries in a large class of 3d gauge theories. The solitonic defects relevant for this analysis are vortex line defects and monopole local operators, which are named so because they induce vortex and monopole configurations for background gauge field for 0-form symmetry in their vicinity. Previous work on generalized symmetries and anomalies of 3d QFTs includes [8, 9, 11, 15, 87–98].

Special cases of understanding 't Hooft anomalies in terms of properties of solitonic defects already appeared in the seminal work [3]. The solitonic defects considered there were rather special, being the topological defects generating the generalized symmetries. The 't Hooft anomalies of generalized symmetries were then interpreted as the fact that these topological defects are charged under themselves. In the present paper we generalize their description to include arbitrary, *generically non-topological* solitonic defects.

At this point, let us mention that our work is a natural continuation of similar recent works [31, 39, 44, 47, 59, 85] that describe generalized symmetries in terms of properties of the spectrum of arbitrary (generically non-topological) defects. As discussed in these works, this point of view provides novel methods for computing generalized symmetries in strongly-coupled non-Lagrangian systems. Examples are 4d $\mathcal{N} = 2$ Class S theories, where one does not have alternative methods for computing these symmetries, as well as strongly-coupled 5d and 6d SCFTs. In this spirit, we believe that the present work will provide insight in the computation of 't Hooft anomalies in strongly-coupled theories. In fact, in an upcoming work [99] we will use the results of this paper to compute generalized symmetries and 't Hooft anomalies in strongly-coupled 3d $\mathcal{N} = 4$ SCFTs.

## 1.1 Relationship between anomalies and solitonic defects

In this subsection, we illustrate the relationship between 't Hooft anomalies of generalized symmetries and solitonic defects. The basic idea can be illustrated with a simple example of a $\mathbb{Z}_2^{(1)} \times \mathbb{Z}_2^{(2)}$ 1-form symmetry in 3d. Consider a solitonic line defect $L$ sourcing a background for $\mathbb{Z}_2^{(1)}$ 1-form symmetry such that

$$\int_{D_2} B_2^{(1)} \neq 0 \in \mathbb{Z}_2 \,, \tag{1}$$

where $B_2^{(1)}$ is the background field for $\mathbb{Z}_2^{(1)}$ 1-form symmetry and $D_2$ is a small disk intersecting the locus of $L$ at a single point – see figure 1. Such a solitonic line defect $L$ is generically a non-topological defect e.g. a defect inducing a vortex configuration for gauge fields, although

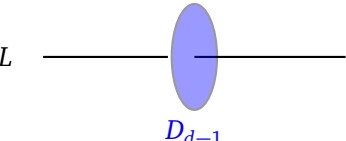

Figure 1: Depiction of a solitonic line operator $L$, which is pierced by a codimension 1 ball $D_{d-1}$. Integrating $w_2, B_2, B_w$ over this disk determines the background for 0- and 1-form as well as 2-group symmetries by the solitonic line operator.

examples include the topological line defects implementing the $\mathbb{Z}_2^{(1)}$ 1-form symmetry. Now, a mixed 't Hooft anomaly between the two 1-form symmetries given by

$$\mathcal{A}_4 = \exp\left(2\pi i \int B_2^{(1)} \cup B_2^{(2)}\right), \tag{2}$$

is equivalent to the statement that $L$ carries a non-trivial charge under $\mathbb{Z}_2^{(2)}$ 1-form symmetry. Similarly, the anomaly is also equivalent to the statement that a solitonic line defect inducing a non-trivial background for $B_2^{(2)}$ has non-trivial charge under $\mathbb{Z}_2^{(1)}$.

Let us illustrate the broad method in a bit more detail by considering solitonic line defects in $d$ dimensions that source background fields for generalised symmetries. We assume there is a group-like symmetry $\mathcal{G}$, which could be at most a $(d-1)$-group symmetry in $d$-dimensions. We denote the background fields background fields collectively by $\mathcal{B}$. A general (non-topological) line defect may have the property that

$$\int_{D_{d-1}} q_{d-1}(\mathcal{B}), \tag{3}$$

is non-trivial for some degree $d-1$ characteristic class $q_{d-1}(\mathcal{B})$ constructed from the background fields valued in some finite Abelian group $A$. Here $D_{d-1}$ denotes a small ball intersecting the line. Some examples of such solitonic lines are:

- In dimension $d = 3$, vortex lines defects may source backgrounds for 0-form, 1-form and 2-group symmetries

$$\int_{D_2} w_2, \quad \int_{D_2} B_2, \quad \int_{D_2} B_w, \tag{4}$$

where $w_2$ is the obstruction for lifting a continuous flavor symmetry bundle, $B_2$ and $B_w$ are the background fields for 1-form and 2-group symmetries.

- In dimension $d = 4$, Wilson-'t Hooft line defects may source a backgrounds for 1-form symmetries

$$\int_{D_3} \text{Bock}(B_2), \tag{5}$$

where $B_2$ is background for a 1-form symmetry and Bock is the Bockstein homomorphism associated to some short exact sequence.

- In dimension $d = 5$, fractional instanton lines may source backgrounds for 1-form symmetries

$$\int_{D_4} \mathcal{P}(B_2), \tag{6}$$

where $B_2$ is background for a 1-form symmetry and $\mathcal{P}$ denotes the Pontryagin square operation.

The properties of such solitonic line defects in terms of how the local operators that they end on transform under 0-form symmetries and whether they are themselves charged under 1-form symmetries determine the form of 't Hooft anomalies. For example:

- Solitonic line defect may end on local operators charged under an extension of a 0-form symmetry group by a finite group $\mathcal{Z}$. This determines a homomorphism $\gamma : A \to \widehat{\mathcal{Z}}$ with associated 't Hooft anomaly

$$2\pi i \int_{M_{d+1}} w_2 \cup \gamma(q_{d-1}(\mathcal{B})), \tag{7}$$

where $w_2$ is the obstruction to lifting the flavour group background to the extension. An example of such 't Hooft anomaly in three-dimensions is

$$\frac{2\pi i}{N} \int_{M_4} (c_1 \bmod N) \cup B_2, \tag{8}$$

between a $U(1)$ 0-form symmetry with obstruction $w_2 = c_1 \bmod N$ ($c_1$ being the first Chern class) and a $\mathbb{Z}_N$ 1-form symmetry with background $B_2$. This arises, for example, in a 3d $U(1)$ gauge theory with a scalar field of charge $N$.

- Solitonic line defects may be charged under a 1-form symmetry $\Gamma^{(1)}$. This determines a homomorphism $\gamma : A \to \widehat{\Gamma^{(1)}}$ with associated 't Hooft anomaly

$$2\pi i \int_{M_{d+1}} B_2 \cup \gamma(q_{d-1}(\mathcal{B})). \tag{9}$$

An example of such an anomaly in four dimensions is a mixed anomaly of the form

$$\int_{M_4} B_2^{(1)} \cup \mathrm{Bock}\left(B_2^{(2)}\right), \tag{10}$$

for a $\mathbb{Z}_2 \times \mathbb{Z}_2$ 1-form symmetry. This arises, for example, in 4d pure $SO(N)$ gauge theory with $N = 4k + 2$.

The above examples can be extended in a variety of directions to incorporate the mixing of the 0-form and 1-form symmetries into a 2-group, or the properties of solitonic defects of different codimension, all of which capture different types of 't Hooft anomalies. The primary focus of this paper is in dimension $d = 3$.

## 1.2 Outline

In more detail, the contents of the paper are as follows:

- In section 2, we begin, in general spacetime dimension $d$, by reviewing how the properties of the spectrum of (generically non-topological) line and local operators capture information about 0-form, 1-form and 2-group symmetries. This lays the groundwork for the considerations of subsequent sections.

- In section 3, we discuss solitonic defects arising in low-codimension, focusing in particular on codimension two and three. For (continuous) 0-form symmetry, we consider vortex defects, which are codimension-two, and monopole operators/'t Hooft defects, which are codimension-three. For 1-form and 2-group symmetries, we discuss codimension-two and codimension-three defects which induce in their vicinity degree-two background fields for these symmetries.

- In section 4, we discuss gaugings of generalized symmetries with a particular focus on the fate of the solitonic defects discussed in section 3 under the gauging process. We find that some of the solitonic defects become non-solitonic, while others remain solitonic, but change their type. For example, vortex defects for 0-form symmetry before gauging become solitonic defects inducing 1-form symmetry background after gauging, etc. Among the defects that become non-solitonic after gauging, we find standard gauge monopole operators (also known as 't Hooft defects) widely studied in the literature. Among solitonic defects inducing 1-form symmetry background, we encounter a slight generalization of standard gauge monopole operators, that we refer to as fractional gauge monopole operators.

- In section 5, we specialize to 3d, systematically exploring various kinds of 't Hooft anomalies of generalized symmetries and describe how they are beautifully captured in terms of charges under generalized symmetries of solitonic defects of the types discussed in section 3.

- In section 6, we describe how the formalism of section 5 can be used to determine generalized symmetries and their 't Hooft anomalies in a large class of 3d gauge theories. These are captured in the gauge and 0-form symmetry charges of various kinds of monopole operators: standard gauge monopole operators, fractional gauge monopole operators, monopole operators for 0-form symmetry, and mixed gauge/0-form monopole operators. The relevant charges of these monopole operators can be computed essentially from the information about Chern-Simons terms for the gauge and 0-form symmetry groups, and the spectrum of fermionic matter fields. A 3d theory cannot have gauge and ABJ anomalies described by anomaly polynomials, but it can have global versions of such anomalies which are not described by anomaly polynomials. We encounter such anomalies and phrase their vanishing as monopole operators (of various types discussed above) having non-fractional integer charges.

- In section 7, we illustrate the discussion of section 6 in many simple and concrete examples. We discuss various kinds of gauge theories having various number of supercharges, including non-supersymmetric theories. The examples have been chosen to illustrate various computational steps in detail.

- In section 8, we discuss various other kinds of solitonic defects, 't Hooft anomalies and the relationship between them. The anomalies that we discuss involve other kinds of anomalies in 3d not discussed in the bulk of this paper, and some well-known anomalies in 4d and 5d.

Sections 2-4 are applicable in general dimensions, whereas we specialize to 3d theories in sections 5-7. Section 8 provides examples in dimensions $d = 4, 5$.

## 2 Review of generalized symmetries

In this section, we begin by reviewing how the 0-form, 1-form and 2-group symmetries are physically encoded in the spectrum of extended and local operators in a $d$-dimensional quantum field theory. The discussion in this section will be dimension independent.

The contents of this section are mostly a review of the discussions appearing in earlier works [44, 47, 85]. However, a few points we make are new.

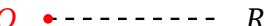
$$O \bullet \text{-------} R$$

Figure 2: A genuine local operator $O$ transforming in a representation $R$ of the 0-form symmetry group $\mathcal{F}$ needs to be attached to a background Wilson line in the representation $R$ of $\mathcal{F}$.

## 2.1 0-form symmetries

**Definition.** We begin with a discussion of the continuous 0-form symmetry of a theory, associated to a Lie algebra $\mathfrak{f}$ of the form

$$\mathfrak{f} = \bigoplus_i \mathfrak{f}_i \,, \tag{11}$$

where each $\mathfrak{f}_i$ is either a simple Lie algebra or an Abelian $\mathfrak{u}(1)$ factor. We require that for each $\mathfrak{f}_i$, the theory contains at least one genuine local operator[1] transforming in a non-trivial representation of $\mathfrak{f}$. This requirement is always satisfied for each $\mathfrak{f}_i$ that is non-Abelian, since the current operator transforms in the adjoint representation, which is non-trivial if $\mathfrak{f}_i$ is non-Abelian. For an Abelian $\mathfrak{f}_i = \mathfrak{u}(1)$, there might not exist such a local operator, and this becomes an assumption on the class of $\mathfrak{f}$ that we study.

The *0-form symmetry group* is the Lie group satisfying the following properties:

1. $\mathcal{F}$ can be expressed as

$$\mathcal{F} = \frac{\prod_i \mathcal{F}_i}{\Gamma} \,, \tag{12}$$

   where each $\mathcal{F}_i$ is a compact connected Lie group whose Lie algebra is $\mathfrak{f}_i$ and $\Gamma$ is a subgroup of the center of $\prod_i \mathcal{F}_i$.

2. If there is a genuine local operator transforming in a representation $R$ of $\mathfrak{f}$, then $R$ must be an allowed representation of $\mathcal{F}$.

3. Conversely, if $R$ is a representation of $\mathfrak{f}$ that is an allowed representation of $\mathcal{F}$, then there exists at least one genuine local operator transforming in $R$.

In other words, $\mathcal{F}$ is a group with Lie algebra $\mathfrak{f}$, which takes the form (12) and acts faithfully on the genuine local operators.

**Backgrounds and genuine local operators.** In the presence of a background gauge field for 0-form symmetry, which is a principal $\mathcal{F}$ bundle with a connection, a genuine local operator $O$ in a representation $R$ of $\mathcal{F}$ is not invariant under background gauge transformations, but can be made gauge invariant by attaching it to a background Wilson line in the representation $R$. See figure 2. After attaching the background Wilson line, correlation functions involving $O$ depend on the precise locus of the background Wilson line. In particular, the background Wilson line is not topological, but "almost topological", in the sense that small deformations of it change the correlation function by fluxes of field strength for background gauge field.

---

[1]A genuine $p$-dimensional defect/operator can be defined independently of any other higher-dimensional defects/operators. On the other hand, a non-genuine $p$-dimensional defect/operator arises at a (possibly complicated) junction of other higher-dimensional defects/operators. Also note that in this paper, we use the words 'defect' and 'operator' interchangeably, as we are working with QFTs with spacetime Euclidean invariance.

**Backgrounds and non-genuine local operators.** One can also consider correlation functions involving non-genuine local operators in the presence of background gauge fields for $\mathcal{F}$. Such local operators may transform in representations of $\mathfrak{f}$ that are not representations of $\mathcal{F}$, but instead representations of a group $F$ with the same Lie algebra $\mathfrak{f}$, which is a central extension of $\mathcal{F}$. That is, we can express $\mathcal{F}$ as

$$\mathcal{F} = F/\mathcal{Z}, \tag{13}$$

where $\mathcal{Z}$ is a finite subgroup of the center $Z_F$ of $F$. A background for 0-form symmetry $\mathcal{F}$ comes equipped with a $\mathcal{Z}$-valued 2-cocycle $w_2$ describing the obstruction of lifting the associated $\mathcal{F}$ bundle to an $F$ bundle. Note that, even though every local operator needs to form a representation of $F$, we do not require that every representation of $F$ needs to be realized by some local operator. Thus, in some situations, one has several consistent choices for $F$, each having a different obstruction class $w_2$.

**Examples.** If the 0-form symmetry group acting faithfully on genuine local operators is $\mathcal{F} = SO(3)$, it may happen that the non-genuine local operators transform in representations of $F = SU(2)$ that are not representations of $SO(3)$, and in this case $\mathcal{Z} = \mathbb{Z}_2$. The obstruction class $w_2$ is then the second Stiefel-Whitney class of the background $SO(3)$-bundle.

This scenario is realized in a $U(1)$ gauge theory with two matter fields $\phi_1$ and $\phi_2$ of charge 1 under the $U(1)$ gauge group, such that $\phi_1$ and $\phi_2$ form a doublet under a flavor symmetry algebra $\mathfrak{f} = \mathfrak{su}(2)$. The genuine local operators are gauge invariant operators, which form only integer spin representations of $\mathfrak{su}(2)$, and hence $\mathcal{F} = SO(3)$. On the other hand, $\phi_i$ can be made gauge invariant by inserting them at the end of a Wilson line of charge 1 under the $U(1)$ gauge group. Thus, $\phi_i$ provide non-genuine operators which transform in half-integer spin representations of $\mathfrak{su}(2)$, and hence $F = SU(2)$.

Another example frequently encountered is that genuine local operators transform with integer charge under a 0-form symmetry $\mathcal{F} = U(1)$, but non-genuine local operators transform with fractional charge in multiples of $1/N$. In such a situation, we can choose $F$ to be a group isomorphic to $U(1)$, which is a $kN$-fold cover of $\mathcal{F} = U(1)$, where $k \geq 1$ is an arbitrary integer

$$1 \rightarrow \mathbb{Z}_{kN} \rightarrow \left(F \cong U(1)\right) \rightarrow \left(\mathcal{F} \cong U(1)\right) \rightarrow 1. \tag{14}$$

A minimal choice is $k = 1$, in which case every representation of $F$ is realized by some genuine or non-genuine local operator, but for $k > 1$, there exist representations of $F$ not realized by local operators. We have $\mathcal{Z} = \mathbb{Z}_{kN}$ and the obstruction class is $w_2 = c_1 \pmod{kN}$. The case $N = 2, k = 1$ is recovered from the previous example by restricting to maximal tori.

The case of general $N$ is realized in a 3d $U(1)$ gauge theory with $N$ matter fields of charge 1 under the $U(1)$ gauge group. This theory has a magnetic 0-form symmetry $\mathcal{F} = U(1)$ which acts on monopole operators for the gauge group, which are genuine local operators. But, as discussed in later sections, one can also consider monopole operators for the electric $PSU(N)$ 0-form symmetry group rotating the matter fields, which induce monopole configurations for the $U(1)$ gauge group that are fractional but quantized in units of $1/N$. Such monopole operators are non-genuine local operators whose charge under magnetic $\mathcal{F} = U(1)$ 0-form symmetry group is also fractional but quantized in units of $1/N$.

**Correlators with non-genuine operators.** The consistency of correlation functions of non-genuine local operators as a function of $w_2$ provides information about the interaction of extended defects attached to non-genuine local operators with the $\mathcal{F}$-background. For example, consider non-genuine local operators arising at the ends of genuine line defects. To each line

(1)

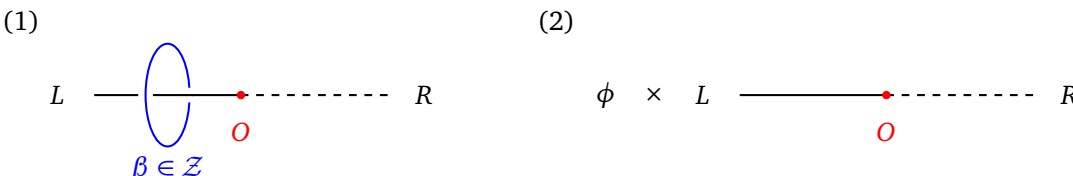

(2)

(3)

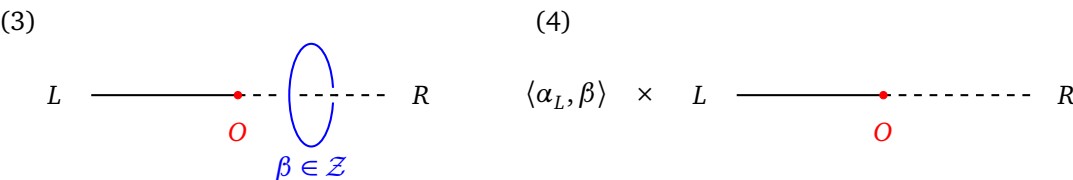

(4)

Figure 3: The figure studies correlation functions involving a non-genuine local operator $O$ arising at the end of a line defect $L$ and transforming in a representation $R$ of $F$. The operator is attached to a background Wilson line in representation $R$ whose locus is displayed as a dashed line. The representation $R$ carries a charge $\alpha_L \in \widehat{\mathcal{Z}}$ under $\mathcal{Z}$. In blue is shown a piece in the $(d-2)$-cycle Poincaré dual to the background 2-cocycle $w_2$ defined in the text, which carries an element $\beta \in \mathcal{Z}$. (1) and (3) are the same correlation functions as we have just moved $\beta$ without crossing any objects. This correlation function can be related to the correlation function without the $\beta$ insertion in two different ways. Starting from the configuration (1), we can collapse $\beta$ onto $L$ to reach the configuration (2), where $\phi$ is an a priori unknown phase obtained by moving $\beta$ across $L$. Or starting from (3), we can collapse $\beta$ onto the background Wilson line $R$ to reach the configuration (4), where we know that moving $\beta$ across the $R$ line changes correlation function by the phase $\langle \alpha_L, \beta \rangle$ using the natural map $\langle \cdot, \cdot \rangle : \widehat{\mathcal{Z}} \times \mathcal{Z} \to U(1)$. Consistency demands that the two phase factors must be equal leading to the determination of $\phi$, which is $\phi = \langle \alpha_L, \beta \rangle$. This justifies the correlation function jump (15).

defect $L$ that can end, we can associate an element $\alpha_L \in \widehat{\mathcal{Z}}$ where $\widehat{\mathcal{Z}}$ denotes the Pontryagin dual of $\mathcal{Z}$. A non-genuine local operator appearing at the end of $L$ must transform in a representation $R$ of $F$ whose charge under the center subgroup $\mathcal{Z}$ is $\alpha_L$.

Now consider a correlation function $\langle L \cdots \rangle_C$ in which the line defect $L$ is placed along a closed loop $C$, whose Poincaré dual $(d-1)$-cocycle is $\delta_C$. This lets us define a $\widehat{\mathcal{Z}}$-valued $(d-1)$-cocycle $\alpha_L \otimes \delta_C$. Then, the correlation function transforms as

$$\langle L \cdots \rangle_C (w_2 + \delta \lambda_1) = \exp\left( 2\pi i \int_M (\alpha_L \otimes \delta_C) \cup \lambda_1 \right) \times \langle L \cdots \rangle_C (w_2), \tag{15}$$

as a function of the background field $w_2$, where the cup product is defined via the natural map $\widehat{\mathcal{Z}} \times \mathcal{Z} \to \mathbb{R}/\mathbb{Z}$. A justification for this result is provided in figure 3.

The above jump (15) in the correlation function looks similar to those encountered when discussing 't Hooft anomalies. In fact, as we discuss in section 5, the above correlation function jumps can be directly connected to 't Hooft anomalies when the line defect $L$ is related to a generalized global symmetry.

## 2.2 1-form symmetries

We say that a theory has a 1-form symmetry group $\Gamma^{(1)}$, which is an Abelian group, if we are provided a (not necessarily injective) map $\mathcal{S}$ from $\Gamma^{(1)}$ to genuine invertible codimension-two

$$L \quad \overline{\qquad\bigodot\qquad} \quad = \quad \widehat{\gamma}_L(\gamma) \quad \times \quad L \quad \overline{\qquad\qquad}$$

$$\gamma \in \Gamma^{(1)}$$

Figure 4: The charges of a line defect $L$ under various elements $\gamma \in \Gamma^{(1)}$ can be described in terms of a particular element $\widehat{\gamma}_L \in \widehat{\Gamma}^{(1)}$. Here $\widehat{\gamma}_L(\gamma) \in U(1)$ describes the image of $\gamma$ under the homomorphism $\widehat{\gamma}_L : \Gamma^{(1)} \to U(1)$.

topological defects in the theory

$$\mathcal{S} : \Gamma^{(1)} \longrightarrow \{\text{Genuine invertible codimension-two topological defects}\}, \qquad (16)$$

satisfying the following properties. Let $U_\gamma$ be the topological defect associated to an element $\gamma \in \Gamma^{(1)}$ under the above map $\mathcal{S}$. Then, the fusion rule of these topological defects must be such that

$$U_\gamma \otimes U_{\gamma'} = U_{\gamma\gamma'}, \qquad (17)$$

where $\gamma\gamma'$ is the element of $\Gamma^{(1)}$ obtained as group multiplication of $\gamma$ and $\gamma'$. To provide full coupling of the theory to $\Gamma^{(1)}$ backgrounds, we also need to provide invertible non-genuine topological defects lying at various types of junctions of $U_\gamma$. The details of the choices of these higher-codimension topological defects will not be relevant for us, so we do not delve into these details here.[2] Instead, what will be relevant for us is that the map $\mathcal{S}$ is injective, and this will assumed throughout the paper.

**Charged objects.**    At the level of charged objects, there is a map $\widehat{\mathcal{S}}$ from the set of all genuine line defects of the theory to the Pontryagin dual $\widehat{\Gamma}^{(1)}$ of $\Gamma^{(1)}$,

$$\widehat{\mathcal{S}} : \{\text{Genuine line defects}\} \longrightarrow \widehat{\Gamma}^{(1)}. \qquad (18)$$

This maps a line defect $L$ to its charge $\widehat{\gamma}_L \in \widehat{\Gamma}^{(1)}$ under the 1-form symmetry $\Gamma^{(1)}$, as illustrated in figure 4.

In general, the map $\widehat{\mathcal{S}}$ is not surjective. A non-trivial example of such a situation is provided by the non-trivial 4d SPT phase protected by $\mathbb{Z}_2$ 1-form symmetry,[3] where the only genuine line defect is the identity line, which has a trivial charge under the $\mathbb{Z}_2$ 1-form symmetry.

However, when $\mathcal{S}$ is injective then $\widehat{\mathcal{S}}$ is expected to be surjective. This will be the case in situations considered throughout this paper. In such situations, $\Gamma^{(1)}$ can be constructed by studying equivalence classes of genuine line defects modulo screenings in the theory. Let us describe this construction below.

**Equivalence classes of genuine line defects.**    Define an equivalence relation $\sim$ on the set of genuine line defects, under which $L_1 \sim L_2$ if there exists a non-zero local operator $O$ living

---

[2]For the interested reader, let us remark that this corresponds to lifting the map $\mathcal{S}$ to a functor from a $(d-2)$-category $\mathcal{C}_{\Gamma^{(1)}}$ with objects, morphisms and higher-morphisms labeled by collections of elements in the group $\Gamma^{(1)}$ to the $(d-2)$-category describing invertible codimension-two topological defects and invertible topological junctions between them.

[3]The partition function of this SPT phase on a compact 4-manifold $M_4$ with 1-form symmetry background $B_2$ is $\exp\left(\frac{\pi i}{2} \int_{M_4} \mathcal{P}(B_2)\right)$ where $\mathcal{P}(B_2)$ is the Pontryagin square of $B_2$.

at the junction of $L_1$ and $L_2$:

$$L_1 \sim L_2 \iff \text{there exists} \quad L_1 \xrightarrow{\hspace{1cm}} \underset{O \neq 0}{\bullet} \xrightarrow{\hspace{1cm}} L_2,$$

(19)

$$\text{or} \qquad L_2 \xrightarrow{\hspace{1cm}} \underset{O \neq 0}{\bullet} \xrightarrow{\hspace{1cm}} L_1.$$

There is a commutative (in $d \geq 3$) product operation on the set of equivalence classes descending from the OPE/fusion of line defects. In many Lagrangian and non-Lagrangian theories of interest the set of equivalence classes forms an Abelian group under this product operation.[4] We study only such theories in this paper. Let us denote the Abelian group formed by the equivalence classes by $\widehat{\Gamma^{(1)}}$.

Now we define invertible topological codimension-2 defects by providing the charges of genuine line defects under them. As explained in figure 5, two genuine line defects $L_1$ and $L_2$ lying in the same equivalence class need to have the same charge under each invertible topological codimension-2 defect. Thus, in this fashion, we can define invertible topological codimension-2 defects in one-to-one correspondence with elements of the Pontryagin dual of $\widehat{\Gamma^{(1)}}$

$$\Gamma^{(1)} = \text{Hom}\left(\widehat{\Gamma^{(1)}}, U(1)\right).$$

(20)

We say that the theory has a 1-form symmetry group $\Gamma^{(1)}$, even though it might be possible to couple the theory to a 1-form symmetry group $\mathcal{O}$ larger than[5] $\Gamma^{(1)}$ by having a non-injective map $\mathcal{S}$ from $\mathcal{O}$ to $\Gamma^{(1)}$. The 1-form symmetry group acts on the line defects belonging to equivalence classes in $\widehat{\Gamma^{(1)}}$ by the standard pairing $\Gamma^{(1)} \times \widehat{\Gamma^{(1)}} \to U(1)$.

**Examples.** Examples encountered in this paper will largely involve 1-form symmetries involving (products of) cyclic groups. This arises when equivalences classes of genuine line defects are labelled by an integer modulo $q$ and fusion corresponds to addition modulo $q$, for some integer $q > 1$. This is the additive Abelian group $\widehat{\Gamma^{(1)}} = \mathbb{Z}/q\mathbb{Z} = \mathbb{Z}_q$ and the corresponding 1-form symmetry is also $\Gamma^{(1)} = \mathbb{Z}_q$. A concrete example is provided by a $U(1)$ gauge theory with a matter field of charge $q$. As discussed before, the matter field leads to a gauge invariant non-genuine local operator sitting at the end of a Wilson line of charge $q$. Thus the Wilson lines form $\widehat{\Gamma^{(1)}} = \mathbb{Z}_q$.

## 2.3 2-group symmetries

**Type of 2-group symmetries studied here.** Intuitively, 2-group symmetries involve a mixing between 0-form and 1-form symmetries. Here we focus on 2-group symmetries where the 0-form symmetry part $\mathcal{F}$ is continuous, the 1-form symmetry $\Gamma^{(1)}$ is discrete and there is no action of 0-form symmetry on 1-form symmetry. Such a 2-group is a specified by a triple

$$\left(\mathcal{F}, \Gamma^{(1)}, \Theta\right),$$

(21)

where

$$\Theta \in H^3(B\mathcal{F}, \Gamma^{(1)}),$$

(22)

---

[4]Examples where this is not the case are provided by gauge theories with disconnected gauge groups, like pure $O(2)$ gauge theory.

[5]For example, we can regard a trivial theory to have any 1-form symmetry $\mathcal{O}$ by mapping all elements of $\mathcal{O}$ to the trivial codimension-two defect in the trivial theory.

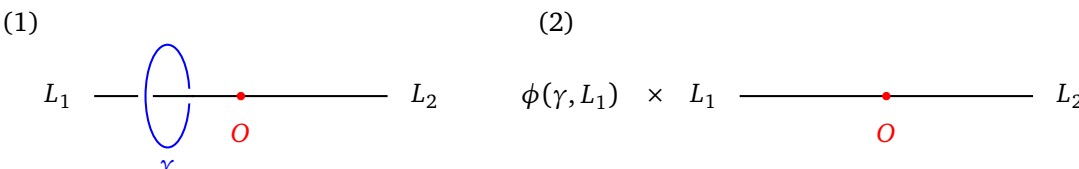

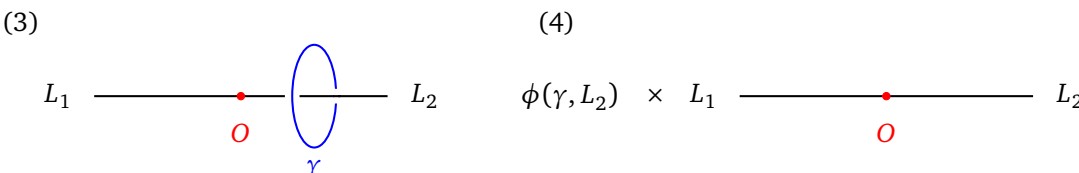

Figure 5: The figure studies correlation functions involving a non-genuine local operator $O$ arising at a junction between two line defects $L_1$ and $L_2$. In blue is shown a topological codimension-two defect corresponding to a 1-form symmetry element $\gamma$. (1) and (3) are the same correlation functions as we have just moved $\gamma$ without crossing any objects. This correlation function can be related to the correlation function without the $\gamma$ insertion in two different ways. Starting from the configuration (1), we can collapse $\gamma$ onto $L_1$ to reach the configuration (2). In the process, we generate an additional phase $\phi(\gamma, L_1)$ which is the charge of $L_1$ under $\gamma$. Or starting from (3), we can collapse $\gamma$ onto $L_2$ to reach the configuration (4), which generates a phase $\phi(\gamma, L_2)$ which is the charge of $L_2$ under $\gamma$. Consistency demands that the two phase factors must be equal. Thus two line defects lying in the same equivalence class have same charge under all 1-form symmetries, and hence the possible 1-form symmetries form a group isomorphic to the Pontryagin dual of the group $\widehat{\Gamma}^{(1)}$ of equivalence classes of line defects.

is known as the *Postnikov class* associated to the 2-group symmetry. If $\Theta$ is trivial, then there is no 2-group symmetry.

The triple specifies the following relationship between the background fields

$$\delta B_2 + B_1^* \Theta = 0, \tag{23}$$

where the background field $B_2$ for 1-form symmetry is a $\Gamma^{(1)}$-valued 2-cochain, $B_1 : M \to B\mathcal{F}$ captures the 0-form symmetry background, and $B_1^* \Theta$ is the pullback of the Postnikov class $\Theta$. In particular, in the presence of a background field for the 0-form symmetry, the background field for the 1-form symmetry need not be closed.

**Bockstein homomorphism.** For the 2-group symmetries studied in this paper, the Postnikov class admits a construction of the following form

$$\Theta = \text{Bock}(w_2), \tag{24}$$

where

$$\text{Bock} : H^2(B\mathcal{F}, \mathcal{Z}) \to H^3(B\mathcal{F}, \Gamma^{(1)}), \tag{25}$$

is the Bockstein homomorphism associated to a short exact sequence

$$0 \to \Gamma^{(1)} \to \mathcal{E} \to \mathcal{Z} \to 0, \tag{26}$$

where $w_2 \in H^2(B\mathcal{F}, \mathcal{Z})$ is the characteristic class capturing the obstruction of lifting $\mathcal{F}$ bundles to $F = \mathcal{F}/\mathcal{Z}$ bundles.

**Equivalence classes of lines.** The physical meaning of such 2-group symmetries becomes more transparent once the 2-groups are encoded in the properties of the spectrum of (extended) operators in the theory, which we now turn to.

The 2-group symmetries under discussion are related to the physical phenomenon that non-genuine local operators living at the junctions of genuine line defects may not form allowed representations of the 0-form symmetry group $\mathcal{F}$. Let $F$ be a central extension of $\mathcal{F}$ under which all non-genuine local operators form allowed representations. Then we can express $\mathcal{F}$ as

$$\mathcal{F} = F/\mathcal{Z}, \tag{27}$$

where $\mathcal{Z}$ is a finite subgroup of the center $Z_F$ of $F$. This defines the auxiliary groups $F$ and $\mathcal{Z}$ appearing in the construction (24) of the Postnikov class.

The computation of the 2-group symmetry is similar to that of 1-form symmetry discussed in section 2.2, but we now define equivalence classes of lines while taking into account the representations under $F$ of the non-genuine local operators lying at the junctions between genuine line defects. To keep track of the representations of junction local operators, we introduce background Wilson lines valued in representations of $F$. A (genuine or non-genuine) local operator transforming in representation $R$ of $F$ is attached to a background Wilson line in representation $R$.

We can now specify the equivalence relation. Consider the set of objects $(L, R)$, where $L$ is a genuine line defect and $R$ is a background Wilson line for $F$. Then impose the equivalence relation[6]

$$(L_1, R_1) \sim (L_2, R_2) \iff \text{there exists} \quad L_1 \xrightarrow[\;O \in R_2 \otimes R_1^*\;]{} L_2,$$

$$\text{or} \quad L_2 \xrightarrow[\;O \in R_1 \otimes R_2^*\;]{} L_1. \tag{28}$$

In other words, $(L_1, R_1) \sim (L_2, R_2)$ if there is a non-genuine local operator $O$ living at the junction of $L_1$ and $L_2$, which transforms in the representation $R_2 \otimes R_1^*$ or $R_1 \otimes R_2^*$ (depending on the orientation of $L_1$ and $L_2$), where $R^*$ denotes the complex conjugate representation of $R$. The set $\widehat{\mathcal{E}}$ of equivalence classes admits a commutative (in $d \geq 3$) product structure obtained by combining OPE of line defects with the tensor product on representations. In this paper, we study theories for which $\widehat{\mathcal{E}}$ forms an Abelian group under this product operation.

**Computation of short exact sequence (26).** Note that elements of the form $(\mathrm{id}, R)$ where id denotes the identity line defect lead to equivalence classes forming the subgroup $\widehat{\mathcal{Z}} \subset \widehat{\mathcal{E}}$, leading to a short exact sequence

$$0 \to \widehat{\mathcal{Z}} \to \widehat{\mathcal{E}} \to \widehat{\Gamma^{(1)}} \to 0. \tag{29}$$

Indeed, as claimed above, the group $\widehat{\mathcal{E}}/\widehat{\mathcal{Z}}$ is $\widehat{\Gamma^{(1)}}$ because modding out by elements of $\widehat{\mathcal{Z}}$ corresponds to forgetting the data of the background Wilson lines, thus reducing the computation to equivalence classes of line defects without regard for 0-form charges of junction local operators, as considered in section 2.2.

The short exact sequence (26) associated to the 2-group symmetry (24) is the Pontryagin dual of the short exact sequence (29). The argument relating the above construction in terms of equivalence classes of lines to the relationship

$$\delta B_2 + \mathrm{Bock}(w_2) = 0, \tag{30}$$

---

[6]We use the same notation $\sim$ as in (19), however it should always be clear, which equivalence relation is meant, from the context of the types of objects it relates.

between the background fields can be found in section 2.3 of [44].

**Example.** An example of such a 2-group symmetry has 0-form symmetry $\mathcal{F} = SO(3)$, 1-form symmetry $\Gamma^{(1)} = \mathbb{Z}_2$, and Postnikov class $\Theta = \text{Bock}(w_2) = w_3$ given by the image of the second Stiefel-Whitney class under the Bockstein homomorphism for the short exact sequence

$$0 \to \mathbb{Z}_2 \to \mathbb{Z}_4 \to \mathbb{Z}_2 \to 0. \tag{31}$$

This coincides with the third Stiefel-Whitney class of the $SO(3)$-bundle.

This scenario is realized in a $U(1)$ gauge theory with two matter fields $\phi_1$ and $\phi_2$ of charge 2, which transform as a doublet under an $\mathfrak{su}(2)$ flavor symmetry algebra. This combines the $U(1)$ gauge theory examples discussed in the previous two subsections. Here we have $\mathcal{F} = SO(3)$ as gauge invariant operators composed out of $\phi_i$ all have integer spin under $\mathfrak{su}(2)$, and we have $\Gamma^{(1)} = \mathbb{Z}_2$ as matter fields $\phi_i$ lead to gauge invariant non-genuine local operators sitting at the ends of a Wilson line defect of gauge charge 2. As $\phi_i$ form a doublet of $F = SU(2)$, we learn that the Wilson line of charge 2 is equivalent to a background Wilson line in fundamental representation of $F = SU(2)$, which corresponds to the non-trivial element inside $\widehat{\mathcal{Z}} = \mathbb{Z}_2$. Thus, we find that the equivalence classes of lines form $\widehat{\mathcal{E}} = \mathbb{Z}_4$, leading to the short exact sequence (31) and the non-trivial Postnikov class $\Theta = w_3$.

**Solitonic Postnikov defects and 1-form twisted sector.** An alternative way of characterizing a 2-group symmetry is via identifications between the following two classes of defects:

- On the one hand, we have twisted sector operators for 1-form symmetry, which are non-genuine (generically non-topological) codimension-3 defects that lie at the ends of the genuine codimension-2 topological defects implementing the 1-form symmetry.

- On the other hand, we have codimension-3 solitonic defects $P$ inducing 0-form symmetry backgrounds such that

$$\int_{D_3} \Theta \neq 0, \tag{32}$$

  where $D_3$ is a small 3-dimensional ball intersecting the locus of $P$ at a single point, and $\Theta$ is the Postnikov class discussed above.

The 2-group symmetry then states the following. Consider a solitonic defect $P$ inducing a 0-form symmetry background such that

$$\int_{D_3} \Theta = \gamma \in \Gamma^{(1)}. \tag{33}$$

Then, $P$ is a non-genuine defect lying in the twisted sector for 1-form symmetry element $\gamma \in \Gamma^{(1)}$. Moreover, any defect lying in the twisted sector for $\gamma$ has to be a solitonic defect for 0-form symmetry inducing (33).

In particular, in every such $\gamma$-twisted sector, we have a topological codimension-3 defect. These codimension-3 defects are placed along the $(d-3)$-cycle Poincare dual to the 3-cocycle $B_1^* \Theta$ on spacetime $M_d$, providing the topological defect realization for the relationship (23) describing 2-group backgrounds.

**Background field for 2-group symmetry.** In this paper, we will use extensively an $\mathcal{E}$-valued background field associated to a 2-group symmetry. This is an $\mathcal{E}$-valued 2-cocycle $B_w$ defined as the following combination [44,50]

$$B_w = i(B_2) + \widetilde{w}_2 \, . \tag{34}$$

Here $i(B_2)$ denotes the $\mathcal{E}$-valued 2-cochain obtained from the $\Gamma^{(1)}$-valued 2-cochain $B_2$ using the injection map

$$i : \Gamma^{(1)} \to \mathcal{E} \, , \tag{35}$$

and $\widetilde{w}_2$ is an $\mathcal{E}$-valued 2-cochain obtained by lifting the $\mathcal{Z}$-valued 2-cocycle $w_2$ under the projection map

$$\pi : \mathcal{E} \to \mathcal{Z} \, , \tag{36}$$

appearing in the short exact sequence (26). Both of the $\mathcal{E}$-valued co-chains $i(B_2)$ and $\widetilde{w}_2$ are not closed if there is a non-trivial 2-group symmetry, but the combination $B_w$ is always closed [44,50].

In the aforementioned example with $\mathcal{F} = SO(3)$, $\Gamma^{(1)} = \mathbb{Z}_2$ and $\Theta = \mathrm{Bock}(w_2) = w_3$, the 2-group background can be written $B_w = 2B_2 + \widetilde{w}_2$ where $\widetilde{w}_2$ is a $\mathbb{Z}_4$-valued co-chain lift of the second Stiefel-Whitney class $w_2$ and $2B_2$ is a $\mathbb{Z}_4$ valued cochain obtained from the $\mathbb{Z}_2$ valued cochain $B_2$ by multiplying it by 2.

# 3 Solitonic defects of codimension two and three

In this section, we discuss some types of codimension-2 and codimension-3 solitonic defects that induce backgrounds for 0-form, 1-form and 2-group symmetries in their vicinity. These include vortex and monopole defects, which are solitonic defects inducing vortex and monopole configurations for the 0-form symmetry group. In section 5, we will explain the utility of the solitonic defects introduced here in determining 't Hooft anomalies between 0-form, 1-form and 2-group symmetries in $d = 3$.

## 3.1 Codimension-two solitonic defects

**Vortex defects.** A *vortex defect* $V$ is a solitonic defect which induces a vortex configuration for the 0-form symmetry group $\mathcal{F}$. Such a vortex configuration for $\mathcal{F}$ is specified by a *co-character* of $\mathcal{F}$, which is a homomorphism

$$\phi : U(1) \to \mathcal{F} \, . \tag{37}$$

On a small circle $S^1$ linking the $(d-2)$-dimensional locus $\mathcal{L}_{d-2} \subset M_d$ occupied by $V$ in space-time $M_d$, the co-character $\phi$ defining the vortex configuration is such that the background connection for $U(1)_\phi := \mathrm{Im}\,\phi \subset \mathcal{F}$ has a winding number 1. In other words,

$$\int_{D_2} c_1 = 1 \, , \tag{38}$$

where $D_2$ is a small disk intersecting $\mathcal{L}_{d-2}$ at one point, whose boundary is the above small circle $S^1$, and $c_1$ is the first Chern class of the background $U(1)_\phi$ bundle. If a codimension-two defect $V$ induces a vortex configuration with co-character $\phi$, we say it lies in the *vortex sector* $\phi$.

To any codimension-two defect $V$ in the vortex sector $\phi$, we can associate an element

$$\alpha_\phi \in \mathcal{Z} \, , \tag{39}$$

which is the obstruction to lifting the co-character $\phi$ of $\mathcal{F}$ to a co-character of $F$. The physical interpretation of $\alpha_\phi$ is that the $\mathcal{Z}$-valued background field $w_2$ capturing the obstruction of lifting the background $\mathcal{F}$ bundle to an $F$ bundle is forced to satisfy

$$\int_{D_2} w_2 = \alpha_\phi \,. \tag{40}$$

**Examples.** An example is a flavor symmetry $\mathcal{F} = SO(3)$ where non-genuine local operators transform in representations of $F = SU(2)$. As we discussed earlier, a concrete example of such a scenario is provided by a $U(1)$ gauge theory with two matter fields of gauge charge 1. In this scenario, vortex sectors are labelled by an integer flux $\phi \in \mathbb{Z}$ and the element $\alpha_\phi = \phi \bmod 2$ captures the obstruction to lifting this to an $SU(2)$ vortex configuration.

As another example, we can suppose that there is a flavor symmetry $\mathcal{F} = U(1)$, but non-genuine local operators transform with fractional charge in multiples of $1/N$, such that $F = U(1)$ is an $N$-fold cover. As we discussed earlier, a concrete example of such a scenario is provided by magnetic $U(1)$ 0-form symmetry in 3d $U(1)$ gauge theory with $N$ matter fields of gauge charge 1. In this scenario, vortex sectors are labelled by an integer $\phi \in \mathbb{Z}$ and the obstruction class is $\alpha_\phi = \phi \bmod N$.

**Codimension-2 defects inducing 1-form backgrounds.** Similarly, we can consider codimension-2 solitonic defects inducing 1-form symmetry backgrounds in their vicinity. Such a defect forces the background field $B_2$ for 1-form symmetry $\Gamma^{(1)}$ to satisfy

$$\int_{D_2} B_2 = \alpha \in \Gamma^{(1)} \,, \tag{41}$$

where $D_2$ is again a small disk intersecting the codimension-2 locus of the defect at a single point.

Examples are provided by topological codimension-two defects implementing the 1-form symmetry $\Gamma^{(1)}$. In section 4, we will see that examples of such solitonic defects are also provided by codimension-2 defects inducing gauge vortex configurations in their vicinity.

**Codimension-2 defects inducing 2-group backgrounds.** Combining the two constructions above, we can consider codimension-two solitonic defects inducing both 0-form and 1-form backgrounds, or in other words 2-group backgrounds, in their vicinity. Such a defect $V$ forces the background field $B_w$ for the 2-group symmetry to satisfy

$$\int_{D_2} B_w = \alpha_V \in \mathcal{E} \,. \tag{42}$$

We can therefore associate the element $\alpha_V \in \mathcal{E}$ to the codimension-two defect $V$.

In particular, let $\alpha_V \in \mathcal{E}$ be associated to a vortex defect $V$ lying in vortex sector $\phi$ for the 0-form symmetry. Then, for consistency, we must have

$$\alpha_V \in \pi^{-1}(\alpha_\phi) \subset \mathcal{E} \,, \tag{43}$$

where $\pi$ is the projection map (36) from $\mathcal{E}$ to $\mathcal{Z}$.

In section 4, we will see that examples of such solitonic defects are provided by codimension-2 defects inducing mixed vortex configurations for gauge and 0-form symmetries in their vicinity.

### 3.2 Codimension-three solitonic defects

**Monopole operators.** Now consider a codimension-three defect $M$ arising at the end of a codimension-two defect $V$ in vortex sector $\phi$. Let $\mathcal{L}_{d-3}$ be the codimension-three locus along with $M$ is placed. Then, along a small sphere $S^2$ linking $\mathcal{L}_{d-3}$, we find an induced

$$\int_{S^2} c_1 = 1 \,, \tag{44}$$

where $c_1$ is the first Chern class for the subgroup of $U(1)_\phi$ of $\mathcal{F}$. If $\phi$ is non-trivial, then such a codimension-three solitonic defect is called a *monopole operator* or a *'t Hooft defect* for the 0-form symmetry $\mathcal{F}$.

To any such monopole operator $M$ we can associate an element $\alpha_\phi \in \mathcal{Z}$, which is the obstruction of lifting $\phi$ to a co-character for $F$. The monopole operator then requires the $\mathcal{Z}$-valued 0-form symmetry background field $w_2$ to satisfy

$$\int_{S^2} w_2 = \alpha_\phi \in \mathcal{Z} \,. \tag{45}$$

**Codimension-3 defects inducing 1-form backgrounds.** In the presence of a 1-form symmetry we may also consider codimension-three defects $M$ lying at the end of codimension-two defects that induce a 1-form background as in equation (41). Such a codimension-3 defect requires that the background field for the 1-form symmetry satisfies

$$\int_{S^2} B_2 = \alpha \in \Gamma^{(1)} \,. \tag{46}$$

This applies in particular to the topological codimension-two defects generating the 1-form symmetry, in which case such codimension-three defects are known as twisted sector operators for the 1-form symmetry $\Gamma^{(1)}$. In section 4, we will see that examples of such solitonic defects are also provided by codimension-3 defects inducing gauge monopole configurations in their vicinity.

**Codimension-3 defects inducing 2-group backgrounds.** In the presence of a 2-group symmetry, a combination of the above leads us to consider codimension-three defects lying at the end of a codimension-two defect $V$ that induces a 2-group background as in equation (42). Correspondingly, such a codimension-three defect $M$ also induces a 2-group background in its vicinity, such that

$$\int_{S^2} B_w = \alpha_V \in \mathcal{E} \,. \tag{47}$$

In other words, $M$ is associated to the element $\alpha_V \in \mathcal{E}$.

For consistency, $\alpha_V \in \mathcal{E}$ associated to a monopole operator $M$ associated to a co-character $\phi$ of $\mathcal{F}$ must satisfy

$$\alpha_V \in \pi^{-1}(\alpha_\phi) \subset \mathcal{E} \,, \tag{48}$$

where $\alpha_\phi$ is the obstruction of lifting $\phi$ to a co-character for $F$ and $\pi$ is the map (36).

In section 4, we will see that examples of such solitonic defects are provided by codimension-3 defects inducing mixed monopole configurations for gauge and 0-form symmetries in their vicinity.

# 4   Gauging symmetries and the fate of solitonic defects

In this section we consider gauging parts of the 0-form and 1-form symmetries in a theory $\mathfrak{T}$ admitting a 2-group symmetry $\left(\mathcal{F}, \Gamma^{(1)}, \text{Bock}(w_2)\right)$ of the type discussed above. We first consider gauging a part of the 0-form symmetry $\mathcal{F}$, assuming that there is no 2-group structure in section 4.1, which is then generalised to a 0-form part of a 2-group in section 4.2. Moreover, initially we discuss gauging such a symmetry in a generic dimension $d \neq 3$, before outlining the additional features arising in dimension $d = 3$. Finally, we will discuss gauging the 1-form symmetry $\Gamma^{(1)}$. We also discuss the fate of solitonic defects introduced in the previous section under these gauging procedures.

Throughout this section, we will assume for simplicity that these symmetries are not afflicted with any 't Hooft anomalies. Later in section 5, we will consider gauging in the presence of 't Hooft anomalies in dimension $d = 3$.

## 4.1   Gauging 0-form symmetry in the absence of 2-groups

Here we consider a theory $\mathfrak{T}$ with a compact connected 0-form symmetry group $\mathcal{F} = F/\mathcal{Z}$ that does not participate in a 2-group. We will further assume there is no 1-form symmetry. These assumptions will be relaxed to allow the symmetry group $\mathcal{F}$ to participate in a 2-group symmetry in section 4.2.

**Gauge group.**   Now consider gauging a sub-algebra $\mathfrak{g} = \mathfrak{f}_g \subset \mathfrak{f}$ with gauge group $\mathcal{G} = F_g$ which is a compact connected Lie group.[7] This gauge group must be such that all the genuine and non-genuine local operators of the original theory $\mathfrak{T}$ transform in allowed representations of $\mathcal{G}$. We denote the theory obtained by gauging this symmetry by $\mathfrak{T}/\mathcal{G}$. Note that this encompasses standard gauge theories constructed from matter transforming in linear representations of the gauge group, but could apply equally well if $\mathfrak{T}$ is an interacting theory without a Lagrangian description.

The residual 0-form symmetry algebra $\mathfrak{f}_r \subseteq \mathfrak{f}$ is the commutant of $\mathfrak{f}_g$ in $\mathfrak{f}$. Let $F_r$ be a compact connected Lie group with Lie algebra $\mathfrak{f}_r$ such that all genuine and non-genuine local operators in $\mathfrak{T}$ transform in allowed representations of $F_r$. The 0-form symmetry group $\mathcal{F}_{gr}$ associated to the 0-form symmetry subalgebra $\mathfrak{f}_g \oplus \mathfrak{f}_r \subseteq \mathfrak{f}$, which is the maximal group that acts faithfully on genuine local operators before gauging, can then be written as

$$\mathcal{F}_{gr} = \frac{F_g \times F_r}{\mathcal{Z}_{gr}}, \tag{49}$$

where $\mathcal{Z}_{gr}$ is the subgroup of the center of $F_g \times F_r$ acting trivially on all genuine local operators in the original theory $\mathfrak{T}$.

The construction of the 0-form symmetries (and therefore 2-group global symmetries) of the gauged theory $\mathfrak{T}/\mathcal{G}$ will have additional features in three-dimensions due to the existence of gauge monopole local operators charged under additional 0-form topological symmetries. We first present the standard construction for $d \neq 3$, before presenting the additional features in $d = 3$.

### 4.1.1   General dimension

**Structure group.**   In dimension $d \neq 3$, the 0-form symmetry group $\mathcal{F}_{gr}$ of $\mathfrak{T}$ can be identified with the structure group of the gauged theory $\mathfrak{T}/\mathcal{G}$, which is the group whose bundles describe

---

[7]The gauging is performed by promoting the background gauge fields for $\mathfrak{f}_g$ to dynamical gauge fields. We do not add additional matter fields charged under $\mathfrak{f}_g$. The choice of the gauge group $\mathcal{G}$ is a choice of the allowed probe particles, or in other words, a choice of the possible Wilson line defects associated to $\mathfrak{f}_g$.

the most general combinations of gauge bundles and 0-form symmetry bundles that the theory can be coupled to. We write the structure group as

$$\mathcal{S} = \frac{\mathcal{G} \times F_r}{\mathcal{E}_r}, \tag{50}$$

with the identifications $\mathcal{S} = \mathcal{F}_{gr}$ and $\mathcal{E}_r = \mathcal{Z}_{gr} \subset Z_{\mathcal{G} \times F_r}$. Here and below we denote by $Z_{\mathcal{G} \times F_r}$ the center of $\mathcal{G} \times F_r$. Let us review some of the general discussion in [35].

**Symmetries.** Let us first discuss the global symmetries of the gauged theory $\mathfrak{T}/\mathcal{G}$. Define the projection maps

$$Z_{\mathcal{G}} \xleftarrow{p_g} Z_{\mathcal{G} \times F_r} \xrightarrow{p_r} Z_{F_r}, \tag{51}$$

and define

$$\mathcal{Z}_r = p_r(\mathcal{E}_r), \qquad \mathcal{Z}_g = p_g(\mathcal{E}_r). \tag{52}$$

Then the residual 0-form symmetry group is

$$\mathcal{F}_r = F_r / \mathcal{Z}_r, \qquad \mathcal{Z}_r = p_r(\mathcal{E}_r). \tag{53}$$

Note that $\mathcal{Z}_r$ is the subgroup of $F_r$ acting by gauge transformations on genuine local operators in the original theory $\mathfrak{T}$. In addition, there is a new electric 1-form symmetry

$$\Gamma_r^{(1)} = Z_{\mathcal{G}} \cap \mathcal{E}_r, \tag{54}$$

which is the subgroup of the center $Z_{\mathcal{G}}$ of the group $\mathcal{G}$ leaving invariant all genuine local operators in the original theory $\mathfrak{T}$. These groups form a natural short exact sequence

$$0 \longrightarrow \Gamma_r^{(1)} \longrightarrow \mathcal{E}_r \longrightarrow \mathcal{Z}_r \longrightarrow 0. \tag{55}$$

If this sequence does not split, there is a non-trivial extension class, which determines the 2-group global symmetry of $\mathfrak{T}/\mathcal{G}$.

**Backgrounds.** Let us now consider coupling $\mathfrak{T}/\mathcal{G}$ to background fields for this 2-group symmetry. They may be summarised as follows:

- A background $\mathcal{F}_r$-bundle with associated $\mathcal{Z}_r$-valued 2-cocyle $w_2^r$ obstructing the lift to an $F_r$ bundle.

- A background field for the 1-form symmetry is a $\Gamma_r^{(1)}$-valued 2-cochain $B_2^r$ satisfying

$$\delta B_2^r = \mathrm{Bock}(w_2^r). \tag{56}$$

- The above background fields $w_2^r, B_2^r$ combine to form a background field for the 2-group, which is an $\mathcal{E}_r$ valued 2-cocycle $B_w^r$.

**Gauge bundles.** Let us now describe which gauge bundles are summed over in $\mathfrak{T}/\mathcal{G}$ in the presence of background fields. We first introduce the notation

$$G = \mathcal{G}/\mathcal{Z}_g, \qquad \mathcal{Z}_g = p_g(\mathcal{E}_r). \tag{57}$$

Note that $G = \mathcal{F}_g$ is the 0-form symmetry group associated to $\mathfrak{g} = \mathfrak{f}_g$ acting faithfully on genuine local operators in the original theory $\mathfrak{T}$. In the presence of background fields for the 2-group symmetry, the gauged theory $\mathfrak{T}/\mathcal{G}$ sums over $G$-bundles constrained such that

$$w_2^g = p_g\left(B_w^r\right), \tag{58}$$

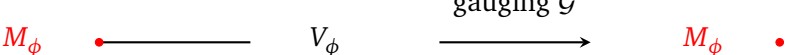

$$M_\phi \quad \bullet \!\!\!\rule[0.5ex]{6em}{0.4pt}\!\!\! \quad V_\phi \qquad \xrightarrow{\text{gauging } \mathcal{G}} \qquad M_\phi \quad \bullet$$

Figure 6: Consider a vortex defect $V_\phi$ labelled by a cocharacter $\phi : U(1) \to \mathcal{G}$ in the original theory $\mathfrak{T}$ that descends to the identity defect in the gauged theory $\mathfrak{T}/\mathcal{G}$. In the process, a monopole operator $M_\phi$ living at the end of $V_\phi$ becomes a genuine non-fractional gauge monopole operator for the gauge group $\mathcal{G}$.

where $w_2^g$ is the $\mathcal{Z}_g$-valued 2-cocyle obstructing the lift of a $G$-bundle to a $\mathcal{G}$-bundle. As a consistency check, in the absence of background fields for global symmetries this reproduces the expectation that the gauged theory sums over $\mathcal{G}$-bundles.

Note that the combination of an $\mathcal{F}_r$-bundle and a $G$-bundle with obstruction class $w_2^g = p_g(B_w^r)$ gives rise to a bundle for the structure group $\mathcal{S}$ with the $\mathcal{E}_r$-valued co-cycle $B_w^r$ obstructing the lift to a $\mathcal{G} \times F_r$-bundle.

**Fate of vortices and monopoles under gauging.** With the description of the structure group, background fields and summation over gauge bundles in hand, we can now describe the fate of the vortex and monopole defects introduced in section 3 in the theory $\mathfrak{T}$ when gauging the symmetry $\mathcal{G}$.

Consider a vortex defect $V$ of the original theory $\mathfrak{T}$ inducing a vortex configuration for the 0-form symmetry group $\mathcal{F}_{gr}$. It is convenient to specify such a vortex configuration by a pair of co-characters $(\phi_g, \phi_r)$ for $G \times \mathcal{F}_r$ together with obstruction classes $(\alpha_g, \alpha_r)$ valued in $\mathcal{Z}_g \times \mathcal{Z}_r$ such that $\alpha_g = p_g \alpha$ and $\alpha_r = p_r \alpha$, where $\alpha$ is obstruction of lifting the $\mathcal{F}_{gr}$ vortex configuration to a $\mathcal{G} \times F_r$ vortex configuration.

Consider first the situation where co-character $\phi$ has non-zero components only along the $G$ direction, namely $\phi = (\phi_g, 0)$. This is equivalent to a co-character $\phi_g : U(1) \to \mathcal{G}/\Gamma_r^{(1)}$ and in such a case $\alpha_g \in \Gamma_r^{(1)}$. Let us begin with the case $\alpha_g = 0$, which corresponds to a situation in which $\phi_g$ can be lifted to a co-character for the gauge group

$$\phi_g : U(1) \to \mathcal{G}. \tag{59}$$

Then the vortex defect $V$ descends to a codimension-two defect in the gauged theory $\mathfrak{T}/\mathcal{G}$ which induces a vortex configuration for the gauge group $\mathcal{G}$. Since, such a configuration may be removed by a gauge transformation, $V$ descends to a codimension-two defect in the trivial vortex sector (for 0-form symmetry $\mathcal{F}_r$) in the gauged theory $\mathfrak{T}/\mathcal{G}$. For example, it may descend to the identity codimension-two defect.

A monopole operator $M$ in the original theory $\mathfrak{T}$ living at the end of the vortex defect $V$ descends to a codimension-three defect in the gauged theory $\mathfrak{T}/\mathcal{G}$ which induces a monopole configuration for the gauge group $\mathcal{G}$ associated to the co-character $\phi_g : U(1) \to \mathcal{G}$. We call an operator inducing a monopole configuration for the gauge group $\mathcal{G}$ as a *non-fractional gauge monopole operator*. If such an operator lives at the end of a codimension-two defect, then it is called a non-genuine non-fractional gauge monopole operator. Thus, if $V$ descends to a non-trivial codimension-2 defect, then $M$ descends to a non-genuine non-fractional gauge monopole operator in the theory $\mathfrak{T}/\mathcal{G}$. On the other hand, a non-fractional gauge monopole operator that does not live at the end of any codimension-2 defect is called a genuine non-fractional gauge monopole operator. These are the familiar monopole operators usually studied in the literature. Thus, if $V$ descends to the identity codimension-two defect, then $M$ descends to a genuine codimension-three non-fractional gauge monopole operator. See figure 6.

Now suppose that the obstruction class $\alpha_g \in \Gamma_r^{(1)}$ is non-vanishing and consequently $\phi_g : U(1) \to \mathcal{G}/\Gamma_r^{(1)}$ cannot be lifted to a co-character for the group $\mathcal{G}$. Then $V$ descends

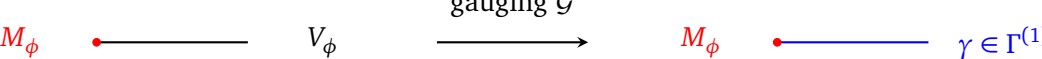

Figure 7: In the original theory $\mathfrak{T}$, $V_\phi$ is a vortex defect associated to a co-character $\phi : U(1) \to G$ with non-trivial obstruction $\alpha$ for lifting to a co-character for $\mathcal{G}$, which descends to the Gukov-Witten operator implementing the 1-form symmetry $\gamma \in \Gamma_r^{(1)}$ in the gauged theory $\mathfrak{T}/\mathcal{G}$. In the process, a monopole operator $M_\phi$ living at the end of $V_\phi$ becomes a twisted sector operator for the 1-form symmetry element $\gamma \in \Gamma_r^{(1)}$.

to what we call a *fractional gauge vortex defect*, which is a codimension-two defect in $\mathfrak{T}/\mathcal{G}$ that induces a vortex configuration of $\mathcal{G}/\Gamma_r^{(1)}$ that cannot be lifted to a vortex configuration of the gauge group $\mathcal{G}$. Examples of fractional gauge vortex defects are provided by *Gukov-Witten operators*, which are topological codimension-two defects implementing the 1-form symmetry $\Gamma_r^{(1)}$ in the gauged theory $\mathfrak{T}/\mathcal{G}$. Moreover, the codimension-two defect in the gauged theory $\mathfrak{T}/\mathcal{G}$, that $V$ descends to, induces a background for the 1-form symmetry $\Gamma_r^{(1)}$ such that

$$\int_{D_2} B_2^r = \alpha_g \in \Gamma_r^{(1)}. \tag{60}$$

Thus $V$ remains a solitonic defect after gauging, but now it induces a 1-form background in its vicinity instead of a 0-form background.

A monopole operator $M$ living at the end of the vortex defect $V$ in the original theory $\mathfrak{T}$ descends to what we call a *fractional gauge monopole operator*, which is a codimension-three operator in the gauged theory $\mathfrak{T}/\mathcal{G}$ living at the end of a fractional gauge vortex defect. Moreover, after gauging $M$ becomes a codimension-3 solitonic defect inducing 1-form background on a sphere $S^2$ linking it, rather than a 0-form background. Examples of fractional gauge monopole operators are provided by operators lying in the twisted sector for the 1-form symmetry $\Gamma_r^{(1)}$, which are operators lying at the ends of the topological codimension-two defects implementing the 1-form symmetry $\Gamma_r^{(1)}$. In particular, if $V$ descends to a Gukov-Witten operator implementing a 1-form symmetry in the gauge theory $\mathfrak{T}/\mathcal{G}$, then $M$ descends to a twisted sector operator for the 1-form symmetry element $\alpha_g \in \Gamma_r^{(1)}$. See figure 7.

Now consider the situation in which $\phi_r \neq 0$. If $\phi_g = 0$, then $V$ descends to a vortex defect for the $\mathcal{F}_r$ 0-form symmetry of the gauged theory $\mathfrak{T}/\mathcal{G}$ inducing the vortex configuration described by the co-character $\phi_r : U(1) \to \mathcal{F}_r$. The descending vortex defect induces a $w_2^r$ background given by

$$\int_{D_2} w_2^r = \alpha_r, \tag{61}$$

in its vicinity. A monopole operator $M$ living at the end of $V$ descends to a monopole operator for $\mathcal{F}_r$ in the gauged theory $\mathfrak{T}/\mathcal{G}$.

Finally consider the most general situation in which both $\phi_g$ and $\phi_r$ are non-trivial. In such a situation, $V$ descends to what we call a *mixed gauge/0-form vortex defect* in the gauged theory $\mathfrak{T}/\mathcal{G}$, which is a codimension-two defect inducing both a vortex configuration for $G$ and a vortex configuration for $\mathcal{F}_r$. The descendant mixed gauge/0-form vortex defect induces a 2-group symmetry background with

$$\int_{D_2} B_w^r = \alpha, \tag{62}$$

in its vicinity, and hence is a codimension-2 solitonic defect inducing a 2-group background. Correspondingly, a monopole operator $M$ living at the end of $V$ descends to what we call

a *mixed gauge/0-form monopole operator* in the gauged theory $\mathfrak{T}/\mathcal{G}$, which is a codimension-three solitonic defect living at the end of mixed gauge/0-form vortex defect inducing a 2-group background.

**Example.** Let us now provide a worked example of these constructions. Suppose that we gauge a symmetry $\mathcal{G} = U(1)$ of the original theory $\mathfrak{T}$ in such a way that the structure group of the gauged theory $\mathfrak{T}/\mathcal{G}$ is

$$\mathcal{S} = \frac{U(1) \times SU(2)}{\mathbb{Z}_{2q}}, \tag{63}$$

where the denominator $\mathcal{E}_r$ is generated by the central element $(e^{i\pi/q}, e^{i\pi}\mathbf{1}_2)$ for some positive integer $q \in \mathbb{Z}$ and therefore $\mathcal{Z}_g = \mathbb{Z}_{2q}$ and $\mathcal{Z}_r = \mathbb{Z}_2$. The 0-form symmetry is $\mathcal{F}_r = SO(3)$ and the 1-form symmetry is $\Gamma_r^{(1)} = \mathbb{Z}_q$, which participate in a non-trivial 2-group if $q$ is even. A concrete example of this scenario for $q = 2$ is provided if we take $\mathfrak{T}/\mathcal{G}$ to a $U(1)$ gauge theory with two matter fields of gauge charge 2.

The vortex configurations may be labelled by a pair of integers $\phi = (\phi_g, \phi_r) \in \mathbb{Z} \times \mathbb{Z}_{\geq 0}$ specifying cocharacters of $G = U(1)/\mathbb{Z}_{2q}$ and $\mathcal{F}_r = SO(3)$, with obstruction classes

$$\alpha_\phi = \phi_g \bmod 2q, \qquad \alpha_r = \phi_r \bmod 2, \tag{64}$$

of lifting them to cocharacters of $\mathcal{G} = U(1)$ and $F_r = SU(2)$ respectively. This data is constrained such that the cocharacters agree modulo 2, or equivalently that the obstruction classes are constrained to satisfy

$$\alpha_\phi \bmod 2 = \alpha_r. \tag{65}$$

We have the following classification:

- The vortex configurations labelled by cocharacters $\phi = (\phi_g, 0)$ with only gauge components must have $\phi_g$ even. If $\phi_g$ is a multiple of $2q$, this descends to a dynamical $U(1)$ gauge vortex and is therefore trivial.

- Therefore, up to fusion with dynamical $U(1)$ gauge vortices, vortex configurations of the form $\phi = (\phi_g, 0)$ are labelled by $\phi_g = 2m$ with $m = 0, \ldots, q-1$. For $m \neq 0$, these are fractional gauge vortices. In each class $m$, we have a special vortex defect which is topological and is recognized as a Gukov-Witten operator generating the element $m$ in the 1-form symmetry group $\mathbb{Z}_q$.

- The mixed gauge/0-form vortices descend from vortex configurations $\phi = (\phi_g, \phi_r)$ with both $\phi_g$ and $\phi_r$ non-trivial.

### 4.1.2 Three dimensions

In dimension $d = 3$, this construction requires modification due to the existence of Chern-Simons interactions and the fact that gauge monopole local operators transform under an additional 0-form topological symmetry.

First, we note that in three dimensions the specification of a 0-form symmetry $\mathcal{F}$ of a theory $\mathfrak{T}$ requires specification of Chern-Simons contact terms. They become gauge Chern-Simon terms upon gauging a subgroup of the global symmetry and have an impact upon the construction of the structure group $\mathcal{S}$.

Second, in addition to the 0-form symmetries described above, there is an additional topological 0-form symmetry group under which gauge monopole operators are charged, which must appear in the structure group. The topological symmetry group is nominally

$$\widehat{\pi_1(\mathcal{G})}, \tag{66}$$

which is Pontrjagin dual to the Abelian group $\pi_1(\mathcal{G})$ which measures the topological type of a $\mathcal{G}$-bundle induced on a small $S^2$ surrounding a monopole operator. However, this may somewhat overlap with the residual 0-form symmetry discussed above. The gauge monopole operators charged under the topological symmetry pick up gauge and 0-form charges due to Chern-Simons interactions, further impacting the construction of the structure group.

**Extension of topological symmetry.** Before discussing the modification to the structure group in three dimensions, we first introduce an extension of the topological symmetry group in order incorporate the topological charges of fractional gauge monopoles residing at the end of line operators.

Specifically, in the gauged theory $\mathcal{T}/\mathcal{G}$, we have seen that there exist fractional mixed gauge/0-form vortices that end on monopole operators charged under $\mathcal{Z}_g$. Such monopole operators carry fractional topological charge under (or more generally form projective representations of) $\widehat{\pi_1(\mathcal{G})}$ but integral charges under (or more generally form genuine representations of) the extension $\widehat{\pi_1(G)}$. We will therefore work with the extension $\widehat{\pi_1(G)}$ in constructing the structure group. Since we are interested in continuous parts of $\widehat{\pi_1(\mathcal{G})}$ and $\widehat{\pi_1(G)}$, we will work with free non-torsional parts of $\pi_1(\mathcal{G})$ and $\pi_1(G)$.

In more detail, our starting point is the quotient $G = \mathcal{G}/\mathcal{Z}_g$, which is associated to the short exact sequence

$$0 \longrightarrow \mathcal{Z}_g \longrightarrow \mathcal{G} \longrightarrow G \longrightarrow 0 \,. \tag{67}$$

The associated long exact sequence in homotopy truncates to a short exact sequence

$$0 \longrightarrow \pi_1(\mathcal{G}) \longrightarrow \pi_1(G) \longrightarrow \mathcal{Z}_g \longrightarrow 0 \,, \tag{68}$$

where we have used that $\pi_1(\mathcal{Z}_g) = 0$, $\pi_0(\mathcal{Z}_g) = \mathcal{Z}_g$ for a discrete group and $\pi_0(\mathcal{G}) = 0$ since we have assumed that the gauge group is connected. Restricting the above short exact sequence to the free parts, we obtain the short exact sequence

$$0 \longrightarrow \text{Free}\,(\pi_1(\mathcal{G})) \longrightarrow \text{Free}\,(\pi_1(G)) \longrightarrow \mathcal{Z}_g \longrightarrow 0 \,. \tag{69}$$

The Pontryagin dual short exact sequence now supplies the required central extension of the topological symmetry

$$0 \longrightarrow \widehat{\mathcal{Z}_g} \longrightarrow F_t \longrightarrow \text{Conn}\left(\widehat{\pi_1(\mathcal{G})}\right) \longrightarrow 0 \,, \tag{70}$$

where

$$F_t := \text{Conn}\left(\widehat{\pi_1(G)}\right) \,, \tag{71}$$

where the connected part $\text{Conn}(H)$ of a group $H$ denotes the subgroup of $H$ obtained by restricting to elements of $H$ path connected to the identity element. The extension $F_t$ of the topological symmetry $\text{Conn}\left(\widehat{\pi_1(\mathcal{G})}\right)$ now acts faithfully on all genuine and non-genuine local operators. This central extension may be necessary even if the 1-form symmetry $\Gamma_r^{(1)}$ of the gauged theory $\mathcal{T}/\mathcal{G}$ is trivial, as even in such a situation $\mathcal{Z}_g$ may not be trivial.

**Structure group.** The structure group of the gauged theory $\mathcal{T}/\mathcal{G}$ must now be extended to incorporate the 0-form topological symmetry. The structure group takes the form

$$\mathcal{S} = \frac{\mathcal{G} \times F_{rt}}{\mathcal{E}_{rt}} \,, \tag{72}$$

where

$$F_{rt} = F_r \times F_t \,, \tag{73}$$

and the central subgroup $\mathcal{E}_{rt}$ in three-dimensions incorporates the topological charges of genuine and non-genuine local operators along with their charges under gauge and residual 0-form symmetries.

The discussion of symmetries, background fields, summing over gauge bundles and vortex configurations now follows as before with the appropriate replacements $F_r \to F_{rt}$ and $\mathcal{E}_r \to \mathcal{E}_{rt}$.

## 4.2 Gauging 0-form symmetry in the presence of a 2-group

Let us now assume that the 0-form symmetry $\mathcal{F}_{gr}$ of the original theory $\mathfrak{T}$ combines with the 1-form symmetry $\Gamma^{(1)}$ of $\mathfrak{T}$ to form a 2-group symmetry based on a short exact sequence

$$0 \to \Gamma^{(1)} \to \mathcal{E}_{gr} \to \mathcal{Z}_{gr} \to 0. \tag{74}$$

**Symmetries after gauging.** As before, after gauging $\mathcal{G} = F_g$, we obtain a new electric 1-form symmetry $\Gamma_g^{(1)}$ from the gauge group $\mathcal{G}$ which is described as

$$\Gamma_g^{(1)} = Z_{\mathcal{G}} \cap \mathcal{Z}_{gr}, \tag{75}$$

where $Z_{\mathcal{G}}$ is the center of the gauge group $\mathcal{G}$. Combining it with the 1-form symmetry $\Gamma^{(1)}$ descending from the theory before gauging, we obtain a 1-form symmetry $\Gamma_r^{(1)}$ of the gauged theory $\mathfrak{T}/\mathcal{G}$ given by

$$\Gamma_r^{(1)} = \pi_{gr}^{-1}\left(\Gamma_g^{(1)}\right), \tag{76}$$

where $\pi_{gr}$ is the projection map $\mathcal{E}_{gr} \to \mathcal{Z}_{gr}$ in the short exact sequence (74).

The 2-group symmetry of $\mathfrak{T}$ descends to a 2-group symmetry of the gauged theory $\mathfrak{T}/\mathcal{G}$ given by the short exact sequence

$$0 \to \Gamma_r^{(1)} \to \mathcal{E}_r \to \mathcal{Z}_r \to 0, \tag{77}$$

where $\mathcal{E}_r = \mathcal{E}_{gr}$ and $\mathcal{Z}_r$ is given by

$$\mathcal{Z}_r = \frac{\mathcal{Z}_{gr}}{\Gamma_g^{(1)}} = p_r\left(\mathcal{Z}_{gr}\right), \tag{78}$$

which is a subgroup of the center of $F_r$, where $p_r$ is the natural projection map from the center of $\mathcal{G} \times F_r$ to the center of $F_r$.

The 0-form symmetry group $\mathcal{F}_r$ associated to the 0-form symmetry $\mathfrak{f}_r$ of the theory after gauging is

$$\mathcal{F}_r = \frac{F_r}{\mathcal{Z}_r}. \tag{79}$$

**Example.** A concrete example in any spacetime dimension $d$ is provided by $\mathfrak{T}$ being a $U(1)$ gauge theory with two matter fields of charge 2, in which case $\mathcal{F} = SO(3)$, $F = SU(2)$ and $\Gamma^{(1)} = \mathbb{Z}_2$ form a non-trivial 2-group based on short exact sequence

$$0 \to \mathbb{Z}_2 \to \mathbb{Z}_4 \to \mathbb{Z}_2 \to 0. \tag{80}$$

Let us now construct $\mathfrak{T}/\mathcal{G}$ by gauging $\mathcal{G} = F = SU(2)$. We now have

$$\Gamma_g^{(1)} = Z_{\mathcal{G}} \cap \mathcal{Z}_{gr} = \mathbb{Z}_2 \cap \mathbb{Z}_2 = \mathbb{Z}_2, \tag{81}$$

and hence

$$\Gamma_r^{(1)} = \mathbb{Z}_4. \tag{82}$$

Indeed, the electric 1-form symmetry of a theory, with gauge group $U(1) \times SU(2)$, and a matter field of charge 2 under $U(1)$ and transforming in fundamental of $SU(2)$, is $\mathbb{Z}_4$. The remaining 0-form symmetry group $\mathcal{F}_r$ is trivial.

**Allowed bundles for the gauged theory.** Let us describe the coupling of the theory obtained after gauging to background and gauge fields. Suppose we have turned on a bundle for the 0-form symmetry group $\mathcal{F}_r$ which is accompanied with a $\mathcal{Z}_r$ valued 2-cocycle $w_2^r$ capturing the obstruction of lifting the $\mathcal{F}_r$ bundle to an $F_r$ bundle. Let the background field for the 1-form symmetry be the $\Gamma_r^{(1)}$ valued 2-cochain $B_2^r$. The background fields $B_2^r$ and $w_2^r$ combine to form a background field for the 2-group symmetry, which is given by an $\mathcal{E}_r$ valued 2-cocycle $B_w^r$.

The gauge theory now sums over bundles for the group $\mathcal{G}/\Gamma'^{(1)}$ with

$$\Gamma'^{(1)} := p_g\left(\mathcal{Z}_{gr}\right), \tag{83}$$

where $p_g$ is the natural projection map from the center of $\mathcal{G} \times F_r$ to the center of $\mathcal{G}$. The bundles being summed over are constrained to have the obstruction class

$$w_2' = p_g \circ \pi_{gr}\left(B_w^r\right), \tag{84}$$

of lifting $\mathcal{G}/\Gamma'^{(1)}$ bundles to $\mathcal{G}$ bundles. A $\mathcal{G}/\Gamma'^{(1)}$ bundle having obstruction class $w_2'$ combined with the 0-form background bundle for $\mathcal{F}_r$ gives rise to a bundle for the structure group $\mathcal{S}$ with $\mathcal{Z}_{gr}$ valued obstruction class

$$w_2^{\mathcal{S}} = \pi_{gr}\left(B_w^r\right), \tag{85}$$

for lifting the $\mathcal{S}$ bundle to a $\mathcal{G} \times F_r$ bundle. The gauge theory sums over such bundles for the structure group $\mathcal{S}$.

**Fate of vortices and monopoles under gauging.** Consider a codimension-two defect $V$ inducing a vortex configuration $V_\phi(\mathcal{F}_{gr})$. Recall that there is associated an element $\alpha_V \in \mathcal{E}_{gr}$ to the defect. Before gauging, $\alpha_V$ describes the 2-group background field $B_w^{gr}$ induced by the defect. Moreover, the 0-form background field $w_2^{gr}$ induced by $V$ is $\pi_{gr}(\alpha_V) = \alpha_\phi \in \mathcal{Z}_{gr}$, which can also be identified with the obstruction of lifting the vortex configuration $V_\phi(\mathcal{F}_{gr})$ to a vortex configuration for the group $F_g \times F_r$. After gauging, since $\mathcal{E}_{gr} = \mathcal{E}_r$, $\alpha_V$ describes also the 2-group background field $B_w^r$ induced by $V$ in the gauged theory $\mathfrak{T}/\mathcal{G}$. The 0-form background field $w_2^r$ induced by $V$ in $\mathfrak{T}/\mathcal{G}$ is $\pi_r(\alpha_V) \in \mathcal{Z}_r$. In fact, the vortex configuration for $\mathcal{F}_r$ induced by $V$ is $V_{\phi_r}(\mathcal{F}_r)$, with

$$\phi_r = \pi_{\mathcal{S}} \circ \phi : \ U(1) \to \mathcal{F}_r, \tag{86}$$

where

$$\pi_{\mathcal{S}} : \ \mathcal{F}_{gr} \to \mathcal{F}_r, \tag{87}$$

is the projection map from $\mathcal{F}_{gr}$ to $\mathcal{F}_r$ obtained by forgetting $F_g$. Monopoles are transformed in a similar fashion as they live at the ends of vortices.

## 4.3 Gauging 1-form symmetry

**Mixed anomaly from 2-group.** Let us now consider gauging the 1-form symmetry $\Gamma^{(1)}$ of a theory $\mathfrak{T}$ with 2-group symmetry $\left(\mathcal{F}, \Gamma^{(1)}, \Theta\right)$. We denote the resulting gauged theory by $\mathfrak{T}/\Gamma^{(1)}$. The gauged theory has a dual $(d-3)$-form symmetry given by the Pontryagin dual group

$$\Gamma^{(d-3)} = \widehat{\Gamma^{(1)}} = \text{Hom}\left(\Gamma^{(1)}, U(1)\right), \tag{88}$$

which has a mixed 't Hooft anomaly with the 0-form symmetry $\mathcal{F}$ whose associated anomaly theory takes the form

$$\mathcal{A}_{d+1} = \exp\left(2\pi i \int_M \Theta \cup B_{d-2}\right), \tag{89}$$

where $B_{d-2}$ is the background for the $(d-3)$-form symmetry and the cup product is taken using the natural pairing $\Gamma^{(1)} \times \widehat{\Gamma^{(1)}} \to \mathbb{R}/\mathbb{Z}$. We can understand this 't Hooft anomaly in terms of operators, or in terms of backgrounds.

In terms of backgrounds [3], gauging the 1-form symmetry is implemented by adding to the action a term of the form

$$\int_M B_2 \cup B_{d-2}, \tag{90}$$

and then summing over $B_2$ backgrounds. A gauge transformation $B_{d-2} \to B_{d-2} + \delta\lambda_{d-3}$ changes correlation functions by a phase

$$\exp\left( 2\pi i \int_M \Theta \cup \lambda_{d-3} \right), \tag{91}$$

implying the mixed 't Hooft anomaly (89).

**Derivation of anomaly from solitonic defects.** In terms of operators, we use the interpretation of the 2-group symmetry in terms of solitonic Postnikov defects. Recall from section 2.3 that the presence of 2-group symmetry is equivalent to the statement that a codimension-3 solitonic Postnikov defect lies in twisted sector for 1-form symmetry. On the other hand, it is well-known that the twisted sector operators are charged under the dual $(d-3)$-form symmetry in the gauge theory.

Combining the two statements, we learn that in the gauged theory $\mathfrak{T}/\Gamma^{(1)}$, the codimension-3 solitonic Postnikov defects are charged under the $(d-3)$-form symmetry, which implies the anomaly (89).

**Fractional to non-fractional gauge monopoles.** Let us now consider an application of this construction to the gauged theory $\mathfrak{T}/\mathcal{G}$ considered in section 4.1. Recall that this theory inherited an electric 1-form symmetry $\Gamma_r^{(1)}$ potentially forming part of a 2-group with the residual 0-form symmetry $\mathcal{F}_r$, or after including the additional topological symmetry in three dimensions, the 0-form symmetry $\mathcal{F}_{rt}$.

If we further gauge the 1-form symmetry $\Gamma_r^{(1)}$ in theory $\mathfrak{T}/\mathcal{G}$, we obtain a theory that we call $\mathfrak{T}/\mathcal{G}'$ whose gauge group is

$$\mathcal{G}' = \mathcal{G}/\Gamma_r^{(1)}. \tag{92}$$

Consider a fractional gauge vortex defect $V$ in theory $\mathfrak{T}/\mathcal{G}$. Recall that $V$ induces a gauge vortex configuration associated to a co-character $\phi$ for the group $\mathcal{G}'$. In the theory $\mathfrak{T}/\mathcal{G}'$, the defect $V$ descends to a non-vortex defect. In particular, the Gukov-Witten operators implementing $\Gamma_r^{(1)}$ 1-form symmetry are examples of fractional gauge vortex defects in the theory $\mathfrak{T}/\mathcal{G}$, and upon gauging, they descend to the identity codimension-two defect in the theory $\mathfrak{T}/\mathcal{G}'$, which is of course a non-vortex defect.

Correspondingly, a fractional gauge monopole operator $M$ living at the end of fractional gauge vortex defect $V$ in theory $\mathfrak{T}/\mathcal{G}$ descends to a non-fractional gauge monopole operator for the gauge group $\mathcal{G}'$ in the theory $\mathfrak{T}/\mathcal{G}'$. In particular, a twisted sector operator living at the end of a Gukov-Witten operator for $\Gamma_r^{(1)}$ 1-form symmetry in the theory $\mathfrak{T}/\mathcal{G}$ becomes a standard genuine monopole operator for the gauge group $\mathcal{G}'$ in the theory $\mathfrak{T}/\mathcal{G}'$. See figure 8.

## 5 't Hooft anomalies in 3d from solitonic defects

In this section we discuss the structure and physical implications of some 't Hooft anomalies between 0-form, 1-form and 2-group symmetries in dimension $d = 3$ and how they can be cleanly formulated in terms of solitonic defects introduced in section 3.

$$M_\phi(\mathcal{G}') \quad\bullet\!\!\!\rule[0.5ex]{6em}{0.4pt}\!\!\!\bullet \quad \alpha_\phi \in \Gamma_r^{(1)} \quad \xrightarrow{\text{gauging } \Gamma_r^{(1)}} \quad M_\phi(\mathcal{G}') \quad \bullet$$

Figure 8: In the theory $\mathfrak{T}/\mathcal{G}$, $M_\phi(\mathcal{G}')$ is a fractional gauge monopole operator inducing a monopole configuration associated to a co-character $\phi$ for the group $\mathcal{G}'$ which has obstruction $\alpha_\phi \in \Gamma_r^{(1)}$ of being lifted to a co-character of the gauge group $\mathcal{G}$. Also assume $M_\phi(\mathcal{G}')$ to be twisted sector operator lying at the end of the Gukov-Witten operator implementing the 1-form symmetry $\alpha_\phi$. Gauging the 1-form symmetry makes the Gukov-Witten operator invisible. Correspondingly, the gauging procedure converts $M_\phi(\mathcal{G}')$ into a standard genuine gauge monopole operator in the theory $\mathfrak{T}/\mathcal{G}'$ inducing a monopole configuration for the new gauge group $\mathcal{G}'$ associated to the co-character $\phi$.

## 5.1 Anomaly for 1-form symmetry

In this subsection, we discuss anomalies for 1-form symmetries. If the 1-form symmetry participates in a 2-group symmetry, then the expressions for anomalies discussed here are only valid if the background for 0-form symmetry participating in the 2-group symmetry is not turned on. If the 0-form symmetry background is turned on, then the 1-form anomaly lifts to a 2-group anomaly. 2-group anomalies are discussed later in this section.

**Definition.** In such a situation, the 1-form symmetry $\Gamma^{(1)}$ has backgrounds specified by the choice of a 2-cocycle $B_2$ on the spacetime manifold $M_3$. For each such choice of background $B_2$, the theory assigns a well-defined partition function $Z(B_2)$. This is the correlation function of the theory on $M_3$ with a network of topological codimension-two defects generating the 1-form symmetry inserted on $M_3$, with the 1-cycle described by the network being Poincaré dual to $B_2$.

An anomaly for the 1-form symmetry arises if the correlation function $Z(B_2)$ is *not* a well-defined function of the cohomology class $[B_2]$ of $B_2$. In other words, we have

$$Z(B_2 + \delta\lambda_1) = \phi(\lambda_1, B_2) \times Z(B_2), \tag{93}$$

where $\phi(\lambda_1, B_2) \in U(1)$ is a phase factor depending on $\lambda_1$ and $B_2$.

To the anomaly, we can associate an *anomaly theory*, which is a 4d SPT phase protected by 1-form symmetry $\Gamma^{(1)}$. The partition function $\mathcal{A}_4(B_2)$ of the SPT phase on a compact 4-manifold $M_4$ with a background $B_2$ is well-defined as a function of $[B_2]$. However, on a 4-manifold $M_4$ with boundary $\partial M_4 = M_3$, we have

$$\mathcal{A}_4(B_2 + \delta\lambda_1) = \phi^{-1}(\lambda_1, B_2) \times \mathcal{A}_4(B_2). \tag{94}$$

Consequently, regarding the 3d theory as a boundary condition of the 4d SPT phase restores invariance under background gauge transformation. That is, the combined correlation function

$$\widetilde{Z}(B_2) = Z(B_2)\mathcal{A}_4(B_2), \tag{95}$$

satisfies

$$\widetilde{Z}(B_2 + \delta\lambda_1) = \widetilde{Z}(B_2). \tag{96}$$

**Type of anomaly being studied.** In this paper, we study anomalies in 3d of the form

$$\mathcal{A}_4 = \exp\left(2\pi i \int \mathcal{P}_\sigma(B_2)\right), \tag{97}$$

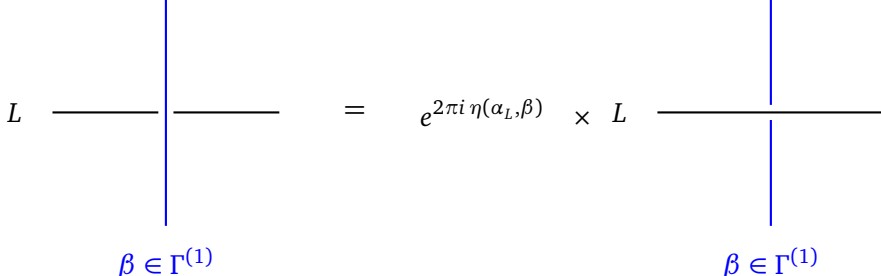

Figure 9: Moving a topological line defect implementing 1-form symmetry $\beta \in \Gamma^{(1)}$ across a (not necessarily topological) line defect $L$ inducing a 1-form symmetry background $\alpha_L \in \Gamma^{(1)}$ generates a phase $\exp\big(2\pi i\, \eta\,(\alpha_L,\beta)\big)$.

where

$$\mathcal{P}_\sigma(B_2) : H^2\big(M_3, \Gamma^{(1)}\big) \to H^4(M_3, \mathbb{R}/\mathbb{Z}), \tag{98}$$

is the Pontryagin square operation associated to a quadratic function

$$\sigma : \ \Gamma^{(1)} \to \mathbb{R}/\mathbb{Z}, \tag{99}$$

with the associated bilinear form being

$$\eta : \ \Gamma^{(1)} \times \Gamma^{(1)} \to \mathbb{R}/\mathbb{Z}. \tag{100}$$

In this paper we will only determine $\eta$ but not $\sigma$, which is sufficient to specify the anomaly on spin manifolds, for which the expression (97) does not depend on the different choices of $\sigma$ associated to a fixed $\eta$.

**Anomaly from solitonic line defects inducing 1-form backgrounds.** The above anomaly can be understood as stating that line defects that induce 1-form symmetry backgrounds around them are themselves charged under 1-form symmetry. Consider a line defect $L$ that induces a 1-form symmetry background forcing the background field $B_2$ for the 1-form symmetry to satisfy

$$\int_{D_2} B_2 = \alpha_L \in \Gamma^{(1)}, \tag{101}$$

where $D_2$ is a small disk intersecting the locus of $L$ at a single point. As a topological line defect generating a 1-form symmetry $\beta \in \Gamma^{(1)}$ is moved across the locus of $L$, the correlation function jumps by the phase factor

$$\exp\big(2\pi i\, \eta(\alpha_L,\beta)\big), \tag{102}$$

as shown in figure 9.

Particular choices for line defects that induce (101) around them are provided by the topological line defects generating the 1-form symmetry $\Gamma^{(1)}$. Thus, according to the above discussion, picking two such topological line defects corresponding to elements $\alpha, \beta \in \Gamma^{(1)}$ and moving them across each other makes the correlation function jump by the phase

$$\exp\big(2\pi i\, \eta(\alpha,\beta)\big), \tag{103}$$

as shown in figure 10. This recovers a standard description of the 't Hooft anomaly of a 1-form symmetry in three dimensions as the braiding phase between the topological line defects generating the symmetry.

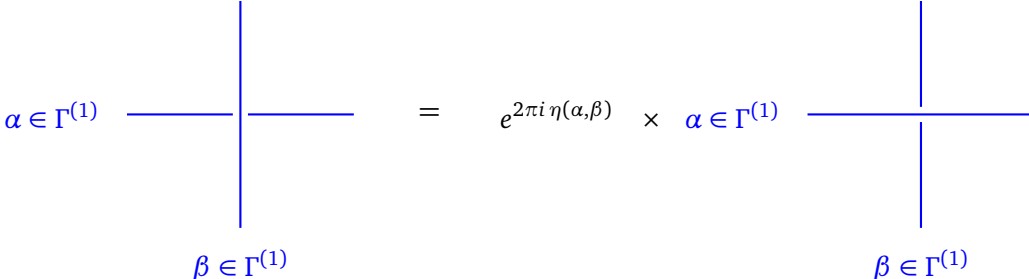

Figure 10: Moving two topological line defects implementing 1-form symmetries $\alpha, \beta \in \Gamma^{(1)}$ across each other generates a phase $\exp\left(2\pi i \, \eta(\alpha, \beta)\right)$.

The information of the bilinear form $\eta$ is equivalent to a homomorphism

$$\gamma : \; \Gamma^{(1)} \to \widehat{\Gamma^{(1)}}. \tag{104}$$

This homomorphism encodes the following physical information. Consider a line defect $L$ inducing a 1-form symmetry background associated to $\alpha \in \Gamma^{(1)}$. Then this line defect lies in the equivalence class

$$\gamma(\alpha) \in \widehat{\Gamma}^{(1)}, \tag{105}$$

of genuine line defects and hence carries the charge $\gamma(\alpha)$ under the 1-form symmetry group $\Gamma^{(1)}$.

**Gauging 1-form symmetry.** Consider now gauging a subgroup $\Gamma^{(1)'}$ of 1-form symmetry $\Gamma^{(1)}$ in a theory with anomaly (97). This corresponds to making the topological line defects corresponding to elements $\alpha \in \Gamma^{(1)'}$ equivalent to the trivial line defect. To avoid *gauge anomalies*, we must make sure that these topological line defects are uncharged under themselves. To describe this requirement mathematically, let us introduce two short exact sequences associated to this gauging:

$$0 \to \Gamma^{(1)'} \to \Gamma^{(1)} \to \Gamma^{(1)}/\Gamma^{(1)'} \to 0, \tag{106}$$

and

$$0 \to \widehat{\Gamma^{(1)}/\Gamma^{(1)'}} \to \widehat{\Gamma^{(1)}} \to \widehat{\Gamma^{(1)}}' \to 0. \tag{107}$$

Gauge anomalies vanish if $\Gamma^{(1)'}$ is such that

$$\widehat{\pi} \circ \gamma \left( \Gamma^{(1)'} \right) = 0 \subset \widehat{\Gamma^{(1)}}', \tag{108}$$

where

$$\widehat{\pi} : \; \widehat{\Gamma^{(1)}} \to \widehat{\Gamma^{(1)}}', \tag{109}$$

is the projection map in (107).

Part of the rest of the $\Gamma^{(1)}/\Gamma^{(1)'}$ 1-form symmetry can suffer from an *ABJ anomaly* after gauging. Consider an element $\alpha \in \Gamma^{(1)}/\Gamma^{(1)'}$ and choose a lift $\widetilde{\alpha} \in \Gamma^{(1)}$. The element $\alpha$ suffers from an ABJ anomaly if $\widetilde{\alpha}$ is charged non-trivially under $\Gamma^{(1)'}$. In other words, the 1-form symmetry of the theory obtained after gauging, which does not suffer from ABJ anomaly, is

$$\Gamma_r^{(1)} := \ker\left( \widehat{\pi} \circ \gamma : \; \Gamma^{(1)}/\Gamma^{(1)'} \to \widehat{\Gamma^{(1)}}' \right) \subseteq \Gamma^{(1)}/\Gamma^{(1)'}, \tag{110}$$

where $\widehat{\pi} \circ \lambda$ is a well-defined map from $\Gamma^{(1)}/\Gamma^{(1)'}$ to $\widehat{\Gamma^{(1)}}'$ because of the condition (108).

We can provide an explicit expression for the ABJ anomaly as

$$\mathcal{A}_{\text{ABJ}} = \exp\left( 2\pi i \int b_2' \cup (\widehat{\pi} \circ \gamma)(B_2^m) \right), \tag{111}$$

where $b_2'$ is the dynamical gauge field for gauged 1-form symmetry $\Gamma^{(1)'}$ and $B_2^m$ is background field for $\Gamma^{(1)}/\Gamma^{(1)'}$. The above expression for the ABJ anomaly vanishes if $B_2^m$ is a background field for $\Gamma^{(1)}/\Gamma^{(1)'}$ which is valued entirely in the subgroup $\Gamma_r^{(1)}$ of $\Gamma^{(1)}/\Gamma^{(1)'}$, or in more concrete words, if $B_2 \in H^2\left(M_3, \Gamma^{(1)}/\Gamma^{(1)'}\right)$ lies in the image of the homomorphism

$$H^2\left(M_3, \Gamma_r^{(1)}\right) \to H^2\left(M_3, \Gamma^{(1)}/\Gamma^{(1)'}\right), \tag{112}$$

descending from the injective map $\Gamma_r^{(1)} \to \Gamma^{(1)}/\Gamma^{(1)'}$ associated to the fact that $\Gamma_r^{(1)} \subseteq \Gamma^{(1)}/\Gamma^{(1)'}$. In other words, $\Gamma_r^{(1)}$ does not suffer from ABJ anomaly.

The anomaly (97) also descends to a *'t Hooft anomaly*

$$\mathcal{A}_4 = \exp\left( 2\pi i \int \mathcal{P}_{\sigma_r}(B_2^r) \right), \tag{113}$$

for the $\Gamma_r^{(1)}$ 1-form symmetry. Here $B_2^r$ is the background for the $\Gamma_r^{(1)}$ 1-form symmetry, and $\sigma_r$ is obtained as a quadratic refinement of the bilinear form $\eta_r : \Gamma_r^{(1)} \times \Gamma_r^{(1)} \to \mathbb{R}/\mathbb{Z}$ obtained from the homomorphism

$$\gamma_r : \Gamma_r^{(1)} \to \widehat{\Gamma}_r^{(1)}, \tag{114}$$

which takes the form

$$\gamma_r(\alpha) = p \circ \gamma(\widetilde{\alpha}), \tag{115}$$

where $\widetilde{\alpha} \in \Gamma^{(1)}$ is a lift of $\alpha \in \Gamma_r^{(1)}$, the image $\gamma(\widetilde{\alpha}) \in \widehat{\Gamma^{(1)}/\Gamma^{(1)'}}$, and $p$ is the projection map in the short exact sequence

$$0 \to \gamma(\Gamma^{(1)'}) \to \widehat{\Gamma^{(1)}/\Gamma^{(1)'}} \to \widehat{\Gamma^{(1)}}_r \to 0. \tag{116}$$

**Semi-simple $\Gamma^{(1)}$.** An important special case of the discussion presented above in this subsection is as follows. Suppose that the 1-form symmetry group can be decomposed as

$$\Gamma^{(1)} = {\Gamma^{(1)}}_1 \times {\Gamma^{(1)}}_2, \tag{117}$$

with the non-trivial part of the homomorphism $\gamma : \Gamma^{(1)} \to \widehat{\Gamma^{(1)}}$ decomposing into two homomorphisms

$$\gamma_{12} : {\Gamma^{(1)}}_1 \to \widehat{\Gamma^{(1)}}_2, \tag{118}$$

and

$$\gamma_{21} : {\Gamma^{(1)}}_2 \to \widehat{\Gamma^{(1)}}_1, \tag{119}$$

such that $\gamma_{21}$ and $\gamma_{12}$ are Pontryagin duals of each other. Then, we can write the (97) as

$$\mathcal{A}_4 = \exp\left( 2\pi i \int \gamma_{12}(B_{2,1}) \cup B_{2,2} \right), \tag{120}$$

where $B_{2,i}$ denotes the background field for ${\Gamma^{(1)}}_i$ 1-form symmetry. The anomaly in this special case is a *mixed 't Hooft anomaly* between the 1-form symmetries ${\Gamma^{(1)}}_1$ and ${\Gamma^{(1)}}_2$.

## 5.2 Mixed anomaly between 1-form and 0-form symmetries

In this subsection, we consider anomalies between a 1-form symmetry $\Gamma^{(1)}$ and a continuous 0-form symmetry group $\mathcal{F}$, such that $\Gamma^{(1)}$ and $\mathcal{F}$ do not mix together to form a 2-group. Moreover, we assume that $\mathcal{F}$ does not participate in any 2-group symmetry, but $\Gamma^{(1)}$ can. If $\Gamma^{(1)}$ does participate in a 2-group symmetry, then the backgrounds for the 0-form symmetry participating in the 2-group are kept off. We focus on anomalies whose dependence on the 0-form symmetry background is only through the degree two characteristic class $w_2 \in H^2(B\mathcal{F}, \mathcal{Z})$ capturing the obstruction of lifting the background 0-form symmetry $\mathcal{F}$ bundle to an $F$ bundle, where $F$ is a central extension of $\mathcal{F}$ under which all genuine and non-genuine local operators form allowed representations, and $\mathcal{Z}$ relates $\mathcal{F}$ and $F$ via $\mathcal{F} = F/\mathcal{Z}$.

**Type of anomaly being studied**   We study anomalies of the form

$$\mathcal{A}_4 = \exp\left( 2\pi i \int \gamma(B_2) \cup w_2 \right), \tag{121}$$

where

$$\gamma: \ \Gamma^{(1)} \to \widehat{\mathcal{Z}}, \tag{122}$$

is a homomorphism, and the cup product $\int \gamma(B_2) \cup c \in \mathbb{R}/\mathbb{Z}$ is evaluated by using the pairing $\widehat{\mathcal{Z}} \times \mathcal{Z} \to \mathbb{R}/\mathbb{Z}$.

The presence of this anomaly means that a partition function of the 3d theory jumps as

$$Z(B_2 + \delta\lambda_1, w_2) = \exp\left( -2\pi i \int \gamma(\lambda_1) \cup w_2 \right) \times Z(B_2, w_2), \tag{123}$$

and

$$Z(B_2, w_2 + \delta\lambda_1') = \exp\left( -2\pi i \int \gamma(B_2) \cup \lambda_1' \right) \times Z(B_2, w_2). \tag{124}$$

**Pictorial representation of the anomaly.**   This anomaly admits a nice pictorial representation. For this pictorial representation, we use the 1-cycle Poincaré dual to the $\mathcal{Z}$ valued 2-cocycle $w_2$ on the spacetime manifold $M_3$. This 1-cycle can be represented as a network of lines, with each line segment in the network carrying an element of $\mathcal{Z}$. Consider one such line segment carrying the element $\alpha \in \mathcal{Z}$, and move it across a topological line operator associated to an element $\beta \in \Gamma^{(1)}$. Then the correlation function jumps by the phase

$$\exp\left( 2\pi i \langle \gamma(\beta), \alpha \rangle \right), \tag{125}$$

where

$$\langle \cdot, \cdot \rangle: \ \widehat{\mathcal{Z}} \times \mathcal{Z} \to \mathbb{R}/\mathbb{Z}, \tag{126}$$

is the canonical pairing. See figure 11.

**Anomaly from solitonic local operators inducing 1-form backgrounds.**   The above mixed anomaly can be understood in terms of 0-form charges of solitonic local operators inducing 1-form backgrounds. Consider such a local operator $O$ such that on a small sphere $S^2$ surrounding it, we have an induce 1-form background

$$\oint_{S^2} B_2 = \alpha \in \Gamma^{(1)}, \tag{127}$$

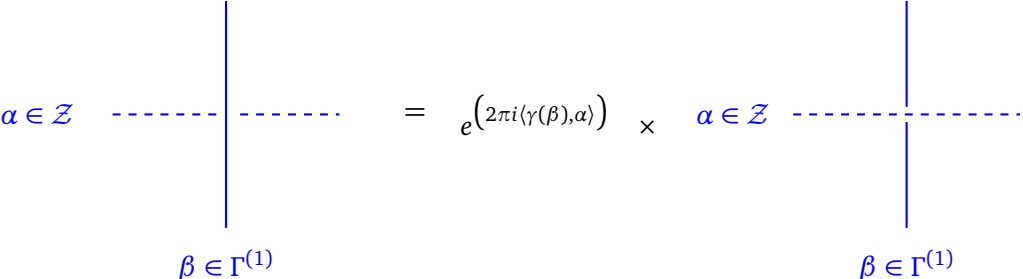

Figure 11: Moving a piece (shown dashed because it does not correspond to a line defect) carrying $\alpha \in \mathcal{Z}$ in the 1-cycle Poincaré dual to $w_2$ across a topological line defect associated to 1-form symmetry $\beta \in \Gamma^{(1)}$ generates a phase $\exp(2\pi i \langle \gamma(\beta), \alpha \rangle)$.

$$\alpha \in \Gamma^{(1)} \quad\underset{O}{\rule{2cm}{0.4pt}\bullet} \text{-}\text{-}\text{-}\text{-}\text{-}\text{-}\text{-}\text{-}\text{-}\text{-} \quad \gamma(\alpha) \in \widehat{\mathcal{Z}}$$

Figure 12: A twisted sector local operator $O$ (shown in red) associated to a topological line operator implementing the 1-form symmetry $\alpha \in \Gamma^{(1)}$ (shown in blue) must carry a representation $R$ of $F$ whose charge under $\mathcal{Z}$ is given by the element $\gamma(\alpha) \in \widehat{\mathcal{Z}}$.

Such a local operator lives at the end of a solitonic line defect $L$ which induces the same 1-form background on a small disk $D_2$ intersecting its locus at a single point.

The above mixed anomaly is the statement that the solitonic local operator $O$ transforms in a representation $R$ of $F$ whose charge under $\mathcal{Z}$ is given by the element

$$\gamma(\alpha) \in \widehat{\mathcal{Z}} . \tag{128}$$

If the anomaly, or in other words the homomorphism $\gamma$, is non-trivial, then such solitonic local operators transform in projective representations, rather than genuine representations, of the 0-form symmetry group $\mathcal{F}$.

Equivalently, the solitonic line defect $L$ lies in the equivalence class

$$\gamma(\alpha) \in \widehat{\mathcal{Z}} \subseteq \widehat{\mathcal{E}} , \tag{129}$$

in the group $\widehat{\mathcal{E}}$ of equivalence classes of line defects plus background Wilson lines defined in section 2.3.

Special examples of such solitonic local operators are provided by twisted sector operators for the 1-form symmetry $\Gamma^{(1)}$, for which the effect of anomaly is described in figure 12.

**Anomaly from vortex line defects for 0-form symmetry.** Consider a vortex defect (which is a line defect in 3d) $V$ inducing a vortex configuration for the 0-form symmetry group specified by a co-character

$$\phi : \ U(1) \to \mathcal{F} , \tag{130}$$

with obstruction $\alpha_\phi \in \mathcal{Z}$ of lifting the vortex configuration to a vortex configuration for the group $F$. The above anomaly can be reinterpreted as stating that $V$ lies in the equivalence class

$$\widehat{\gamma}(\alpha_\phi) \in \widehat{\Gamma^{(1)}} , \tag{131}$$

of line defects, where

$$\widehat{\gamma} : \ \mathcal{Z} \to \widehat{\Gamma^{(1)}} , \tag{132}$$

is the homomorphism Pontryagin dual to the homomorphism $\gamma$. In other words, $V$ carries the charge $\widehat{\gamma}(\alpha_\phi) \in \widehat{\Gamma^{(1)}}$ under the 1-form symmetry $\Gamma^{(1)}$.

$$\alpha \in \Gamma^{(1)'} \quad\underline{\hspace{1.5cm}}\cdots\cdots\cdots\gamma(\alpha)\in \widehat{\mathcal{Z}} \quad \xrightarrow{\text{gauging } \Gamma^{(1)'}} \quad \cdots\cdots\cdots\gamma(\alpha)\in\widehat{\mathcal{Z}}$$
$$O \qquad\qquad\qquad\qquad\qquad\qquad O$$

Figure 13: Gauging $\Gamma^{(1)'}$ makes the topological operators associated to elements of $\Gamma^{(1)'}$ invisible. As a consequence the twisted sector operators for $\Gamma^{(1)'}$ become genuine local operators of the theory obtained after gauging. These new genuine operators might carry representations of $\mathfrak{f}$ that are not allowed representations of $\mathcal{F}$. As a consequence, the 0-form symmetry group of the theory obtained after gauging $\Gamma^{(1)'}$ is generally larger that $\mathcal{F}$.

**Gauging 1-form symmetry.** Consider gauging a subgroup $\Gamma^{(1)'}$ of the $\Gamma^{(1)}$ 1-form symmetry. We assume that there is no anomaly for the 1-form symmetry other than (121). Gauging means that we regard topological line defects carrying elements of $\Gamma^{(1)'}$ as trivial line defects. This implies that the background Wilson lines valued in

$$\gamma\left(\Gamma^{(1)'}\right) \subseteq \widehat{\mathcal{Z}}, \tag{133}$$

become equivalent to trivial lines. That is, the theory obtained after gauging admits genuine local operators transforming in representations of $F$ whose charges under $\mathcal{Z}$ are valued in $\gamma\left(\Gamma^{(1)'}\right)$. In fact, these local operators are furnished by the twisted sector operators for $\Gamma^{(1)'}$. See figure 13.

In other words, one of the consequences of the existence of the anomaly (121) is that the 0-form symmetry group of the theory obtained after gauging is enhanced from $\mathcal{F}$ to $\mathcal{F}_r$ which in general is a central extension of $\mathcal{F}$ such that

$$\mathcal{F} = \mathcal{F}_r \big/ \widehat{\gamma}\left(\Gamma^{(1)'}\right), \tag{134}$$

and

$$\mathcal{F}_r = F/\mathcal{Z}_r, \tag{135}$$

where $\mathcal{Z}_r \subseteq \mathcal{Z}$ sits in a short exact sequence

$$0 \to \mathcal{Z}_r \to \mathcal{Z} \to \widehat{\gamma}\left(\Gamma^{(1)'}\right) \to 0, \tag{136}$$

Pontryagin dual to the short exact sequence

$$0 \to \gamma\left(\Gamma^{(1)'}\right) \to \widehat{\mathcal{Z}} \to \widehat{\mathcal{Z}}_r \to 0, \tag{137}$$

descending from (133).

The theory after gauging has a

$$\Gamma_r^{(1)} := \Gamma^{(1)}/\Gamma^{(1)'}, \tag{138}$$

1-form symmetry. The mixed anomaly (121) descends to a mixed anomaly of the form

$$\mathcal{A}_4 = \exp\left(2\pi i \int \gamma_r\left(B_2^r\right) \cup w_2^r\right), \tag{139}$$

between the $\Gamma_r^{(1)}$ 1-form symmetry and the $\mathcal{F}_r$ 0-form symmetry, where $B_2^r$ is the background for the $\Gamma_r^{(1)}$ 1-form symmetry and $w_2^r$ is the $\mathcal{Z}_r$ valued characteristic class capturing the obstruction to lifting $\mathcal{F}_r$ bundles to $F$ bundles.

$$\gamma_r : \Gamma_r^{(1)} \to \widehat{\mathcal{Z}}_r, \tag{140}$$

is a homomorphism which can be specified as

$$\gamma_r(\alpha) = \widehat{\pi} \circ \gamma(\widetilde{\alpha}), \tag{141}$$

where $\widetilde{\alpha} \in \Gamma^{(1)}$ is a lift of the element $\alpha \in \Gamma^{(1)}{}_r$ and

$$\widehat{\pi}: \ \widehat{\mathcal{Z}} \to \widehat{\mathcal{Z}}_r, \tag{142}$$

is the projection map appearing in the short exact sequence (137).

**Gauging 0-form symmetry.**     Consider gauging the 0-form symmetry $\mathfrak{f}$ via an $F$ gauge group. According to the discussion of section 4, the resulting theory then has a 1-form symmetry

$$\Gamma_r^{(1)} = \Gamma^{(1)} \times \mathcal{Z}. \tag{143}$$

The anomaly (121) becomes a mixed 't Hooft anomaly

$$\mathcal{A}_4 = \exp\left( 2\pi i \int \gamma\left( B_2^{\Gamma^{(1)}} \right) \cup B_2^{\mathcal{Z}} \right), \tag{144}$$

between the $\Gamma^{(1)}$ and $\mathcal{Z}$ subfactors of the $\Gamma_r^{(1)}$ 1-form symmetry, where $B_2^{\Gamma^{(1)}}$ and $B_2^{\mathcal{Z}}$ are background fields for the $\Gamma^{(1)}$ and $\mathcal{Z}$ valued 1-form symmetries respectively. This 1-form symmetry anomaly takes the form of the special case (120) of the 1-form symmetry anomalies discussed in section 5.1.

**Anomaly (144) in terms of solitonic defects inducing 1-form symmetry.**     The anomaly (144) can also be obtained using the properties of the 1-form solitonic defects of the theory before gauging. Consider a solitonic local operator $O$ inducing a 1-form symmetry background $\alpha \in \Gamma^{(1)}$. $O$ comes attached to a background Wilson line transforming in a representation $R$ of $F$ having charge $\gamma(\alpha)$ under $\mathcal{Z}$, and it is also attached to a solitonic line defect $L$ inducing 1-form background $\alpha$.

After gauging $F$, the background Wilson line becomes a gauge Wilson line having charge $\gamma(\alpha)$ under the 1-form symmetry $\mathcal{Z}$, and $O$ becomes a local operator interpolating between this gauge Wilson line and the solitonic line $L$. This implies that the solitonic line defect $L$ for 1-form symmetry $\Gamma^{(1)}$ has charge $\gamma(\alpha) \in \widehat{\mathcal{Z}}$ under the other 1-form symmetry $\mathcal{Z}$, leading to the mixed 't Hooft anomaly (144) between the two 1-form symmetries $\Gamma^{(1)}$ and $\mathcal{Z}$.

A special case is provided by $L$ being the topological line defect implementing the 1-form symmetry $\alpha \in \Gamma^{(1)}$ and $O$ being a twisted sector operator for this 1-form symmetry. See figure 14.

**Anomaly (144) in terms of fractional gauge vortex defects.**     Before the $F$ gauging, the anomaly (121) means that a vortex line defect $V$ carrying $\alpha \in \mathcal{Z}$ carries a charge $\widehat{\gamma}(\alpha) \in \widehat{\Gamma^{(1)}}$ under the 1-form symmetry. After the $F$ gauging, $V$ descends to a gauge vortex defect which is fractional if $\alpha \neq 0$. In particular, it is a solitonic line defect inducing 1-form background $\alpha \in \mathcal{Z}$. Since this solitonic line defect for $\mathcal{Z}$ 1-form symmetry is charged under $\Gamma^{(1)}$ 1-form symmetry, we obtain the mixed anomaly (144) between the two 1-form symmetries.

## 5.3   Anomaly for 0-form symmetry

In this subsection, we consider anomalies for a continuous 0-form symmetry group $\mathcal{F}$, which are specified purely in terms of the obstruction class $w_2$ for lifting $\mathcal{F}$ bundles to $F$ bundles. We again assume that $\mathcal{F}$ does not participate in any 2-group symmetry.

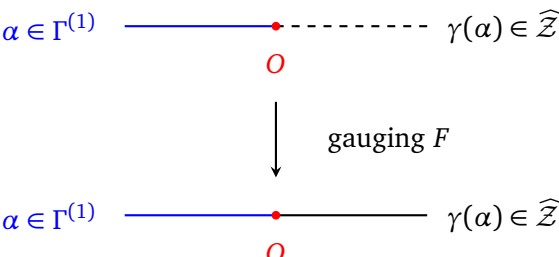

Figure 14: As we discussed above, the anomaly (121) implies that a twisted sector local operator $O$ for $\alpha \in \Gamma^{(1)}$ comes attached to a background Wilson line transforming in a representation of $F$ having charge $\gamma(\alpha)$ under $\mathcal{Z}$. After gauging $F$, the background Wilson line becomes a gauge Wilson line having charge $\gamma(\alpha)$ under the 1-form symmetry $\mathcal{Z}$. This implies that the topological line operator implementing the 1-form symmetry $\alpha \in \Gamma^{(1)}$ carries charge $\gamma(\alpha) \in \widehat{\mathcal{Z}}$ under the 1-form symmetry $\mathcal{Z}$, leading to the mixed anomaly (144) between the two 1-form symmetries.

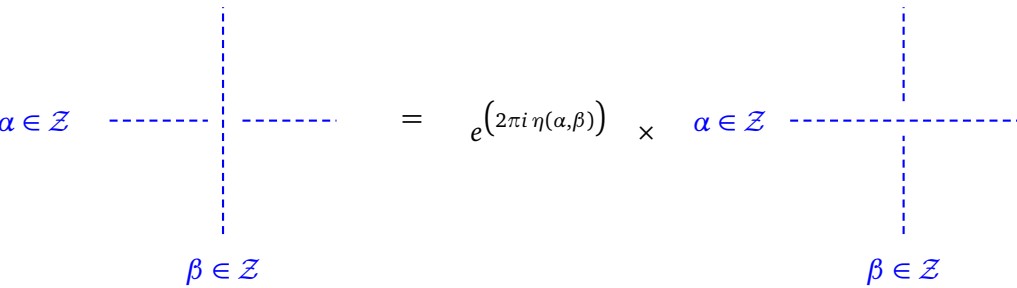

Figure 15: Moving a piece carrying $\alpha \in \mathcal{Z}$ in the 1-cycle Poincaré dual to $w_2$ across another piece carrying $\alpha \in \mathcal{Z}$ generates a phase $\exp\big(2\pi i\, \eta(\alpha, \beta)\big)$.

**Type of anomaly being studied.**    We study anomalies of the form

$$\mathcal{A}_4 = \exp\left( 2\pi i \int \mathcal{P}_\sigma(w_2) \right), \tag{145}$$

associated to a quadratic function

$$\sigma : \ \mathcal{Z} \to \mathbb{R}/\mathbb{Z}, \tag{146}$$

with the associated bilinear form being

$$\eta : \ \mathcal{Z} \times \mathcal{Z} \to \mathbb{R}/\mathbb{Z}. \tag{147}$$

**Pictorial representation of the anomaly.**    Pictorially, the anomaly is seen by passing a line segment carrying $\alpha \in \mathcal{Z}$ in the 1-cycle Poincaré dual to the background 2-cocycle $w_2$ across a line segment carrying $\beta \in \mathcal{Z}$ in the same 1-cycle. This move results in the correlation function jumping by the phase

$$\exp\big(2\pi i\, \eta(\alpha, \beta)\big). \tag{148}$$

See figure 15.

**Anomaly from monopole operators for 0-form symmetry.**    Consider a monopole operator $M$ living at the end of a vortex line defect $V$ inducing a 0-form symmetry background associated to a co-character

$$\phi : \ U(1) \to \mathcal{F}, \tag{149}$$

$$V \quad \text{———} \quad \gamma(\alpha_\phi) \in \widehat{\mathcal{Z}}$$
$$M$$

Figure 16: A monopole operator $M$ (shown in red) for the 0-form symmetry group $\mathcal{F}$ arising at the end of a vortex line defect $V$ must carry a representation $R$ of $F$ whose charge under the 0-form center $\mathcal{Z}$ is given by the element $\gamma(\alpha_\phi) \in \widehat{\mathcal{Z}}$, where $\alpha_\phi \in \mathcal{Z}$ is the obstruction to regarding the co-character $\phi$ for $\mathcal{F}$ as a co-character for the central extension $F$.

with obstruction $\alpha_\phi \in \mathcal{Z}$ of lifting the co-character $\phi$ to a co-character of $F$. The above anomaly is the statement that $M$ must transform in a representation of $F$ whose charge under $\mathcal{Z}$ is

$$\gamma(\alpha) \in \widehat{\mathcal{Z}}, \tag{150}$$

where

$$\gamma : \mathcal{Z} \to \widehat{\mathcal{Z}}, \tag{151}$$

is the homomorphism descending from $\eta$. See figure 16.

Equivalently, the vortex line defect $V$ carrying $\alpha_\phi \in \mathcal{Z}$ lies in the equivalence class $\gamma(\alpha_\phi) \in \widehat{\mathcal{Z}} \subseteq \widehat{\mathcal{E}}$.

**Gauging 0-form symmetry.** Consider gauging the 0-form symmetry $\mathfrak{f}$ via an $F$ gauge group. The resulting theory then has a 1-form symmetry

$$\Gamma^{(1)}_r = \mathcal{Z}, \tag{152}$$

coming from the center of the gauge group $F$ and the anomaly (145) descends to the anomaly

$$\mathcal{A}_4 = \exp\left( 2\pi i \int \mathcal{P}_\sigma(B_2) \right), \tag{153}$$

where $B_2$ is the $\mathcal{Z}$ valued background field for the $\Gamma^{(1)}_r$ 1-form symmetry of the resulting theory. Notice that this is a 1-form anomaly of the type discussed in section 5.1.

**Anomaly (153) from solitonic defects** As we have discussed above, the anomaly (153) makes solitonic line defects inducing 1-form symmetry backgrounds charged non-trivially under the 1-form symmetry. We can see this happening explicitly by applying the gauging procedure to the vortex-monopole configuration of figure 16. After gauging, the vortex defect $V$ becomes a line defect inducing the 1-form symmetry background $\alpha_\phi \in \mathcal{Z}$, $M$ becomes a fractional gauge monopole operator, and the background Wilson line becomes a gauge Wilson line defect having charge $\gamma(\alpha_\phi)$ under the $\mathcal{Z}$ 1-form symmetry. Thus, $V$ lies in the equivalence class $\gamma(\alpha_\phi)$ under the equivalence relation (19), and hence carries the charge $\gamma(\alpha_\phi)$ under the $\mathcal{Z}$ 1-form symmetry.

## 5.4 Anomaly for 2-group symmetry

In this subsection, we consider anomalies for a 2-group symmetry comprising of a 1-form symmetry $\Gamma^{(1)}$ and a 0-form symmetry group $\mathcal{F}$. We consider anomalies that can be represented purely in terms of the $\mathcal{E}$-valued background field $B_w$ for the 2-group symmetry.

We study anomalies of the form

$$\mathcal{A}_4 = \exp\left( 2\pi i \int \mathcal{P}_\sigma(B_w) \right), \tag{154}$$

associated to a quadratic function

$$\sigma : \mathcal{E} \to \mathbb{R}/\mathbb{Z}, \tag{155}$$

with the associated bilinear form being

$$\eta : \mathcal{E} \times \mathcal{E} \to \mathbb{R}/\mathbb{Z}. \tag{156}$$

The $\mathcal{E}$ valued 1-cycle Poincaré dual to the background 2-cocycle $B_w$ for the 2-group symmetry can be represented in terms of a network of lines, such that each line segment carries an element of $\mathcal{E}$. Consider passing a line segment carrying $\alpha \in \mathcal{E}$ in the $\widehat{B}_w$ background across a line segment carrying $\beta \in \mathcal{E}$ in the same background. This move results in the correlation function jumping by the phase

$$\exp\bigl(2\pi i\, \eta(\alpha,\beta)\bigr). \tag{157}$$

**Anomaly from solitonic defects inducing 2-group backgrounds.**   Consider a vortex line defect $V$ inducing a vortex configuration associated to a co-character $\phi$ of $\mathcal{F}$ with obstruction $\alpha_\phi \in \mathcal{Z}$. Then, $V$ induces a 2-group background with background field $B_w$ taking a value $\widetilde{\alpha}_\phi \in \pi^{-1}(\alpha_\phi) \subset \mathcal{E}$ in the vicinity of $V$. The above anomaly can be reinterpreted as stating that $V$ lies in the equivalence class

$$\gamma(\widetilde{\alpha}_\phi) \in \widehat{\mathcal{E}}, \tag{158}$$

under the equivalence relation (28), where

$$\gamma : \mathcal{E} \to \widehat{\mathcal{E}}, \tag{159}$$

is the homomorphism descending from $\eta$. In particular, $V$ is charged under the 1-form symmetry $\Gamma^{(1)}$ with the charge

$$\widehat{\pi} \circ \gamma(\widetilde{\alpha}_\phi) \in \widehat{\Gamma^{(1)}}, \tag{160}$$

where

$$\widehat{\pi} : \widehat{\mathcal{E}} \to \widehat{\Gamma^{(1)}}, \tag{161}$$

is the projection map in the short exact sequence (29).

**Gauging 0-form symmetry.**   Consider gauging the 0-form symmetry $\mathfrak{f}$ via an $F$ gauge group. The resulting theory then has a 1-form symmetry

$$\Gamma_r^{(1)} = \mathcal{E}, \tag{162}$$

and the anomaly (154) descends to the anomaly

$$\mathcal{A}_4 = \exp\left(2\pi i \int \mathcal{P}_\sigma(B_2^r)\right), \tag{163}$$

where $B_2^r$ is the $\mathcal{E}$ valued background field for the $\Gamma_r^{(1)}$ 1-form symmetry of the resulting theory. This is now a 1-form anomaly of the type discussed in section 5.1.

The above anomaly (163) can also be deduced from an operator point of view as follows. After gauging, a vortex line defect inducing $\widetilde{\alpha}_\phi \in \mathcal{E}$ descends to a line defect inducing a background $\widetilde{\alpha}_\phi$ for the resulting 1-form symmetry $\mathcal{E}$. The fact that it lies in the equivalence class $\gamma(\widetilde{\alpha}_\phi) \in \widehat{\mathcal{E}}$ means that it has charge $\gamma(\widetilde{\alpha}_\phi)$ under the $\mathcal{E}$ 1-form symmetry obtained after gauging $F$, leading to the anomaly (163).

# 6 Applications to 3d gauge theories

In this section, we provide a recipe for computing generalized symmetries and anomalies (of the types being studied in this paper) for a general class of 3d gauge theories. During the course of this discussion, we require charges of various kinds of monopole operators, the expressions for which are obtained by generalizing the expressions presented in [100–102] (see also [103–108]).

## 6.1 1-form symmetry

**Gauge group.** We consider gauge theories whose gauge groups do not contain discrete factors. The gauge group $\mathcal{G}$ is associated to a gauge algebra $\mathfrak{g}$ which takes the form

$$\mathfrak{g} = \bigoplus_i \mathfrak{g}_i \oplus \bigoplus_a \mathfrak{u}(1)_a \,, \tag{164}$$

where $\mathfrak{g}_i$ is a simple compact non-Abelian Lie algebra and $\mathfrak{u}(1)_a$ is a copy of the Abelian Lie algebra $\mathfrak{u}(1)$. The gauge group can then be expressed as

$$\mathcal{G} = \frac{\prod_i G_i \times \prod_a U(1)_a}{Z} \,, \tag{165}$$

where $G_i$ is the simply connected, compact Lie group with Lie algebra $\mathfrak{g}_i$ and $U(1)_a$ is a copy of $U(1)$ whose Lie algebra is $\mathfrak{u}(1)_a$. The normalization of $U(1)_a$ must be such that each matter field carries a non-fractional integer charge under it. In other words, every matter field transforms in an allowed representation under $U(1)_a$. On the other hand, $Z$ is a subgroup of the center $\prod_i Z_{G_i} \times \prod_a U(1)_a$ of the group $\prod_i G_i \times \prod_a U(1)_a$, where $Z_{G_i}$ is the center of $G_i$. The fact that $Z$ does not participate in the gauge group $\mathcal{G}$ requires for consistency that no matter field is charged under $Z$.

**Chern-Simons terms.** We can include tree-level Chern-Simons terms which are described in terms of a vector $k_i$ describing CS level for $\mathfrak{g}_i$ and a matrix $k_{ab}$ describing mixed CS term between $\mathfrak{u}(1)_a$ and $\mathfrak{u}(1)_b$. There are restrictions on the possible values of these CS terms depending on the spectrum of fermions in the matter content, and on the choice $\mathcal{G}$ of the gauge group uplifting the gauge algebra $\mathfrak{g}$. These restrictions take the form of quantization conditions which can be understood as the requirement that gauge monopole operators have well-defined charges under the center

$$Z_{\mathcal{G}} = \frac{\prod_i Z_{G_i} \times \prod_a U(1)_a}{Z} \,, \tag{166}$$

of the gauge group $\mathcal{G}$.

We discuss these quantization conditions in what follows. If these quantization conditions are not satisfied, then the gauge group $\mathcal{G}$ suffers from a *global gauge anomaly*,[8] which renders the gauge theory inconsistent.

**Determination of 1-form symmetry.** For a 3d gauge theory with a connected gauge group, the 1-form symmetry group is purely 'electric', i.e. it is obtained from the center $Z_{\mathcal{G}}$ of the gauge group $\mathcal{G}$. The 1-form symmetry group is obtained as the maximal subgroup of $Z_{\mathcal{G}}$ which leaves matter content and non-fractional gauge monopole operators invariant.

---

[8]Not to be confused with a mixed anomaly between a global symmetry and the gauge symmetry.

Let us begin by considering matter content first. Consider a matter field $\chi$ transforming in an irrep $R_\chi$ of $\mathcal{G}$. It carries a charge $[R_\chi] \in \widehat{Z}_\mathcal{G}$ under $Z_\mathcal{G}$. Define a subgroup $Y_{\text{mat}}$ of the group $\widehat{Z}_\mathcal{G}$

$$Y_{\text{mat}} := \text{Span}\big([R_\chi]\big), \tag{167}$$

obtained by taking the $\mathbb{Z}$-span[9] of elements $[R_\chi]$ for all possible matter fields $\chi$. Let us also define

$$\widehat{\mathcal{O}}_{\text{mat}} := \widehat{Z}_\mathcal{G}/Y_{\text{mat}}. \tag{168}$$

Now, let us consider a non-fractional gauge monopole operator $M_\phi$ inducing a monopole configuration described by a co-character

$$\phi : U(1) \to \mathcal{G}, \tag{169}$$

of the gauge group. Let $R_\phi$ be the representation of $\mathcal{G}$ formed by $M_\phi$ which carries a charge $[R_\phi] \in \widehat{Z}_\mathcal{G}$ under $Z_\mathcal{G}$. Applying the projection map $\widehat{Z}_\mathcal{G} \to \widehat{\mathcal{O}}_{\text{mat}}$ to the element $[R_\phi]$, we obtain an element $q_\phi \in \widehat{\mathcal{O}}_{\text{mat}}$.

We claim that any other monopole operator $M'_\phi$ inducing the same monopole configuration described by $\phi$ leads to the same element $q_\phi \in \widehat{\mathcal{O}}_{\text{mat}}$ even though its representation $R'_\phi$ under $\mathcal{G}$ is in general different from the representation $R_\phi$ of $M_\phi$. This claim relies on expected OPE properties of monopole operators:

Consider a monopole operator $M_{\phi^{-1}}$ for the inverse co-character $\phi^{-1}$ of $\mathcal{G}$. Let $q_{\phi^{-1}}$ be the element of $\widehat{\mathcal{O}}_{\text{mat}}$ capture its charge. Now consider the OPE of $M_\phi$ and $M_{\phi^{-1}}$. The operators resulting from this OPE should all be non-monopole operators, so should arise as combinations of matter fields. Moreover, the gauge representations of these operators should have the charge

$$q_\phi + q_{\phi^{-1}} \in \widehat{\mathcal{O}}_{\text{mat}}. \tag{170}$$

But, we know that matter fields have the charge $0 \in \widehat{\mathcal{O}}_{\text{mat}}$. This leads to the conclusion

$$q_\phi = -q_{\phi^{-1}}. \tag{171}$$

Similarly, considering the OPE of $M'_\phi$ and $M_{\phi^{-1}}$, we are lead to the conclusion

$$q'_\phi = -q_{\phi^{-1}}, \tag{172}$$

where $q'_\phi \in \widehat{\mathcal{O}}_{\text{mat}}$ captures the charge of $M'_\phi$. Combining the two results, we find

$$q'_\phi = q_\phi, \tag{173}$$

which is what we wanted to show.

As a result of this argument, we obtain a unique element $q_\phi \in \widehat{\mathcal{O}}_{\text{mat}}$ for each co-character $\phi$ of $\mathcal{G}$. Let us define

$$Y_{\text{mono}} := \text{Span}_\mathbb{Z}(\{q_\phi\}), \tag{174}$$

obtained by taking the $\mathbb{Z}$-span of elements $q_\phi$ for all possible co-characters $\phi$. This is a subgroup of $\widehat{\mathcal{O}}_{\text{mat}}$. This lets us describe the Pontryagin dual of the 1-form symmetry group $\Gamma^{(1)}$ as

$$\widehat{\Gamma}^{(1)} = \widehat{\mathcal{O}}_{\text{mat}}/Y_{\text{mono}}. \tag{175}$$

---

[9]We are here regarding the Abelian group $\widehat{Z}_\mathcal{G}$ as a $\mathbb{Z}$-module.

**Charges of non-fractional gauge monopole operators.**    To complete the computation of $\Gamma^{(1)}$, we now only need to describe the computation of the charge $q_\phi \in \widehat{\mathcal{O}}_{\text{mat}}$ for each co-character $\phi$. We compute $q_\phi$ using the charges of a special monopole operator that we call $M_\phi(\mathcal{G})$. For theories with $\mathcal{N} \geq 2$ supersymmetry, this can be recognized as the 'bare' BPS monopole operator associated to the co-character $\phi$ of $\mathcal{G}$. For $\mathcal{N} = 1$ and $\mathcal{N} = 0$ theories, we propose that there exists, for each $\phi$, at least one monopole operator carrying the charges described below.

The charges of $M_\phi(\mathcal{G})$ receive two contributions: one from the CS terms, and the other from the fermions in the matter content. From the Chern-Simons level $k_i$, the operator $M_\phi(\mathcal{G})$ obtains a charge (see references at the beginning of this section for more details)

$$q^{(k)}_{\phi,i,\alpha} = -\frac{k_i}{h_i^\vee} \sum_{\rho_i} q_{i,\alpha}(\rho_i) q_\phi(\rho_i), \tag{176}$$

under the $U(1)_{i,\alpha}$ component of the $\prod_\alpha U(1)_{i,\alpha}$ maximal torus of $G_i$, where the sum is over positive roots $\rho_i$ of $\mathfrak{g}_i$, $h_i^\vee$ is the dual Coxeter number for $\mathfrak{g}_i$, $q_{i,\alpha}(\rho_i)$ is the charge under $U(1)_{i,\alpha}$ of $\rho_i$, and $q_\phi(\rho_i)$ is the charge of $\rho_i$ under the subgroup $U(1)_\phi \subseteq \mathcal{G}$ defined by the co-character $\phi$. Note that $q_\phi(\rho_i)$ can be fractional if the projection of $Z$ onto $Z_{G_i}$ is non-trivial and the intersection $U(1)_\phi \cap G_i$ is not a complete circle, but only a segment inside $G_i$. Similarly, from the Chern-Simons levels $k_{ab}$, the monopole operator $M_\phi(\mathcal{G})$ obtains a charge

$$q^{(k)}_{\phi,a} = -\sum_b k_{ab} \phi_b, \tag{177}$$

under $U(1)_a$, where $\phi_b$ is the winding number of $\phi$ along $U(1)_b$. The winding $\phi_b$ is fractional if the projection of $Z$ onto $U(1)_b$ is non-trivial and $U(1)_\phi \cap U(1)_b$ is a segment rather than a circle.

Now let us describe the contributions of fermions. Decompose the fermions into irreducible 1-dimensional representations under the group $\prod_i \prod_\alpha U(1)_{i,\alpha} \times \prod_a U(1)_a$. Let $\psi$ parametrize these representations. Then, from the fermion $\psi$, the monopole operator $M_\phi(\mathcal{G})$ obtains a charge

$$q^{(\psi)}_{\phi,i,\alpha} = -\frac{1}{2} q_{i,\alpha}(\psi) |q_\phi(\psi)|, \tag{178}$$

under $U(1)_{i,\alpha}$, where $q_{i,\alpha}(\psi)$ is the charge of $\psi$ under $U(1)_{i,\alpha}$ and $q_\phi(\psi)$ is the charge of $\psi$ under $U(1)_\phi$. Similarly, from the fermion $\psi$, the monopole operator $M_\phi(\mathcal{G})$ obtains a charge

$$q^{(\psi)}_{\phi,a} = -\frac{1}{2} q_a(\psi) |q_\phi(\psi)|, \tag{179}$$

under $U(1)_a$, where $q_a(\psi)$ is the charge of $\psi$ under $U(1)_a$.

In total, the charge of monopole operator $M_\phi(\mathcal{G})$ under $U(1)_{i,\alpha}$ is

$$q_{\phi,i,\alpha} = q^{(k)}_{\phi,i,\alpha} + \sum_\psi q^{(\psi)}_{\phi,i,\alpha}, \tag{180}$$

and its charge under $U(1)_a$ is

$$q_{\phi,a} = q^{(k)}_{\phi,a} + \sum_\psi q^{(\psi)}_{\phi,a}. \tag{181}$$

The full quantization conditions on the Chern-Simons terms are discussed later, but a part of the condition can already be described. The quantization of Chern-Simons terms $k_i, k_{ab}$ must be such that $q_{\phi,i,\alpha}$ and $q_{\phi,a}$ are integers for all possible values of $i, \alpha, a, \phi$.

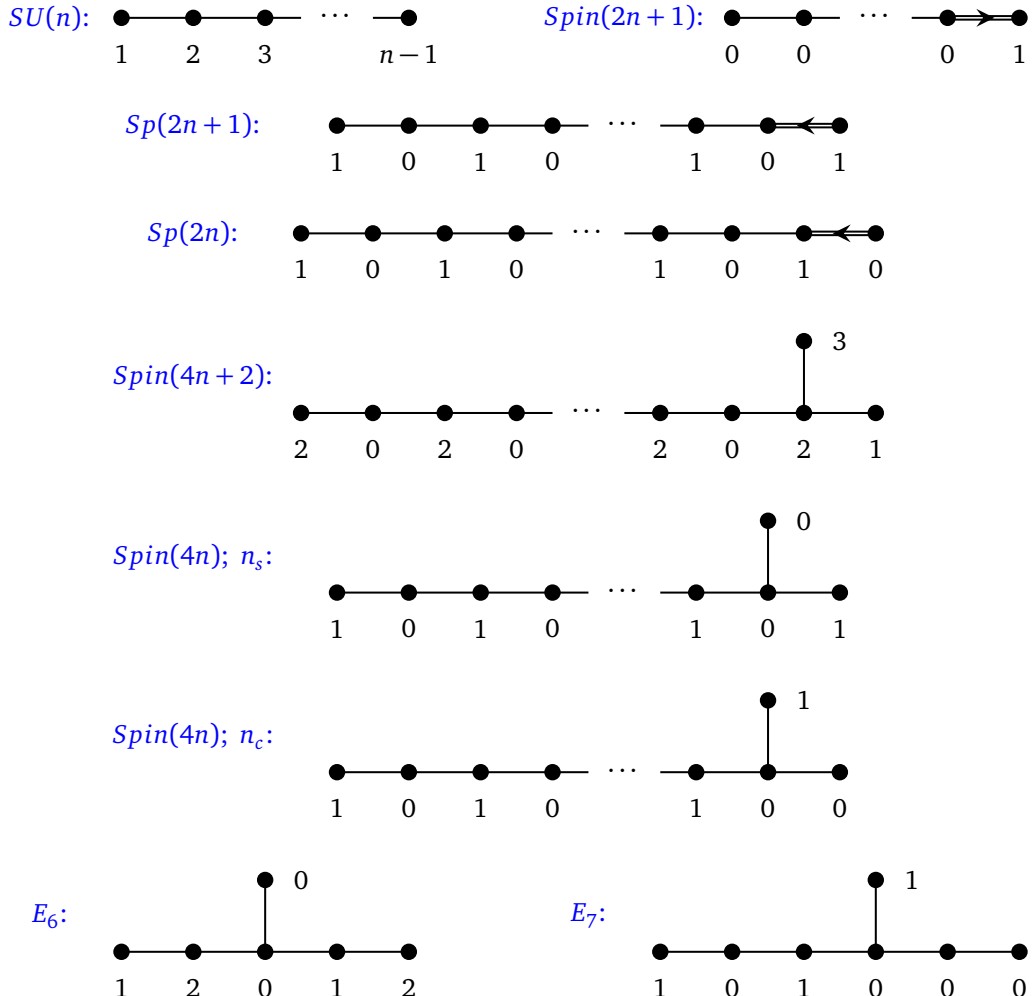

Figure 17: The figure displays the integers $n_{i,\alpha}$ for each simply connected simple compact Lie group, which are in one-to-one correspondence with the nodes in the corresponding Dynkin diagram. For $Spin(4n)$, there are two sets of such integers $n_{s,i,\alpha}$ and $n_{c,i,\alpha}$, which we label by $n_s$ and $n_c$ respectively in the figure. For $E_8, F_4, G_2$, the integers $n_{i,\alpha}$ are all zero, and hence are not displayed in the above figure.

Now, using the charges $q_{\phi,i,\alpha}$, we can deduce the charge $q_{\phi,i} \in \widehat{Z}_{G_i}$ of $M_\phi(\mathcal{G})$ under the center $Z_{G_i}$ as

$$q_{\phi,i} = n_{i,\alpha} q_{\phi,i,\alpha} \pmod{n_i}, \tag{182}$$

for $G_i = SU(n), Spin(2n+1), Sp(n), Spin(4n+2), E_n, F_4, G_2$, where $\widehat{Z}_{G_i} \simeq \mathbb{Z}_{n_i}$ (note that $n_i = 1$ for $G_i = E_8, F_4, G_2$) and $n_{i,\alpha}$ are certain integers collected in figure 17, and as

$$q_{\phi,i} = \left( n_{s,i,\alpha} q_{\phi,i,\alpha} \pmod{2}, \; n_{c,i,\alpha} q_{\phi,i,\alpha} \pmod{2} \right), \tag{183}$$

for $G_i = Spin(4n)$, where $\widehat{Z}_{G_i} \simeq \mathbb{Z}_2 \times \mathbb{Z}_2$ and $n_{s,i,\alpha}, n_{c,i,\alpha}$ are certain integers collected in figure 17.

The charge $(q_{\phi,i}, q_{\phi,a})$ is valued in $\prod_i \widehat{Z}_{G_i} \times \prod_a \widehat{U(1)}_a$. Applying the projection map

$$\prod_i \widehat{Z}_{G_i} \times \prod_a \widehat{U(1)}_a \to \widehat{Z}, \tag{184}$$

Pontryagin dual to the injection map

$$Z \to \prod_i Z_{G_i} \times \prod_a U(1)_a \,, \tag{185}$$

onto $(q_{\phi,i}, q_{\phi,a})$, we obtain an element $q_{\phi,Z} \in \widehat{Z}$ describing the charge of the monopole operator $M_\phi(\mathcal{G})$ under the subgroup $Z$ of $\prod_i Z_{G_i} \times \prod_a U(1)_a$. Since $Z$ is not a part of the gauge group, the quantization condition on the CS terms (which is discussed in full generality later) must be such that we have

$$q_{\phi,Z} = 0 \,, \tag{186}$$

for all possible $\phi$.

The last condition $q_{\phi,Z} = 0$ implies that $[R_\phi] := (q_{\phi,i}, q_{\phi,a})$ is an element of the subgroup $\widehat{Z}_{\mathcal{G}} \subseteq \prod_i \widehat{Z}_{G_i} \times \prod_a \widehat{U(1)}_a$, where $\widehat{Z}_{\mathcal{G}}$ is the Pontryagin dual of the center $Z_{\mathcal{G}}$ of the gauge group. We obtain $q_\phi \in \widehat{\mathcal{O}}_{\mathrm{mat}}$ by applying the projection map $\widehat{Z}_{\mathcal{G}} \to \widehat{\mathcal{O}}_{\mathrm{mat}}$ to the element $[R_\phi] \in \widehat{Z}_{\mathcal{G}}$.

## 6.2 1-form anomaly and quantization of gauge Chern-Simons terms

**General gauge monopole operators.** To describe the most general quantization condition on the gauge Chern-Simons terms, and anomaly for 1-form symmetry, we need to study the charges of general fractional and non-fractional gauge monopole operators inducing monopole configurations associated to co-characters

$$\phi' : U(1) \to G \,, \tag{187}$$

where

$$G := \mathcal{G}/\Gamma^{(1)} \,. \tag{188}$$

Recall that we call such a gauge monopole operator fractional if $\phi'$ cannot be lifted to a co-character $\phi$ for the gauge group $\mathcal{G}$.

**1-form anomaly.** Using information about charges of fractional gauge monopole operators, we can compute 1-form anomaly of the 3d gauge theory under study. Recall that such an anomaly is valid only if the backgrounds for 0-form symmetries forming a non-trivial 2-group with the 1-form symmetry $\Gamma^{(1)}$ are turned off. Otherwise, the 1-form anomaly is lifted to a 2-group anomaly, which we discuss in later subsections.

We argued above that any non-fractional gauge monopole operator associated to a co-character $\phi$ of $\mathcal{G}$ has a unique charge $q_\phi \in \widehat{\mathcal{O}}_{\mathrm{mat}}$. By a similar argument, a general gauge monopole operator associated to a co-character $\phi'$ of $G$ also has a unique charge $q_{\phi'} \in \widehat{\mathcal{O}}_{\mathrm{mat}}$. Let $q'_{\phi'} \in \widehat{\Gamma}^{(1)}$ be the element obtained by applying the projection map $\widehat{\mathcal{O}}_{\mathrm{mat}} \to \widehat{\Gamma}^{(1)}$ to the element $q_{\phi'} \in \widehat{\mathcal{O}}_{\mathrm{mat}}$. We know that $q'_{\phi'} = 0$ if $\phi'$ can be lifted to a co-character $\phi$ of $\mathcal{G}$. This implies, using again an OPE argument, that

$$q'_{\phi'_1} = q'_{\phi'_2} \,, \tag{189}$$

if

$$\alpha_{\phi'_1} = \alpha_{\phi'_2} \,, \tag{190}$$

where $\alpha_{\phi'_i} \in \Gamma^{(1)}$ is the obstruction of lifting $\phi'_i$ to a co-character for $\mathcal{G}$.

Thus, we obtain a function

$$\gamma : \Gamma^{(1)} \to \widehat{\Gamma}^{(1)} \,, \tag{191}$$

obtained by mapping the obstruction $\alpha_{\phi'}$ to the charge $q'_{\phi'}$. This function is actually a group homomorphism as can be argued by a similar OPE based argument as above. This homomorphism $\gamma$ captures the 1-form anomaly as discussed in section 5.1.

**Charges of general gauge monopole operators.** To complete the discussion of 1-form anomaly, we need to describe the computation of the charge $Q_{\phi'} \in \widehat{\Gamma}^{(1)}$. We compute $q'_{\phi'}$ using the charges of a special monopole operator that we call $M_{\phi'}(G)$ inducing the monopole configuration associated to the co-character $\phi'$ of $G$. For theories with $\mathcal{N} \geq 2$ supersymmetry, this special monopole operator can be recognized as the 'bare' BPS monopole operator associated to the co-character $\phi'$ of $G$. For $\mathcal{N} = 1, 0$ theories, we propose that there exists, for each $\phi'$, at least one monopole operator carrying the charges described below.

The charges of $M_{\phi'}(G)$ are obtained in the same way as the charges for non-fractional gauge monopole operators $M_\phi(\mathcal{G})$ were obtained above, with the only difference being that $\phi$ is now replaced by $\phi'$ which is a co-character into $G$ and so has more fractional windings around various $U(1)$s as compared to windings of co-characters $\phi$ into $\mathcal{G}$. Similar to (176) and (177), the Chern-Simons terms contribute charges

$$q^{(k)}_{\phi',i,\alpha} = -\frac{k_i}{h_i^\vee} \sum_{\rho_i} q_{i,\alpha}(\rho_i) q_{\phi'}(\rho_i), \tag{192}$$

under $U(1)_{i,\alpha}$, where $q_{\phi'}(\rho_i)$ is the (possibly fractional) charge of the positive root $\rho_i$ under $U(1)_{\phi'} \subseteq G$ defined by the co-character $\phi'$, and

$$q^{(k)}_{\phi',a} = -\sum_b k_{ab} \phi'_b, \tag{193}$$

under $U(1)_a$ where $\phi'_b$ is the (possibly fractional) winding number of $\phi'$ along $U(1)_b$. Similar to (178) and (179), the fermion $\psi$ contributes charges

$$q^{(\psi)}_{\phi',i,\alpha} = -\frac{1}{2} q_{i,\alpha}(\psi) |q_{\phi'}(\psi)|, \tag{194}$$

under $U(1)_{i,\alpha}$ and

$$q^{(\psi)}_{\phi',a} = -\frac{1}{2} q_a(\psi) |q_{\phi'}(\psi)|, \tag{195}$$

under $U(1)_a$, where $q_{\phi'}(\psi)$ is the charge of $\psi$ under $U(1)_{\phi'}$.

**First quantization condition.** Similar to (180) and (181),

$$q_{\phi',i,\alpha} = q^{(k)}_{\phi',i,\alpha} + \sum_\psi q^{(\psi)}_{\phi',i,\alpha}, \tag{196}$$

and

$$q_{\phi',a} = q^{(k)}_{\phi',a} + \sum_\psi q^{(\psi)}_{\phi',a}, \tag{197}$$

are the total charges of the fractional gauge monopole operator $M_{\phi'}(G)$ under $U(1)_{i,\alpha}$ and $U(1)_a$ respectively.

The first quantization condition[10] on Chern-Simons levels is that $q_{\phi',i,\alpha}$ and $q_{\phi',a}$ have to be integers for all possible values of $i, \alpha, a, \phi'$.

---

[10]There are two types of charge quantization conditions: the first ensures that the charges of monopoles are correctly quantized for each individual gauge group factor. The second ensures that the charges are compatible with the quotient by $Z$.

**Second quantization condition.** Similar to (182) and (183), the charge of $M_{\phi'}(G)$ under $Z_{G_i}$ is obtained as

$$q_{\phi',i} = n_{i,\alpha}q_{\phi',i,\alpha} \pmod{n_i}, \tag{198}$$

for $G_i = SU(n), Spin(2n+1), Sp(n), Spin(4n+2), E_6, E_7$, and as

$$q_{\phi',i} = \left( n_{s,i,\alpha}q_{\phi',i,\alpha} \pmod 2, \ n_{c,i,\alpha}q_{\phi',i,\alpha} \pmod 2 \right), \tag{199}$$

for $G_i = Spin(4n)$.

The second quantization condition on CS terms is obtained by requiring that

$$q_{\phi',Z} = 0, \tag{200}$$

for all possible $\phi'$, where $q_{\phi',Z} \in \widehat{Z}$ is obtained from $(q_{\phi',i}, q_{\phi',a}) \in \prod_i \widehat{Z}_{G_i} \times \prod_a \widehat{U(1)}_a$ by applying the projection map (184). Then, $q_{\phi'} := (q_{\phi',i}, q_{\phi',a}) \in \widehat{Z}_{\mathcal{G}}$.

The last condition $q_{\phi',Z} = 0$ implies that $[R_{\phi'}] := (q_{\phi',i}, q_{\phi',a})$ is an element of the subgroup $\widehat{Z}_{\mathcal{G}} \subseteq \prod_i \widehat{Z}_{G_i} \times \prod_a \widehat{U(1)}_a$, where $\widehat{Z}_{\mathcal{G}}$ is the Pontryagin dual of the center $Z_{\mathcal{G}}$ of the gauge group. We obtain $q_{\phi'} \in \widehat{\mathcal{O}}_{\text{mat}}$ by applying the projection map $\widehat{Z}_{\mathcal{G}} \to \widehat{\mathcal{O}}_{\text{mat}}$ to the element $[R_{\phi'}] \in \widehat{Z}_{\mathcal{G}}$. Similarly, we obtain $q'_{\phi'} \in \widehat{\Gamma}^{(1)}$ by applying the projection map $\widehat{\mathcal{O}}_{\text{mat}} \to \widehat{\Gamma}^{(1)}$ to the element $q_{\phi'} \in \widehat{\mathcal{O}}_{\text{mat}}$.

## 6.3 2-group and 0-form symmetries

Moving forward, we specialize the form of the gauge group to be

$$\mathcal{G} = \prod_i G'_i \times \prod_a U(1)_a, \tag{201}$$

where $G'_i$ is a compact but not necessarily simply connected Lie group with Lie algebra $\mathfrak{g}_i$. This constrains the form of the group $Z$ appearing in the denominator of (165). The reason for imposing this constraint is that it allows an easy identification for the covering group $F_t$ associated to topological 0-form symmetry discussed below.

We only study the continuous part of the 0-form symmetry, which is associated to a 0-form symmetry algebra

$$\mathfrak{f} = \mathfrak{f}_t \oplus \mathfrak{f}_{nt}, \tag{202}$$

where

$$\mathfrak{f}_t = \bigoplus_a \mathfrak{u}(1)'_a, \tag{203}$$

is the topological/magnetic 0-form symmetry such that $\mathfrak{u}(1)'_a$ is identified with the factor $\mathfrak{u}(1)_a$ in the gauge algebra $\mathfrak{g}$, and

$$\mathfrak{f}_{nt} = \bigoplus_I \mathfrak{f}_I \oplus \bigoplus_A \mathfrak{u}(1)_A, \tag{204}$$

collects other 0-form symmetries, where $\mathfrak{f}_I$ is a simple compact non-Abelian Lie algebra and $\mathfrak{u}(1)_A$ is a copy of the Abelian Lie algebra $\mathfrak{u}(1)$. We begin by considering a group

$$F = F_t \times F_{nt}, \tag{205}$$

which in general is a central extension of the 0-form symmetry group $\mathcal{F}$. The group $F_t$ with Lie algebra $\mathfrak{f}_t$ is taken to have the form

$$F_t = \prod_a U(1)'_a, \tag{206}$$

where $U(1)'_a \subseteq F_t$ is identified with the factor $U(1)_a$ in the gauge group $\mathcal{G}$, while the group $F_{nt}$ with Lie algebra $\mathfrak{f}_{nt}$ takes the form

$$F_{nt} = \prod_I F_I \times \prod_A U(1)_A, \tag{207}$$

where $F_I$ is the simply connected, compact Lie group with Lie algebra $\mathfrak{f}_I$ and $U(1)_A$ is a copy of $U(1)$ whose Lie algebra is $\mathfrak{u}(1)_A$. The normalization of $U(1)_A$ must be such that each matter field carries a non-fractional integer charge under it. In other words, every matter field transforms in an allowed representation under $U(1)_A$.

Note for later purposes that the center of $F$ is

$$Z_F = \prod_a U(1)'_a \times \prod_I Z_{F_I} \times \prod_A U(1)_A, \tag{208}$$

where $Z_{F_I}$ is the center of $F_I$.

**Background and mixed background-gauge Chern-Simons terms.** We can include tree-level background Chern-Simons terms, which are described in terms of a vector $k_I$ describing CS level for $\mathfrak{f}_I$, and a matrix $k_{AB}$ describing mixed CS terms between $\mathfrak{u}(1)_A$ and $\mathfrak{u}(1)_B$

We can also include tree-level mixed background-gauge Chern-Simons terms, which are described in terms of a matrix $k_{Aa}$ describing mixed CS terms between $\mathfrak{u}(1)_A$ and gauge algebra $\mathfrak{u}(1)_a$. There are quantization conditions on the possible values of these CS terms that we discuss later.

**0-form and 2-group symmetry.** As usual for gauge theories, the global form of the 0-form symmetry group and 2-group symmetry can be determined in tandem. The relevant group $\mathcal{E}$ is the subgroup of $Z_{\mathcal{G}} \times Z_F$ that leaves all the matter content and the monopole operators $M_\phi(\mathcal{G})$ for all $\phi$ invariant.[11] We have already discussed the $Z_{\mathcal{G}}$ charges of $M_\phi(\mathcal{G})$ in the previous subsections. In a similar way, we can compute their $Z_F$ charges, which receive contributions from the mixed background-gauge Chern-Simons terms and fermionic matter fields. The detailed computation of these contributions is described later in this subsection. After incorporating these contributions, the monopole operator $M_\phi(\mathcal{G})$ obtains a charge $Q_\phi \in \widehat{Z}_{\mathcal{G}} \times \widehat{Z}_F$. On the other hand, to obtain the $Z_{\mathcal{G}} \times Z_F$ charges of matter fields, let us decompose them into irreps $R$ of $\mathcal{G} \times F$ and let $Q_R \in \widehat{Z}_{\mathcal{G}} \times \widehat{Z}_F$ be the charge under $Z_{\mathcal{G}} \times Z_F$ of $R$. Then, the subgroup

$$\mathrm{Span}(Q_R, Q_\phi) \subseteq \widehat{Z}_{\mathcal{G}} \times \widehat{Z}_F, \tag{209}$$

generated by the charges $Q_R, Q_\phi$ for all $R, \phi$ is the set of $Z_{\mathcal{G}} \times Z_F$ charges occupied by all combinations of matter and monopole operators. Thus, $\mathcal{E}$ is the subgroup of $Z_{\mathcal{G}} \times Z_F$ which takes the elements of the subgroup $\mathrm{Span}(Q_R, Q_\phi) \subseteq \widehat{Z}_{\mathcal{G}} \times \widehat{Z}_F$ to the identity element of $U(1)$ under the natural homomorphism

$$\left(Z_{\mathcal{G}} \times Z_F\right) \times \left(\widehat{Z}_{\mathcal{G}} \times \widehat{Z}_F\right) \to U(1). \tag{210}$$

We can now summarize the rest of the computation: the 0-form symmetry group $\mathcal{F}$ can be written as

$$\mathcal{F} = F/\mathcal{Z}, \tag{211}$$

where

$$\mathcal{Z} = \pi_F(\mathcal{E}), \tag{212}$$

---

[11] $\mathcal{E}$ also leaves invariant other non-fractional gauge monopole operators, as one can argue using OPE arguments discussed in previous subsections, that the $Z_{\mathcal{G}} \times Z_F$ charges of non-fractional gauge monopole operators can be obtained as combinations of charges of monopole operators $M_\phi(\mathcal{G})$ and matter fields.

is the subgroup of $Z_F$ obtained by applying the projection map

$$\pi_F : Z_{\mathcal{G}} \times Z_F \to Z_F. \tag{213}$$

The 1-form symmetry group can also be identified from $\mathcal{E}$ via

$$\Gamma^{(1)} = \mathcal{E} \cap Z_{\mathcal{G}}. \tag{214}$$

The above groups naturally sit in a short exact sequence

$$0 \to \Gamma^{(1)} \to \mathcal{E} \to \mathcal{Z} \to 0, \tag{215}$$

specifying the 2-group symmetry.

**Charges of non-fractional gauge monopole operators.** In the above computation, the charges $Q_\phi$ of non-fractional gauge monopole operators $M_\phi(\mathcal{G})$ played a key role. Earlier we have discussed the $Z_{\mathcal{G}}$ charges of $M_\phi(\mathcal{G})$. Below, we discuss the computation of $Z_F$ charges of $M_\phi(\mathcal{G})$.

There are now three contributions: one coming from the fact that monopole operators are classically charged under $F_t$, and the other two coming from Chern-Simons terms and fermions. The classical $F_t$ charge of $M_\phi(\mathcal{G})$ is

$$q^{(t)}_{\phi,a} = \phi_a, \tag{216}$$

under the topological symmetry $U(1)'_a$, where we recall that $\phi_a$ is the winding number of $\phi$ along the gauge group $U(1)_a$.

From the mixed background-gauge Chern-Simons terms $k_{Aa}$, the monopole operator $M_\phi(\mathcal{G})$ obtains a charge

$$q^{(k)}_{\phi,A} = -\sum_a k_{Aa} \phi_a, \tag{217}$$

under $U(1)_A$.

To describe the contribution of fermions, let us decompose them into irreducible 1-dimensional representations under the group

$$\prod_i \prod_\alpha U(1)_{i,\alpha} \times \prod_a U(1)_a \times \prod_a U(1)'_a \times \prod_I \prod_\alpha U(1)_{I,\alpha} \times \prod_A U(1)_A, \tag{218}$$

where $\prod_\alpha U(1)_{i,\alpha}$ is the maximal torus of $G_i$ (not $G'_i$) and $\prod_\alpha U(1)_{I,\alpha}$ is the maximal torus of $F_I$. Let $\psi$ parametrize these representations. Then, from the fermion $\psi$, the monopole operator $M_\phi(\mathcal{G})$ obtains a charge

$$q^{(\psi)}_{\phi,I,\alpha} = -\frac{1}{2} q_{I,\alpha}(\psi) |q_\phi(\psi)|, \tag{219}$$

under $U(1)_{I,\alpha}$, where $q_{I,\alpha}(\psi)$ is the charge of $\psi$ under $U(1)_{I,\alpha}$. Similarly, from the fermion $\psi$, the monopole operator $M_\phi(\mathcal{G})$ obtains a charge

$$q^{(\psi)}_{\phi,A} = -\frac{1}{2} q_A(\psi) |q_\phi(\psi)|, \tag{220}$$

under $U(1)_A$, where $q_A(\psi)$ is the charge of $\psi$ under $U(1)_A$. The fermions do not contribute to modify the charge of $M_\phi(\mathcal{G})$ under $F_t = \prod_a U(1)'_a$.

In total, monopole operator $M_\phi(\mathcal{G})$ has the charge

$$q_{\phi,I,\alpha} = \sum_\psi q^{(\psi)}_{\phi,I,\alpha}, \tag{221}$$

under $U(1)_{I,\alpha}$, the charge

$$q_{\phi,A} = q_{\phi,A}^{(k)} + \sum_{\psi} q_{\phi,A}^{(\psi)}, \tag{222}$$

under $U(1)_a$, and the charge $q_{\phi,a}^{(t)}$ under the topological symmetry $U(1)_a'$. The integrality of these charges imposes quantization conditions on the Chern-Simons terms $k_{Aa}$. The most general quantization condition is discussed later.

Now, using the charges $q_{\phi,I,\alpha}$, we can deduce the charge $q_{\phi,I} \in \widehat{Z}_{F_I}$ of $M_\phi(\mathcal{G})$ under the center $Z_{F_I}$ of $F_I$ as

$$q_{\phi,I} = n_{I,\alpha} q_{\phi,I,\alpha} \ (\text{mod } n_I), \tag{223}$$

for $F_I = SU(n), Spin(2n+1), Sp(n), Spin(4n), E_6, E_7$, and as

$$q_{\phi,I} = \left( n_{s,I,\alpha} q_{\phi,I,\alpha} \ (\text{mod } 2), \ n_{c,I,\alpha} q_{\phi,I,\alpha} \ (\text{mod } 2) \right), \tag{224}$$

for $F_I = Spin(4n)$. Due to the condition (186), the charge $Q_\phi := (q_{\phi,i}, q_{\phi,a}, q_{\phi,a}', q_{\phi,I}, q_{\phi,A})$ of $M_\phi(\mathcal{G})$ is valued in $\widehat{Z}_{\mathcal{G}} \times \widehat{Z}_F$, where $\widehat{Z}_F$ is the Pontryagin dual of the center $Z_F$ of $F$.

## 6.4 Quantization of Chern-Simons terms involving background fields

Suppose we have chosen a set of gauge Chern-Simons terms $k_i, k_{ab}$ satisfying the quantization conditions of section 6.2. Then, the background and mixed background-gauge Chern-Simons terms are quantized in terms of the gauge Chern-Simons terms, as we discuss in this subsection.

**Mixed monopole operators for 0-form and gauge symmetries**  To describe the most general quantization condition on the background and mixed background-gauge Chern-Simons terms, and to describe the anomalies involving 0-form and 2-group symmetries discussed in the next subsection, we need to study the charges of mixed monopole operators for 0-form and gauge symmetries, which induce monopole configurations associated to co-characters

$$\widetilde{\phi} : \ U(1) \to \mathcal{S}, \tag{225}$$

where

$$\mathcal{S} = \frac{\mathcal{G} \times F}{\mathcal{E}}, \tag{226}$$

is the structure group associated to the gauge theory.

We again propose that the charges of a general monopole operator inducing such a monopole configuration $\widetilde{\phi}$ can be obtained by combining the charges of matter fields with the charges of a special monopole operator, that we call $M_{\widetilde{\phi}}(\mathcal{S})$, inducing the monopole configuration $\widetilde{\phi}$. Let us discuss the computation of charges of monopole operators $M_{\widetilde{\phi}}(\mathcal{S})$ below.

**Charges of mixed monopole operators.**  The computation is an extension of similar computations discussed above, along with a new ingredient to the $U(1)_a$ gauge charge

$$q_{\widetilde{\phi},a}' = \widetilde{\phi}_a', \tag{227}$$

where $\widetilde{\phi}_a'$ is the (possibly fractional) winding number of the subgroup $U(1)_{\widetilde{\phi}} \subseteq \mathcal{S}$ defined by the co-character $\widetilde{\phi}$ along the topological symmetry $U(1)_a'$.[12]

---

[12]This follows from the fact that such a vortex induces $F_a' = \widetilde{\phi}_a' \delta_L$, where $F_a'$ is the field strength for the background $U(1)_a'$, and $\delta_L$ is the 2-form dual to the locus $L$ of the vortex line. This background is represented in the Lagrangian by a coupling $\widetilde{\phi}_a' A_a \wedge \delta_L$, where $A_a$ is the gauge field for $U(1)_a$. Consequently, the vortex carries a gauge charge $\widetilde{\phi}_a'$ under $U(1)_a$.

The classical $F_t$ charge of $M_{\widetilde{\phi}}(\mathcal{S})$ is

$$q^{(t)}_{\widetilde{\phi},a} = \widetilde{\phi}_a, \tag{228}$$

under topological symmetry $U(1)'_a$, where $\widetilde{\phi}_a$ is the (possibly fractional) winding number of the subgroup $U(1)_{\widetilde{\phi}}$. If $\widetilde{\phi}_a$ is fractional, we replace the topological symmetry group $U(1)'_a$ by its $n$-fold cover $\widetilde{U(1)}'_a$ inside $F$, where the cover $\widetilde{U(1)}'_a$ has the property that $n\widetilde{\phi}_a$ is an integer for all possible $\widetilde{\phi}$. Then the charge under $\widetilde{U(1)}'_a$ is

$$\widetilde{q}^{(t)}_{\widetilde{\phi},a} = n\widetilde{\phi}_a. \tag{229}$$

Since we have redefined $F$, we need to redefine $\mathcal{Z}$ by adding a new $\mathbb{Z}_n$ subgroup which acts solely on $\widetilde{U(1)}'_a$, so that only $U(1)'_a$ enters the 0-form symmetry group $\mathcal{F}$.

From the Chern-Simons terms, we obtain the charge

$$q^{(k)}_{\widetilde{\phi},i,\alpha} = -\frac{k_i}{h^{\vee}_i} \sum_{\rho_i} q_{i,\alpha}(\rho_i) q_{\widetilde{\phi}}(\rho_i), \tag{230}$$

under $U(1)_{i,\alpha}$, where $q_{\widetilde{\phi}}(\rho_i)$ is the (possibly fractional) charge of the positive root $\rho_i$ under $U(1)_{\widetilde{\phi}}$; the charge

$$q^{(k)}_{\widetilde{\phi},a} = -\sum_b k_{ab}\widetilde{\phi}_b + \sum_A k_{Aa}\widetilde{\phi}_A, \tag{231}$$

under the gauge group $U(1)_a$, where $\widetilde{\phi}_A$ is the winding number of $U(1)_{\widetilde{\phi}}$ along $U(1)_A$; the charge

$$q^{(k)}_{\widetilde{\phi},I,\alpha} = -\frac{k_I}{h^{\vee}_I} \sum_{\rho_I} q_{I,\alpha}(\rho_I) q_{\widetilde{\phi}}(\rho_I), \tag{232}$$

under $U(1)_{I,\alpha}$, where the sum is over positive roots $\rho_I$ of $\mathfrak{f}_I$, $h^{\vee}_I$ is the dual Coxeter number for $\mathfrak{f}_I$, $q_{I,\alpha}(\rho_I)$ is the charge under $U(1)_{I,\alpha}$ of $\rho_I$, and $q_{\widetilde{\phi}}(\rho_I)$ is the charge of $\rho_I$ under $U(1)_{\widetilde{\phi}}$; and the charge

$$q^{(k)}_{\widetilde{\phi},A} = -\sum_a k_{Aa}\widetilde{\phi}_a - \sum_B k_{AB}\widetilde{\phi}_B, \tag{233}$$

under $U(1)_A$.

On the other hand, from the fermion $\psi$, we obtain the charge

$$q^{(\psi)}_{\widetilde{\phi},i,\alpha} = -\frac{1}{2} q_{i,\alpha}(\psi) |q_{\widetilde{\phi}}(\psi)|, \tag{234}$$

under $U(1)_{i,\alpha}$, where $q_{\widetilde{\phi}}(\psi)$ is the charge of $\psi$ under $U(1)_{\widetilde{\phi}}$; the charge

$$q^{(\psi)}_{\widetilde{\phi},a} = -\frac{1}{2} q_a(\psi) |q_{\widetilde{\phi}}(\psi)|, \tag{235}$$

under $U(1)_a$; zero charge under $\widetilde{U(1)}'_a$; the charge

$$q^{(\psi)}_{\widetilde{\phi},I,\alpha} = -\frac{1}{2} q_{I,\alpha}(\psi) |q_{\widetilde{\phi}}(\psi)|, \tag{236}$$

under $U(1)_{I,\alpha}$; and the charge

$$q^{(\psi)}_{\widetilde{\phi},A} = -\frac{1}{2} q_A(\psi) |q_{\widetilde{\phi}}(\psi)|, \tag{237}$$

under $U(1)_A$.

In total, the monopole operator $M_{\widetilde{\phi}}(\mathcal{S})$ has the charges as summarized in table 1.

Table 1: Charges of the monopole operator $M_{\widetilde{\phi}}(\mathcal{S})$ under various $U(1)$ symmetries. $U(1)_{i,\alpha}$ are Cartans of non-Abelian gauge group factors, $U(1)_a$ are Abelian gauge group factors, $U(1)_{I,\alpha}$ are Cartans of non-Abelian 0-form factors, $U(1)_A$ are Abelian 0-form factors, and $\widetilde{U(1)}'_a$ are suitably rescaled topological symmetries associated to $U(1)_a$ gauge groups.

| G | Charge Label | Charge |
|---|---|---|
| $U(1)_{i,\alpha}$ | $q_{\widetilde{\phi},i,\alpha}$ | $q^{(k)}_{\widetilde{\phi},i,\alpha} + \sum_\psi q^{(\psi)}_{\widetilde{\phi},i,\alpha}$ |
| $U(1)_a$ | $q_{\widetilde{\phi},a}$ | $q'_{\widetilde{\phi},a} + q^{(k)}_{\widetilde{\phi},a} + \sum_\psi q^{(\psi)}_{\widetilde{\phi},a}$ |
| $\widetilde{U(1)}'_a$ | $\widetilde{q}^{(t)}_{\widetilde{\phi},a}$ | $\widetilde{q}^{(t)}_{\widetilde{\phi},a}$ |
| $U(1)_{I,\alpha}$ | $q_{\widetilde{\phi},I,\alpha}$ | $q^{(k)}_{\widetilde{\phi},I,\alpha} + \sum_\psi q^{(\psi)}_{\widetilde{\phi},I,\alpha}$ |
| $U(1)_A$ | $q_{\widetilde{\phi},A}$ | $q^{(k)}_{\widetilde{\phi},A} + \sum_\psi q^{(\psi)}_{\widetilde{\phi},A}$ |

**Necessary quantization conditions.** The gauge charges $q_{\widetilde{\phi},i,\alpha}, q_{\widetilde{\phi},a}$ of the monopole operators $M_{\widetilde{\phi}}(\mathcal{S})$ must be integers to avoid global gauge anomalies that render the gauge theory inconsistent. This provides the *first quantization condition* for the mixed background-gauge Chern-Simons terms $k_{Aa}$, quantizing them in terms of the chosen values for the gauge Chern-Simons terms $k_i$ and $k_{ab}$.

From the charge $q_{\widetilde{\phi},i,\alpha}$, we can deduce the charge $q_{\widetilde{\phi},i} \in \widehat{Z}_{G_i}$ as explained in equations (182) and (183). Then, the charge $(q_{\widetilde{\phi},i}, q_{\widetilde{\phi},a})$ is an element of $\prod_i \widehat{Z}_{G_i} \times \prod_a \widehat{U(1)}_a$. Since $Z$ is not a part of the gauge group, we must have

$$q_{\widetilde{\phi},Z} = 0\,, \tag{238}$$

where $q_{\widetilde{\phi},Z} \in \widehat{Z}$ is the element obtained by applying the projection map

$$\prod_i \widehat{Z}_{G_i} \times \prod_a \widehat{U(1)}_a \to \widehat{Z}\,, \tag{239}$$

on the charge $(q_{\widetilde{\phi},i}, q_{\widetilde{\phi},a})$. This provides the *second quantization condition* for the mixed background-gauge Chern-Simons terms $k_{Aa}$ in terms of the chosen values for the gauge Chern-Simons terms $k_i$ and $k_{ab}$.

Moving forward, let us choose a set of mixed background-gauge Chern-Simons terms $k_{Aa}$ satisfying the above two quantization conditions.

**Other quantization conditions.** We can also demand the 0-form charges $q_{\widetilde{\phi},I,\alpha}, q_{\widetilde{\phi},A}$ to be integral. This acts as a quantization condition for the background Chern-Simons terms $k_I, k_{AB}$ in terms of the chosen values for the gauge and mixed background-gauge Chern-Simons terms $k_i, k_{ab}, k_{Aa}$.

However, this is not a necessary condition. Suppose a charge $q_{\widetilde{\phi},I,\alpha}$ is fractional. Then, this means that the non-Abelian 0-form symmetry $\mathfrak{f}_I$ suffers from a global ABJ anomaly. That is, even though it is a 0-form symmetry at the classical level, it does not lift to a consistent 0-form symmetry of the quantum theory. In such a situation, we can simply remove $\mathfrak{f}_I$ from the full 0-form symmetry algebra $\mathfrak{f}$ and redo the analysis for the 2-group and 0-form symmetries.

If, on the other hand, a charge $q_{\widetilde{\phi},A}$ is fractional, then the corresponding 0-form symmetry $U(1)_A$ suffers from ABJ anomaly. In this case, however, there is a simpler cure of the anomaly. One can simply change the normalization of $U(1)_A$, as we did for $U(1)'_a$ earlier, to get rid of such ABJ anomalies.

Moving forward, we assume that we have chosen a set of background Chern-Simons terms $k_I, k_{AB}$ consistent with the above quantization condition.

## 6.5 Anomalies

**2-group anomaly.** Our above computation of the charges of mixed monopole operators allows us to extract the anomaly of the 2-group symmetry of the 3d gauge theory.

To determine this anomaly, consider a monopole operator $M_{\widetilde{\phi}}(\mathcal{S})$ with associated co-character $\widetilde{\phi}$ having obstruction $\alpha_\phi \in \mathcal{E}$ of being regarded as a $\mathcal{G} \times F$ co-character. Its charge $q_{\widetilde{\phi}} := (q_{\widetilde{\phi},i}, q_{\widetilde{\phi},a}, \widetilde{q}^{(t)}_{\widetilde{\phi},a}, q_{\widetilde{\phi},I}, q_{\widetilde{\phi},A})$ becomes an element of $\widehat{Z}_\mathcal{G} \times \widehat{Z}_F$ after imposing the quantization condition $q_{\widetilde{\phi},Z} = 0$. From $q_{\widetilde{\phi}}$, we can extract an element $\beta_{\widetilde{\phi}} \in \widehat{\mathcal{E}}$ by applying the projection map

$$\widehat{Z}_\mathcal{G} \times \widehat{Z}_F \to \widehat{\mathcal{E}}, \tag{240}$$

to $q_{\widetilde{\phi}}$. Define a map

$$\gamma : \mathcal{E} \to \widehat{\mathcal{E}}, \tag{241}$$

by sending the element $\alpha_{\widetilde{\phi}} \in \mathcal{E}$ to the element $\beta_{\widetilde{\phi}} \in \widehat{\mathcal{E}}$. This map is a well-defined function because another monopole operator $M_{\widetilde{\phi}'}(\mathcal{S})$ associated to co-character having the same obstruction $\alpha_{\widetilde{\phi}'} = \alpha_{\widetilde{\phi}}$ leads to the same charge $\beta_{\widetilde{\phi}'} = \beta_{\widetilde{\phi}}$. This can be argued using an OPE argument as the one employed around equation (191). Moreover, $\gamma$ is a group homomorphism, which can also be argued using an OPE argument. The homomorphism $\gamma$ determines the 2-group anomaly as in section 5.4.

**1-form anomaly.** If we do not turn on 0-form background and only turn on 1-form background, then we observe an anomaly for the 1-form symmetry of the form (97) that can be obtained from the information of the above 2-group anomaly. This 1-form anomaly is associated to a homomorphism

$$\gamma' : \Gamma^{(1)} \to \widehat{\Gamma}^{(1)}, \tag{242}$$

which can obtained from the homomorphism $\gamma$ associated to the 2-group anomaly by composing $\gamma$ with the maps $\Gamma^{(1)} \to \mathcal{E}$ and $\widehat{\mathcal{E}} \to \widehat{\Gamma}^{(1)}$. This 1-form anomaly is the same as obtained in section 6.2 above.

**Anomalies when 2-group symmetry is trivial.** Consider the situation where the short exact sequence (26) splits, leading to a trivial 2-group symmetry. In such a situation, as we describe below, the 2-group anomaly splits into a 1-form anomaly, a 0-form anomaly and a mixed 1-form 0-form anomaly.

Splitting of the short exact sequence means that we have an isomorphism

$$\Gamma^{(1)} \times \mathcal{Z} \to \mathcal{E}, \tag{243}$$

which induces a Pontryagin dual isomorphism

$$\widehat{\mathcal{E}} \to \widehat{\Gamma}^{(1)} \times \widehat{\mathcal{Z}}. \tag{244}$$

Combining these isomorphisms with the data of $\gamma$, we obtain a map

$$\gamma_2 : \Gamma^{(1)} \times \mathcal{Z} \to \widehat{\Gamma}^{(1)} \times \widehat{\mathcal{Z}}. \tag{245}$$

Restricting the domain of $\gamma_2$ to $\Gamma^{(1)}$ and using projections $\widehat{\Gamma}^{(1)} \times \widehat{\mathcal{Z}} \to \widehat{\Gamma}^{(1)}$ and $\widehat{\Gamma}^{(1)} \times \widehat{\mathcal{Z}} \to \widehat{\mathcal{Z}}$, we obtain homomorphisms

$$\gamma_1 : \Gamma^{(1)} \to \widehat{\Gamma}^{(1)}, \tag{246}$$

and

$$\gamma_{10} \colon \ \Gamma^{(1)} \to \widehat{\mathcal{Z}}, \tag{247}$$

capturing the pure 1-form anomaly (97) and the mixed 1-form/0-form anomaly (121) respectively. On the other hand, restricting the domain of $\gamma_2$ to $\mathcal{Z}$ and using projection $\widehat{\Gamma}^{(1)} \times \widehat{\mathcal{Z}} \to \widehat{\mathcal{Z}}$, we obtain the homomorphism

$$\gamma_0 \colon \ \mathcal{Z} \to \widehat{\mathcal{Z}}, \tag{248}$$

capturing the pure 0-form anomaly (145).

# 7 Examples

Let us discuss some examples to illustrate the general discussions of the previous section.

## 7.1 QED with charge 1 fermions

**Symmetries.**   Consider a 3d gauge theory

$$\mathcal{G} = U(1)_g, \qquad k, \qquad N_f \times \text{ fermions of } q_g = 1, \tag{249}$$

with gauge group $\mathcal{G} = U(1)_g$, Chern-Simons level $k$ and $N_f$ fermions, all of gauge charge $q_g = 1$. The 1-form symmetry group $\Gamma^{(1)}$ is clearly trivial as the fermions screen all the gauge Wilson line defects.

There is a non-topological 0-form symmetry

$$\mathfrak{f}_{nt} = \mathfrak{su}(N_f), \tag{250}$$

rotating the $N_f$ fermions, and a topological 0-form symmetry

$$\mathfrak{f}_t = \mathfrak{u}(1)_t. \tag{251}$$

We choose a central extension

$$F = F_t \times F_{nt} = U(1)_t \times SU(N_f), \tag{252}$$

of the 0-form symmetry group $\mathcal{F}$.

Consider the non-fractional gauge monopole operator $M_\phi(\mathcal{G})$ defined by the co-character $\phi$ of winding number 1 along $\mathcal{G} = U(1)_g$. The contribution of gauge CS level $k$ to its gauge charge is

$$q_{g,\phi}^{(k)} = -k. \tag{253}$$

The contribution of a fermion $\psi$ to its gauge charge is

$$q_{g,\phi}^{(\psi)} = -\frac{1}{2}, \tag{254}$$

as the gauge charge $q_g(\psi)$ of the fermion is 1 and its charge $q_\phi(\psi)$ is also 1. Thus, the total gauge charge of the monopole operator $M_\phi(\mathcal{G})$ is

$$q_{g,\phi} = -k - \frac{N_f}{2}. \tag{255}$$

The quantization condition for the gauge CS level $k$ is that it is half-integral if $N_f$ is odd, and it is integral if $N_f$ is even.

The topological charge of the monopole operator $M_\phi(\mathcal{G})$ is

$$q_{t,\phi} = 1\,, \tag{256}$$

under $F_t = U(1)_t$. There is no Chern-Simons contribution to its charge under the Cartan $\prod_{\alpha=1}^{N_f-1} U(1)_\alpha$ of the non-topological 0-form symmetry $F_{nt} = SU(N_f)$. Thus, the charge $q_{\alpha,\phi}$ under $U(1)_\alpha$ of $M_\phi(\mathcal{G})$ is given purely by the fermion contribution, which is

$$q_{\alpha,\phi} = -\frac{1}{2}\sum_{\rho=1}^{N_f} q_\alpha(\rho) = 0\,, \tag{257}$$

where $\rho$ are the weights for the fundamental representation of $\mathfrak{su}(N_f)$ and $q_\alpha(\rho)$ is the charge of $\rho$ under $U(1)_\alpha$. The sum of $q_\alpha(\rho)$ over $\rho$ is zero. Thus, the charge of $M_\phi(\mathcal{G})$ under the center $Z_f = \mathbb{Z}_{N_f}$ of $SU(N_f)$ is

$$q_{f,\phi} = 0\,. \tag{258}$$

From this we can now compute the 0-form symmetry group and the structure group of the gauge theory. Let us reparametrize CS level as

$$k = -\frac{N_f}{2} - mN_f - n\,, \tag{259}$$

where $m$ can be an arbitrary integer and $n$ is an integer such that $0 \le n < N_f$. Then the 0-form symmetry group can be written as

$$\mathcal{F} = \frac{SU(N_f) \times U(1)_t}{\mathbb{Z}_{N_f}}\,, \tag{260}$$

where

$$\mathcal{Z} = \mathbb{Z}_{N_f}\,, \tag{261}$$

in the denominator is generated by the element

$$\left(e^{\frac{2\pi i}{N_f}}, e^{\frac{-2\pi in}{N_f}}\right) \in Z_f \times U(1)_t\,. \tag{262}$$

A special case is $n = 0$, for which we can write $\mathcal{F}$ as

$$\mathcal{F} = PSU(N_f) \times U(1)_t\,. \tag{263}$$

The structure group, on the other hand, turns out to be

$$\mathcal{S} = \frac{U(1)_g \times SU(N_f) \times U(1)_t}{\mathbb{Z}_{N_f}}\,, \tag{264}$$

where

$$\mathcal{E} = \mathbb{Z}_{N_f}\,, \tag{265}$$

in the denominator is generated by the element

$$\left(e^{\frac{-2\pi i}{N_f}}, e^{\frac{2\pi i}{N_f}}, e^{\frac{-2\pi in}{N_f}}\right) \in U(1)_g \times Z_f \times U(1)_t\,. \tag{266}$$

For the special case $n = 0$, we can write $\mathcal{S}$ as

$$\mathcal{S} = \frac{U(1)_g \times SU(N_f)}{\mathbb{Z}_{N_f}} \times U(1)_t\,. \tag{267}$$

To determine (266) it is useful to recall the computation approach outlined in [50], where from the charges one can determine the generators using a Smith normal form decomposition. Note that in the following this computation might have to be repeated, by replacing some of the groups with their covers (e.g. $U(1)_t$ will be lifted to a cover $\widetilde{U(1)}_t$ as otherwise there are fractionally charged states).

**Anomalies for $n = 0$.** Let us discuss the computation of anomalies for two special cases $n = 0$ and $n = 1$, beginning with the case $n = 0$. Let $k_f$ be the CS level for $SU(N_f)$.

Consider the mixed monopole operator $M_{\widetilde{\phi}}(\mathcal{S})$ with $\widetilde{\phi}$ having winding

$$\widetilde{\phi}_g = -\frac{1}{N_f}, \tag{268}$$

along $U(1)_g$ and windings

$$\widetilde{\phi}_\alpha = \frac{\alpha}{N_f}, \tag{269}$$

along the component $U(1)_\alpha$ in the maximal torus $\prod_{\alpha=1}^{N_f - 1} U(1)_\alpha$ of $SU(N_f)$. This monopole has the obstruction $1 \in \mathbb{Z}_{N_f} = \mathcal{E}$ to being regarded as a monopole for $U(1)_g \times SU(N_f)$.

The topological charge of this monopole operator $M_{\widetilde{\phi}}(\mathcal{S})$ is

$$q_{t,\widetilde{\phi}} = -\frac{1}{N_f}, \tag{270}$$

under $U(1)_t$. Thus we choose an $N_f$-fold cover $\widetilde{U(1)}_t$ of $U(1)_t$ to be the new $F_t$. This modifies $F$ to

$$F = SU(N_f) \times \widetilde{U(1)}_t, \tag{271}$$

thus modifying $\mathcal{Z}$ to

$$\mathcal{Z} = \mathbb{Z}_{N_f}^{(nt)} \times \mathbb{Z}_{N_f}^{(t)}, \tag{272}$$

such that the 0-form symmetry group is expressed as

$$\mathcal{F} = PSU(N_f) \times U(1)_t = \frac{SU(N_f)}{\mathbb{Z}_{N_f}^{(nt)}} \times \frac{\widetilde{U(1)}_t}{\mathbb{Z}_{N_f}^{(t)}}. \tag{273}$$

Correspondingly, $\mathcal{S}$ is re-expressed as

$$\mathcal{S} = \frac{U(1)_g \times SU(N_f)}{\mathbb{Z}_{N_f}^{(nt,\mathcal{E})}} \times \frac{\widetilde{U(1)}_t}{\mathbb{Z}_{N_f}^{(t)}}, \tag{274}$$

with the modified $\mathcal{E}$ being

$$\mathcal{E} = \mathbb{Z}_{N_f}^{(nt,\mathcal{E})} \times \mathbb{Z}_{N_f}^{(t)}, \tag{275}$$

where

$$\left( e^{\frac{-2\pi i}{N_f}}, e^{\frac{2\pi i}{N_f}} \right) \in U(1)_g \times Z_f, \tag{276}$$

is the generator of $\mathbb{Z}_{N_f}^{(nt,\mathcal{E})}$. The obstruction associated to $M_{\widetilde{\phi}}(\mathcal{S})$ is now $1 \in \mathbb{Z}_{N_f}^{(nt,\mathcal{E})} \subset \mathcal{E}$ and

$$\widetilde{q}_{t,\widetilde{\phi}} = -1, \tag{277}$$

is its topological charge under $\widetilde{U(1)}_t$.

The gauge charge of $M_{\widetilde{\phi}}(\mathcal{S})$ receives a contribution

$$q_{g,\widetilde{\phi}}^{(k)} = \frac{k}{N_f} = -\frac{1}{2} - m, \tag{278}$$

from CS level $k$ and a contribution

$$\sum_{\psi} q_{g,\widetilde{\phi}}^{(\psi)} = -\frac{1}{2}, \tag{279}$$

from the fermions. To see the fermionic contribution, note that a fermion of charge 1 under $U(1)_g$ and transforming in a weight having Dynkin coefficients $(\rho_1, \cdots, \rho_{N_f-1})$ of $SU(N_f)$ has charge

$$-\frac{1}{N_f} + \sum_{\alpha=1}^{N_f-1} \rho_\alpha \frac{\alpha}{N_f}, \tag{280}$$

under $U(1)_{\widetilde{\phi}}$. Thus, the only weight that contributes from the fundamental representation of $SU(N_f)$ is $(0, \cdots, 0, -1)$, and its contribution is $-\frac{1}{2}$ as claimed above. Consequently,

$$q_{g,\widetilde{\phi}} = -m - 1, \tag{281}$$

is the total gauge charge of $M_{\widetilde{\phi}}(\mathcal{S})$.

Now let us compute $\prod_{\alpha=1}^{N_f-1} U(1)_\alpha$ charges of $M_{\widetilde{\phi}}(\mathcal{S})$. A root having Dynkin coefficients $(\rho_1, \cdots, \rho_{N_f-1})$ has charge

$$\sum_{\alpha=1}^{N_f-1} \rho_\alpha \frac{\alpha}{N_f}, \tag{282}$$

under $U(1)_{\widetilde{\phi}}$. Thus, the CS level $k_f$ contributes charge

$$(0, \cdots, 0, -k_f), \tag{283}$$

under $\prod_{\alpha=1}^{N_f-1} U(1)_\alpha$. From our above discussion, we can easily compute

$$\left(0, \cdots, 0, -\frac{1}{2}\right), \tag{284}$$

to be the fermion contribution. Thus, the Chern-Simons level $k_f$ should be quantized as

$$k_f = \frac{2l+1}{2}, \tag{285}$$

with $l \in \mathbb{Z}$, in order to avoid global[13] ABJ anomaly for $SU(N_f)$. This implies that

$$q_{f,\widetilde{\phi}} = k_f + \frac{1}{2} \ (\mathrm{mod} \ N_f), \tag{286}$$

is the $Z_f$ charge of $M_{\widetilde{\phi}}(\mathcal{S})$.

The charge

$$\left(q_{g,\widetilde{\phi}}, \widetilde{q}_{t,\widetilde{\phi}}, q_{f,\widetilde{\phi}}\right) = \left(-m-1, -1, k_f + \frac{1}{2} \ (\mathrm{mod} \ N_f)\right), \tag{287}$$

of $M_{\widetilde{\phi}}(\mathcal{S})$ under $U(1)_g \times \widetilde{U(1)}_t \times Z_f$ is equivalent to the charge

$$\left(q_{g,\widetilde{\phi}}, \widetilde{q}_{t,\widetilde{\phi}}, q_{f,\widetilde{\phi}}\right) = \left(0, -1, m+1+k_f + \frac{1}{2} \ (\mathrm{mod} \ N_f)\right), \tag{288}$$

---

[13]Here the word 'global' refers to the traditional usage of the term: A global anomaly is one that cannot be captured in terms of an anomaly polynomial.

as charges in $\widehat{\mathcal{E}}$. Let us reparametrize $k_f$ as

$$k_f = -m - \frac{3}{2} + m'N_f + n', \tag{289}$$

where $m' \in \mathbb{Z}$ and $0 \le n' < N_f$. Then, the above charge is

$$\left(q_{g,\widetilde{\phi}}, \widetilde{q}_{t,\widetilde{\phi}}, q_{f,\widetilde{\phi}}\right) = \left(0, -1, n' \ (\mathrm{mod} \ N_f)\right), \tag{290}$$

This fixes the homomorphisms determining the anomalies. The first homomorphism is

$$\gamma_{nt,t}: \ \mathbb{Z}_{N_f}^{(nt)} \to \widehat{\mathbb{Z}}_{N_f}^{(t)}, \tag{291}$$

mapping $1 \in \mathbb{Z}_{N_f}^{(nt)}$ to $-1 \in \widehat{\mathbb{Z}}_{N_f}^{(t)}$. Concretely, the corresponding anomaly is

$$\exp\left(-\frac{2\pi i}{N_f} \int w_2^f \cup c_1^t \ (\mathrm{mod} \ N_f)\right), \tag{292}$$

where $w_2^f$ is the characteristic class capturing the obstruction of lifting $\mathcal{F}_{nt} = PSU(N_f)$ bundles to $F_{nt} = SU(N_f)$ bundles, and $c_1^t$ is the first Chern class for $U(1)_t$ bundles. Consequently, $c_1^t \ (\mathrm{mod} \ N_f)$ is the obstruction class for lifting $\mathcal{F}_t = U(1)_t$ bundles to $F_t = \widetilde{U(1)}_t$ bundles.

The second homomorphism is

$$\gamma_{nt,nt}: \ \mathbb{Z}_{N_f}^{(nt)} \to \widehat{\mathbb{Z}}_{N_f}^{(nt)}, \tag{293}$$

mapping $1 \in \mathbb{Z}_{N_f}^{(nt)}$ to $n' \in \widehat{\mathbb{Z}}_{N_f}^{(t)}$. Concretely, for odd $N_f$, the corresponding anomaly is

$$\exp\left(\frac{2\pi i n'}{N_f} \int w_2^f \cup w_2^f\right). \tag{294}$$

For even $N_f$, the corresponding anomaly is

$$\exp\left(\frac{\pi i n'}{N_f} \int \mathcal{P}\left(w_2^f\right)\right), \tag{295}$$

where $\mathcal{P}\left(w_2^f\right)$ is obtained by applying the standard Pontryagin square operation

$$\mathcal{P}: \ H^2\left(M_3, \mathbb{Z}_{N_f}\right) \to H^4\left(M_3, \mathbb{Z}_{2N_f}\right). \tag{296}$$

We can verify the anomaly (292) by considering the monopole operator $M_{\widetilde{\phi}}(\mathcal{S})$ with $U(1)_{\widetilde{\phi}}$ having winding 1 along $U(1)_t$. Such a monopole has only a gauge charge given by

$$q_{g,\widetilde{\phi}} = 1. \tag{297}$$

Thus, this monopole has charge

$$\left(q_{g,\widetilde{\phi}}, \widetilde{q}_{t,\widetilde{\phi}}, q_{f,\widetilde{\phi}}\right) = (1, 0, 0), \tag{298}$$

under $U(1)_g \times U(1)_t \times Z_f$. This charge is equivalent to the charge

$$\left(q_{g,\widetilde{\phi}}, \widetilde{q}_{t,\widetilde{\phi}}, q_{f,\widetilde{\phi}}\right) = (0, -1, 0), \tag{299}$$

as a charge in $\widehat{\mathcal{E}}$. From this, we can read the homomorphism

$$\gamma_{t,nt}: \ \mathbb{Z}_{N_f}^{(t)} \to \widehat{\mathbb{Z}}_{N_f}^{(nt)}, \tag{300}$$

mapping $1 \in \mathbb{Z}_{N_f}^{(t)}$ to $-1 \in \widehat{\mathbb{Z}}_{N_f}^{(nt)}$. This homomorphism leads to the same anomaly (292).

**Anomalies for $n = 1$.** Let us now discuss the case $n = 1$. Consider the mixed monopole operator $M_{\widetilde{\phi}}(\mathcal{S})$ with $\widetilde{\phi}$ having winding under the following $U(1)$s

$$
\begin{aligned}
U(1)_g : \quad & \widetilde{\phi}_g = -\frac{1}{N_f}, \\
U(1)_\alpha : \quad & \widetilde{\phi}_\alpha = \frac{\alpha}{N_f}, \\
U(1)_t : \quad & \phi_t = \frac{1}{N_f}.
\end{aligned}
\tag{301}
$$

This monopole has the obstruction $1 \in \mathbb{Z}_{N_f} = \mathcal{E}$ to being regarded as a monopole for $U(1)_g \times SU(N_f) \times U(1)_t$.

The topological charge of this monopole operator $M_{\widetilde{\phi}}(\mathcal{S})$ is

$$
q_{t,\widetilde{\phi}} = -\frac{1}{N_f},
\tag{302}
$$

under $U(1)_t$. Thus we choose an $N_f$-fold cover $\widetilde{U(1)}_t$ of $U(1)_t$ to be the new $F_t$. This modifies $F$ to

$$
F = SU(N_f) \times \widetilde{U(1)}_t,
\tag{303}
$$

thus modifying $\mathcal{Z}$ to

$$
\mathcal{Z} = \mathbb{Z}_{N_f^2},
\tag{304}
$$

such that the 0-form symmetry group is expressed as

$$
\mathcal{F} = \frac{SU(N_f) \times U(1)_t}{\mathbb{Z}_{N_f}} = \frac{SU(N_f) \times \widetilde{U(1)}_t}{\mathbb{Z}_{N_f^2}}.
\tag{305}
$$

Correspondingly, $\mathcal{S}$ is re-expressed as

$$
\mathcal{S} = \frac{U(1)_g \times SU(N_f) \times \widetilde{U(1)}_t}{\mathbb{Z}_{N_f^2}},
\tag{306}
$$

with the modified $\mathcal{E}$ being

$$
\mathcal{E} = \mathbb{Z}_{N_f^2},
\tag{307}
$$

where

$$
\left( e^{\frac{-2\pi i}{N_f}}, e^{\frac{2\pi i}{N_f}}, e^{\frac{-2\pi i}{N_f^2}} \right) \in U(1)_g \times Z_f \times \widetilde{U(1)}_t,
\tag{308}
$$

is the generator of $\mathcal{E}$. The obstruction associated to $M_{\widetilde{\phi}}(\mathcal{S})$ is now $1 \in \mathbb{Z}_{N_f^2}$ for regarding it as a monopole for $U(1)_g \times SU(N_f) \times \widetilde{U(1)}_t$, and

$$
\widetilde{q}_{t,\widetilde{\phi}} = -1,
\tag{309}
$$

is its topological charge under $\widetilde{U(1)}_t$.

The gauge charge of $M_{\widetilde{\phi}}(\mathcal{S})$ receives a contribution

$$
q_{g,\widetilde{\phi}}^{(k)} = \frac{k}{N_f} = -\frac{1}{2} - m - \frac{1}{N_f},
\tag{310}
$$

from CS level $k$, a contribution

$$\sum_\psi q^{(\psi)}_{g,\widetilde{\phi}} = -\frac{1}{2}, \tag{311}$$

from the fermions, and a contribution

$$q'_{g,\widetilde{\phi}} = \frac{1}{N_f}, \tag{312}$$

from the fact that $U(1)_{\widetilde{\phi}}$ has a winding number $\frac{1}{N_f}$ along the topological symmetry $U(1)_t$ associated to the gauge group $U(1)_g$. Consequently,

$$q_{g,\widetilde{\phi}} = -m-1, \tag{313}$$

is the total gauge charge of $M_{\widetilde{\phi}}(\mathcal{S})$.

The CS level $k_f$ contributes charge

$$(0,\cdots,0,-k_f), \tag{314}$$

under $\prod_{\alpha=1}^{N_f-1} U(1)_\alpha$ and

$$\left(0,\cdots,0,-\frac{1}{2}\right), \tag{315}$$

is the fermion contribution. Thus, the Chern-Simons level $k_f$ should again be quantized as

$$k_f = \frac{2l+1}{2}, \tag{316}$$

with $l \in \mathbb{Z}$, in order to avoid global ABJ anomaly for $SU(N_f)$. This implies that

$$q_{f,\widetilde{\phi}} = k_f + \frac{1}{2} \ (\mathrm{mod}\ N_f), \tag{317}$$

is the $Z_f$ charge of $M_{\widetilde{\phi}}(\mathcal{S})$.

Thus the total charge of $M_{\widetilde{\phi}}(\mathcal{S})$ under $U(1)_g \times \widetilde{U(1)}_t \times Z_f$ is

$$\left(q_{g,\widetilde{\phi}}, \widetilde{q}_{t,\widetilde{\phi}}, q_{f,\widetilde{\phi}}\right) = \left(-m-1, -1, k_f + \frac{1}{2} \ (\mathrm{mod}\ N_f)\right), \tag{318}$$

which is equivalent to the charge

$$\left(q_{g,\widetilde{\phi}}, \widetilde{q}_{t,\widetilde{\phi}}, q_{f,\widetilde{\phi}}\right) = \left(0, -1, n' \ (\mathrm{mod}\ N_f)\right), \tag{319}$$

as charges in $\widehat{\mathcal{E}}$, where $n'$ is defined in terms of $k_f$ via

$$k_f = -m - \frac{3}{2} + m'N_f + n', \tag{320}$$

where $m' \in \mathbb{Z}$ and $0 \le n' < N_f$.

This leads to the anomaly

$$\exp\left(\pi i \frac{n'N_f - n}{N_f^2} \int \mathcal{P}(w_2)\right), \tag{321}$$

for even $N_f$, where $w_2$ is the $\mathbb{Z}_{N_f^2}$ valued characteristic class capturing the obstruction of lifting

$$\mathcal{F} = \frac{SU(N_f) \times \widetilde{U(1)}_t}{\mathbb{Z}_{N_f^2}} \,, \tag{322}$$

bundles to

$$F = F_{nt} \times F_t = SU(N_f) \times \widetilde{U(1)}_t \,, \tag{323}$$

bundles. For odd $N_f$, the anomaly is

$$\exp\left( 2\pi i \, \frac{n'N_f - n}{N_f^2} \int w_2 \cup w_2 \right). \tag{324}$$

## 7.2 $\mathcal{N} = 4$ SQED with charge 1 hypers

Consider 3d $\mathcal{N} = 4$ gauge theory with gauge group

$$\mathcal{G} = U(1)_g \,, \tag{325}$$

and $N_f$ hypermultiplets of charge 1 under $U(1)_g$. The gauge and flavor Chern-Simons levels must be zero

$$k = k_f = 0 \,, \tag{326}$$

for compatibility with $\mathcal{N} = 4$ supersymmetry. We again consider 0-form symmetry

$$\mathfrak{f} = \mathfrak{f}_t \oplus \mathfrak{f}_{nt} = \mathfrak{u}(1)_t \oplus \mathfrak{su}(N_f) \,. \tag{327}$$

We have a total of $2N_f$ fermions, $N_f$ of which have gauge charge 1 and transform in fundamental representation of $SU(N_f)$, and the other $N_f$ have gauge charge $-1$ and transform in anti-fundamental representation of $SU(N_f)$. The fermion contributions cancel while computing $U(1)_g$ and $Z_f$ charges of fermions.

A non-fractional gauge monopole $M_\phi(\mathcal{G})$ with $U(1)_\phi = U(1)_g$ has charges

$$\left( q_{g,\phi}, q_{t,\phi}, q_{f,\phi} \right) = \left( 0, 1, 0 \ (\mathrm{mod} \ N_f) \right), \tag{328}$$

under $U(1)_g \times U(1)_t \times Z_f$. This implies the structure group is

$$\mathcal{S} = \frac{U(1)_g \times SU(N_f)}{\mathbb{Z}_{N_f}} \times U(1)_t \,, \tag{329}$$

and 0-form symmetry group is

$$\mathcal{F} = PSU(N_f) \times U(1)_t \,. \tag{330}$$

Consider now a mixed gauge/0-form monopole operator $M_{\widetilde{\phi}}(\mathcal{S})$ with $U(1)_{\widetilde{\phi}}$ winding

$$\widetilde{\phi}_g = -\frac{1}{N_f} \,, \tag{331}$$

along $U(1)_g$ and windings

$$\widetilde{\phi}_\alpha = \frac{\alpha}{N_f} \,, \tag{332}$$

along the component $U(1)_\alpha$ in the maximal torus $\prod_{\alpha=1}^{N_f-1} U(1)_\alpha$ of $SU(N_f)$. This monopole has the obstruction $1 \in \mathbb{Z}_{N_f}$ to being regarded as a monopole for $U(1)_g \times SU(N_f)$. This monopole has charges

$$\left(q_{g,\widetilde{\phi}}, q_{t,\widetilde{\phi}}, q_{f,\widetilde{\phi}}\right) = \left(0, -\frac{1}{N_f}, 0 \;(\mathrm{mod}\; N_f)\right), \tag{333}$$

under $U(1)_g \times U(1)_t \times \mathbb{Z}_{N_f}$. This implies that there is no pure anomaly for $PSU(N_f)$ part of 0-form symmetry group $\mathcal{F}$, but there is a mixed anomaly between $PSU(N_f)$ and $U(1)_t$ 0-form symmetries given by

$$\exp\left(-\frac{2\pi i}{N_f} \int w_2^f \cup c_1^t \;(\mathrm{mod}\; N_f)\right), \tag{334}$$

where $w_2^f$ is the characteristic class capturing the obstruction of lifting $PSU(N_f)$ bundles to $SU(N_f)$ bundles, and $c_1^t$ is the first Chern class for $U(1)_t$ bundles.

## 7.3 QED with 2 fermions of charge 2

Consider a 3d gauge theory with gauge group

$$\mathcal{G} = U(1)_g, \tag{335}$$

with Chern-Simons level $k$ and 2 fermions, both having gauge charge 2.

**1-form symmetry.** Let us determine the 1-form symmetry. Operators constructed out of the charge 2 fermions screen Wilson lines of even charges. So we need to determine whether non-fractional gauge monopole operators can screen the odd charge Wilson lines, following (175). Consider the non-fractional gauge monopole $M_\phi(\mathcal{G})$ with $U(1)_\phi$ having winding number 1 along $U(1)_g$. The gauge charge of this monopole operator is

$$q_{g,\phi} = -k-4. \tag{336}$$

Thus if $k$ is even, then no new screenings are introduced and we have a non-trivial 1-form symmetry

$$k \text{ even}: \qquad \Gamma^{(1)} = \mathbb{Z}_2. \tag{337}$$

On the other hand, if $k$ is odd, then odd charge Wilson lines are also screened, and we have a trivial 1-form symmetry

$$k \text{ odd}: \qquad \Gamma^{(1)} = 0. \tag{338}$$

**2-group and 0-form symmetries.** 0-form symmetry algebra is

$$\mathfrak{f} = \mathfrak{f}_t \oplus \mathfrak{f}_{nt} = \mathfrak{u}(1)_t \oplus \mathfrak{su}(2)_f, \tag{339}$$

where $\mathfrak{su}(2)_f$ rotates the two fermions. The monopole operator $M_\phi(\mathcal{G})$ has charge

$$q_t = 1, \tag{340}$$

under the topological symmetry $U(1)_t$ and a charge 0 under the Cartan of $SU(2)_f$.

This information allows us to compute the structure group to be

$$\mathcal{S} = \frac{U(1)_g \times SU(2)_f \times U(1)_t}{\mathbb{Z}_4}, \tag{341}$$

where $\mathbb{Z}_4$ is generated by the element

$$\left(e^{\frac{\pi i}{2}}, e^{\pi i}, e^{\frac{\pi i k}{2}}\right) \in U(1)_g \times Z_f \times U(1)_t, \tag{342}$$

where $Z_f = \mathbb{Z}_2$ is the center of $SU(2)_f$. Thus, we have

$$\mathcal{E} = \mathbb{Z}_4. \tag{343}$$

Computing the 0-form symmetry group from the above structure group, we find it to be

$$\mathcal{F} = SO(3)_f \times U(1)_t, \tag{344}$$

for $k = 0 \pmod 4$;

$$\mathcal{F} = \frac{SU(2)_f \times U(1)_t}{\mathbb{Z}_4}, \tag{345}$$

for $k = 1, 3 \pmod 4$, where the element

$$\left(e^{\pi i}, e^{\frac{\pi i}{2}}\right) \in Z_f \times U(1)_t, \tag{346}$$

is the generator of $\mathbb{Z}_4$; and

$$\mathcal{F} = \frac{SU(2)_f \times U(1)_t}{\mathbb{Z}_2} \simeq U(2), \tag{347}$$

for $k = 2 \pmod 4$, where the element

$$\left(e^{\pi i}, e^{\pi i}\right) \in Z_f \times U(1)_t, \tag{348}$$

is the generator of $\mathbb{Z}_2$.

For even $k$, the non-trivial $\mathbb{Z}_2$ 1-form symmetry combines with the above 0-form symmetry group to form a 2-group symmetry associated to the short exact sequence

$$0 \to \mathbb{Z}_2 \to \mathbb{Z}_4 \to \mathbb{Z}_2 \to 0. \tag{349}$$

For odd $k$, there is no 2-group symmetry as there is no 1-form symmetry.

**Anomalies.** Let us compute anomalies for cases in which the CS level is a multiple of 4

$$k = 4m. \tag{350}$$

Let $k_f$ be background CS level for $SU(2)_f$.

Consider the mixed monopole operator $M_{\widetilde{\phi}}(\mathcal{S})$ with $\widetilde{\phi}$ having winding

$$\widetilde{\phi}_g = \frac{1}{4}, \tag{351}$$

along $U(1)_g$ and winding

$$\widetilde{\phi}_f = \frac{1}{2}, \tag{352}$$

along the maximal torus $U(1)_f$ of $SU(2)_f$. This monopole has the obstruction $1 \in \mathbb{Z}_4 = \mathcal{E}$ to being regarded as a monopole for $U(1)_g \times SU(2)_f \times U(1)_t$.

The topological charge of this monopole operator $M_{\widetilde{\phi}}(\mathcal{S})$ is

$$q_{t,\widetilde{\phi}} = \frac{1}{4}, \tag{353}$$

under $U(1)_t$. Thus we choose a 4-fold cover $\widetilde{U(1)}_t$ of $U(1)_t$. This modifies $F$ to

$$F = SU(2)_f \times \widetilde{U(1)}_t \,, \tag{354}$$

thus modifying $\mathcal{Z}$ to

$$\mathcal{Z} = \mathbb{Z}_4^{(nt)} \times \mathbb{Z}_4^{(t)} \,, \tag{355}$$

such that the 0-form symmetry group is expressed as

$$\mathcal{F} = SO(3)_f \times U(1)_t = \frac{SU(2)_f}{\mathbb{Z}_4^{(nt)}} \times \frac{\widetilde{U(1)}_t}{\mathbb{Z}_4^{(t)}} \,. \tag{356}$$

Correspondingly, $\mathcal{S}$ is re-expressed as

$$\mathcal{S} = \frac{U(1)_g \times SU(2)_f}{\mathbb{Z}_4^{(nt,\mathcal{E})}} \times \frac{\widetilde{U(1)}_t}{\mathbb{Z}_4^{(t)}} \,, \tag{357}$$

with the modified $\mathcal{E}$ being

$$\mathcal{E} = \mathbb{Z}_4^{(nt,\mathcal{E})} \times \mathbb{Z}_4^{(t)} \,, \tag{358}$$

where

$$\left( e^{\frac{\pi i}{2}}, e^{\pi i} \right) \in U(1)_g \times Z_f \,, \tag{359}$$

is the generator of $\mathbb{Z}_4^{(nt,\mathcal{E})}$. The obstruction associated to $M_{\widetilde{\phi}}(\mathcal{S})$ is now $1 \in \mathbb{Z}_4^{(nt,\mathcal{E})} \subset \mathcal{E}$ and

$$\widetilde{q}_{t,\widetilde{\phi}} = 1 \,, \tag{360}$$

is its topological charge under $\widetilde{U(1)}_t$.

The charges of $M_{\widetilde{\phi}}(\mathcal{S})$ can be computed to be

$$\left( q_{g,\widetilde{\phi}}, \widetilde{q}_{t,\widetilde{\phi}}, q_{f,\widetilde{\phi}} \right) = \left( -m - 1, 1, k_f + \frac{1}{2} \ (\mathrm{mod}\ 2) \right) \,, \tag{361}$$

under $U(1)_g \times \widetilde{U(1)}_t \times Z_f$. The CS level $k_f$ is taken to be a half-integer that is not an integer, in order to avoid global ABJ anomaly for $\mathfrak{f}_{nt} = \mathfrak{su}(2)_f$. This monopole operator has charges

$$\left( -m - 2 - 2k_f \ (\mathrm{mod}\ 4), 1 \ (\mathrm{mod}\ 4) \right) \,, \tag{362}$$

under $\mathbb{Z}_4^{(nt,\mathcal{E})} \times \mathbb{Z}_4^{(t)}$.

From the above charges, we observe that the theory has an anomaly

$$\exp\left( \frac{\pi i}{2} \int B_w \cup c_1^t \ (\mathrm{mod}\ 4) - (m + 2 + 2k_f) \frac{\mathcal{P}(B_w)}{2} \right) \,, \tag{363}$$

where $c_1^t$ is the first Chern class for $U(1)_t$ and $B_w$ is the $\mathbb{Z}_4$ valued 2-cocycle acting as the background field for the 2-group symmetry involving the $\mathbb{Z}_2$ 1-form symmetry and the $SO(3)_f$ 0-form symmetry. $B_w$ can be expressed as

$$B_w = 2B_2 + \widetilde{w}_2^f \,, \tag{364}$$

where $B_2$ is the background for the $\mathbb{Z}_2$ 1-form symmetry background and $\tilde{w}_2^f$ is a $\mathbb{Z}_4$ valued 2-cochain which is a lift of the $\mathbb{Z}_2$ valued 2-cocycle $w_2^f$ capturing the obstruction of lifting $SO(3)_f$ bundles to $SU(2)_f$ bundles.

## 7.4   $\mathcal{N} = 4$ SQED with 2 hypers of charge 2

Consider 3d $\mathcal{N} = 4$ gauge theory with gauge group

$$\mathcal{G} = U(1)_g \,, \tag{365}$$

and 2 hypermultiplets of charge 2 under $U(1)_g$. The gauge and flavor Chern-Simons levels must be zero

$$k = k_f = 0 \,, \tag{366}$$

for compatibility with $\mathcal{N} = 4$ supersymmetry. We again consider 0-form symmetry

$$\mathfrak{f} = \mathfrak{f}_t \oplus \mathfrak{f}_{nt} = \mathfrak{u}(1)_t \oplus \mathfrak{su}(2)_f \,. \tag{367}$$

We have a total of 4 fermions: a doublet of $SU(2)_f$ with gauge charge 2 and a doublet of $SU(2)_f$ with gauge charge -2. The fermion contributions cancel while computing $U(1)_g$ and $Z_f$ charges of fermions.

A non-fractional gauge monopole $M_\phi(\mathcal{G})$ with $U(1)_\phi = U(1)_g$ has charges

$$\left( q_{g,\phi}, q_{t,\phi}, q_{f,\phi} \right) = (0, 1, 0 \; (\mathrm{mod}\ 2)) \,, \tag{368}$$

under $U(1)_g \times U(1)_t \times Z_f$. This implies the structure group is

$$\mathcal{S} = \frac{U(1)_g \times SU(2)_f}{\mathbb{Z}_4} \times U(1)_t \,, \tag{369}$$

with

$$\left( e^{\frac{\pi i}{2}}, e^{\pi i} \right) \in U(1)_g \times Z_f \,, \tag{370}$$

is the generator of $\mathbb{Z}_4$. The 0-form symmetry group is

$$\mathcal{F} = SO(3)_f \times U(1)_t \,, \tag{371}$$

and 1-form symmetry group is

$$\Gamma^{(1)} = \mathbb{Z}_2 \,. \tag{372}$$

The $SO(3)_f$ 0-form symmetry and $\mathbb{Z}_2$ 1-form symmetry combine to form a 2-group symmetry based on the short exact sequence

$$0 \to \mathbb{Z}_2 \to \mathbb{Z}_4 \to \mathbb{Z}_2 \to 0 \,. \tag{373}$$

Consider now a mixed gauge/0-form monopole operator $M_{\widetilde{\phi}}(\mathcal{S})$ with $U(1)_{\widetilde{\phi}}$ winding

$$\widetilde{\phi}_g = \frac{1}{4} \,, \tag{374}$$

along $U(1)_g$ and winding

$$\widetilde{\phi}_f = \frac{1}{2} \,, \tag{375}$$

along the maximal torus $U(1)_f$ of $SU(2)_f$. This monopole has the obstruction $1 \in \mathbb{Z}_4$ to being regarded as a monopole for $U(1)_g \times SU(2)_f$. This monopole has charges

$$\left( q_{g,\widetilde{\phi}}, q_{t,\widetilde{\phi}}, q_{f,\widetilde{\phi}} \right) = \left( 0, \frac{1}{4}, 0 \; (\mathrm{mod}\ 2) \right) \,, \tag{376}$$

under $U(1)_g \times U(1)_t \times Z_f$. This implies that there is a mixed anomaly between the $U(1)_t$ 0-form symmetry and the 2-group symmetry given by

$$\exp\left( \frac{\pi i}{2} \int B_w \cup c_1^t \; (\mathrm{mod}\ 4) \right) \,, \tag{377}$$

where $B_w$ is the $\mathbb{Z}_4$ valued 2-cocycle associated to the 2-group background and $c_1^t$ is the first Chern class for $U(1)_t$ backgrounds.

## 7.5  $\mathcal{N} = 2$ dual of $\mathcal{N} = 4$ T[SU(2)] theory

T[SU(2)] [106] is a 3d $\mathcal{N} = 4$ SCFT which has a 0-form flavor symmetry

$$\mathfrak{f} = \mathfrak{su}(2)_C \oplus \mathfrak{su}(2)_H , \tag{378}$$

where $\mathfrak{su}(2)_C$ acts on its Coulomb branch of vacua and $\mathfrak{su}(2)_H$ acts on its Higgs branch of vacua. It is well-known that the 0-form flavor symmetry group of T[SU(2)] is

$$\mathcal{F} = SO(3)_C \times SO(3)_H , \tag{379}$$

and there is a mixed 0-form anomaly given by

$$\mathcal{A}_4 = \exp\left( \pi i \int w_2^C \cup w_2^H \right) , \tag{380}$$

where $w_2^{C,H}$ are characteristic classes capturing the obstruction of lifting $SO(3)_{C,H}$ bundles to $SU(2)_{C,H}$ bundles.

It is also known that there is a 3d $\mathcal{N} = 2$ gauge theory that flows to the T[SU(2)] theory, and notably manifests the full 0-form symmetry $\mathfrak{f}$ in the UV. In this subsection, we use our method to recover the 0-form symmetry group $\mathcal{F}$ and the anomaly $\mathcal{A}_4$ from this 3d $\mathcal{N} = 2$ gauge theory. A similar check was also performed in [11].

The 3d $\mathcal{N} = 2$ gauge theory under discussion has gauge group

$$\mathcal{G} = SU(2)_g , \tag{381}$$

with CS level $-1$, with 2 chiral multiplets that transform in tri-fundamental representation of

$$SU(2)_g \times SU(2)_C \times SU(2)_H , \tag{382}$$

where $\mathfrak{su}(2)_C \oplus \mathfrak{su}(2)_H$ is flavor symmetry algebra, and with background CS levels 1 for both $SU(2)_C$ and $SU(2)_H$. It is clear that there is no 1-form symmetry as the operators composed out of these chiral multiplets screen all the gauge Wilson line defects for $SU(2)_g$.

To compute the 0-form symmetry group, we need to consider also non-fractional gauge monopoles alongside the matter content contributions. Consider the non-fractional gauge monopole $M_\phi(\mathcal{G})$ with the co-character $\phi$ having winding number 1 along the maximal torus $U(1)_g \subset SU(2)_g$. The gauge charge under $U(1)_g$ of the monopole operator receives a contribution

$$q_{g,\phi}^{(k)} = 2 , \tag{383}$$

from the gauge CS level, and a contribution

$$\sum_\psi q_{g,\phi}^{(\psi)} = -2 , \tag{384}$$

from the fermions. Thus, the total gauge charge $q_{g,\phi}$ of $M_\phi(\mathcal{G})$ is 0. Moreover, the charges $q_{C,\phi}$ and $q_{H,\phi}$ of $M_\phi(\mathcal{G})$ under Cartan $U(1)_{C,H} \subset SU(2)_{C,H}$ are also 0. Thus monopole operators do not make any non-trivial contribution to the computation of 0-form symmetry group.

The structure group is easily computed to be

$$\mathcal{S} = \frac{SU(2)_g \times SU(2)_C \times SU(2)_H}{\mathbb{Z}_2^{C,\mathcal{E}} \times \mathbb{Z}_2^{H,\mathcal{E}}} , \tag{385}$$

where $\mathbb{Z}_2^{C,\mathcal{E}}$ is generated by the element

$$\left( e^{\pi i}, e^{\pi i}, 1 \right) \in \mathbb{Z}_2^g \times \mathbb{Z}_2^C \times \mathbb{Z}_2^H , \tag{386}$$

where $\mathbb{Z}_2^x$ is the center of the group $SU(2)_x$, and $\mathbb{Z}_2^{H,\mathcal{E}}$ is generated by the element

$$\left(e^{\pi i}, 1, e^{\pi i}\right) \in \mathbb{Z}_2^g \times \mathbb{Z}_2^C \times \mathbb{Z}_2^H. \tag{387}$$

Thus we have

$$\mathcal{E} = \mathbb{Z}_2^{C,\mathcal{E}} \times \mathbb{Z}_2^{H,\mathcal{E}}, \tag{388}$$

and we can compute 0-form symmetry group to be

$$\mathcal{F} = \frac{SU(2)_C \times SU(2)_H}{\mathbb{Z}_2^C \times \mathbb{Z}_2^H} = SO(3)_C \times SO(3)_H, \tag{389}$$

thus matching the 0-form symmetry group for $T[SU(2)]$.

To compute the anomaly, we need to consider mixed gauge/0-form monopole operators. So, consider the monopole $M_{\widetilde{\phi}}(\mathcal{S})$ with the co-character $\widetilde{\phi}$ having winding $\frac{1}{2}$ along $U(1)_g$ and winding $\frac{1}{2}$ along $U(1)_C$. The charge under $U(1)_g$ of this monopole is

$$q_{g,\widetilde{\phi}} = 1, \tag{390}$$

which is entirely from the gauge CS level. Similarly, the charge of $M_{\widetilde{\phi}}(\mathcal{S})$ under $U(1)_C$ is

$$q_{C,\widetilde{\phi}} = -1, \tag{391}$$

which is entirely from the background CS level for $SU(2)_C$. On the other hand, the charge of $M_{\widetilde{\phi}}(\mathcal{S})$ under $U(1)_H$ is

$$q_{H,\widetilde{\phi}} = 0, \tag{392}$$

as both the background CS level for $SU(2)_H$ and the fermions make zero contributions.

Thus, the monopole operator $M_{\widetilde{\phi}}(\mathcal{S})$, which has obstruction $1 \in \mathbb{Z}_2^C$ for being regarded as a monopole for $SU(2)_g \times SU(2)_C \times SU(2)_H$, carries charges

$$\left(0 \ (\mathrm{mod}\ 2), \ 1 \ (\mathrm{mod}\ 2)\right) \in \widehat{\mathbb{Z}}_2^C \times \widehat{\mathbb{Z}}_2^H, \tag{393}$$

implying that we have the mixed 0-form anomaly

$$\exp\left(\pi i \int w_2^C \cup w_2^H\right), \tag{394}$$

between $SO(3)_C$ and $SO(3)_H$ 0-form symmetries, which matches the anomaly for $T[SU(2)]$.

# 8 Generalisations: Solitonic defects and anomalies in various dimensions

The methods outlined in this paper have numerous generalisations to invertible generalized symmetries in higher dimensions. We will not attempt a systematic exploration here but confine ourselves to provide some examples in three, four and five dimensions, which illustrate the vast possibilities for extension of this work.

### 8.1 More solitonic defects

In previous sections, we have explored the utility of defects inducing solitonic background field configurations in describing generalised symmetries and their 't Hooft anomalies. Our focus was on codimension-two solitonic defects inducing non-trivial backgrounds for continuous 0-forms, discrete 1-form and 2-groups, with the property that at least one of the following backgrounds is non-trivial on a small disk $D_2$ intersecting the locus of the defect

$$\int_{D_2} w_2, \qquad \int_{D_2} B_2, \qquad \int_{D_2} B_w, \tag{395}$$

However, there are further possibilities for solitonic backgrounds induced by higher codimension defects.

We may consider codimension-three solitonic defects with the property that they induce non-trivial symmetry backgrounds (on a small ball $D_3$ intersecting the locus of the defect) such that one of the following is non-trivial

$$\int_{D_3} \mathrm{Bock}(w_2), \qquad \text{or} \qquad \int_{D_3} \mathrm{Bock}(B_2), \tag{396}$$

where we have used the Bockstein homomorphism for some short exact sequence. For example, if the short exact sequence involved is

$$0 \to \mathbb{Z}_2 \to \mathbb{Z}_4 \to \mathbb{Z}_2 \to 0, \tag{397}$$

then

$$\mathrm{Bock}(B_2) = \frac{\delta \widetilde{B}_2}{2} \quad \mathrm{mod}\ 2, \tag{398}$$

where $\widetilde{B}_2$ is a $\mathbb{Z}_4$-valued 2-cochain which is a lift of the $\mathbb{Z}_2$-valued 2-cocycle $B_2$.

In codimension-four we may consider solitonic defects with nontrivial

$$\int_{D_4} \mathcal{P}(w_2), \qquad \int_{D_4} \mathcal{P}(B_2), \tag{399}$$

where $\mathcal{P}$ denotes the Pontryagin square operation, where $D_4$ is a small ball intersecting the locus of the defect.

The properties of such defects capture various other types of anomalies not considered in this paper. Some examples, in various spacetime dimensions, are explored in the remainder of this section.

### 8.2 Discrete symmetries in 3d

Suppose we consider a theory with a discrete 0-form symmetry $\Gamma^{(0)}$ in 3d in addition to a continuous symmetry $\mathcal{F} = F/\mathcal{Z}$ of the form described in the earlier sections of the paper.

Consider a solitonic local operator having the property that

$$\int_{D_3} \mathrm{Bock}(w_2), \tag{400}$$

is a non-trivial element of $\mathcal{Z}$. As local operators may be charged under the discrete 0-form symmetry, this determines a homomorphism $\gamma : \mathcal{Z} \to \widehat{\Gamma^{(0)}}$ and captures a mixed 't Hooft anomaly of the form

$$\mathcal{A}_4 = \exp\left(2\pi i \int B_1 \cup \gamma\big(\mathrm{Bock}(w_2)\big)\right), \tag{401}$$

where $B_1$ is a $\Gamma^{(0)}$-valued 1-cocycle background for the discrete symmetry.

### 8.3 Solitonic defects in 4d

Let us now consider 't Hooft anomalies for 0-form and 1-form symmetries in four dimensions. For simplicity, we assume these symmetries do not form part of a non-trivial 2-group. We use the same notation as in earlier sections.

A line operator in 4d may have the property that

$$\int_{D_3} \text{Bock}(B_2),\tag{402}$$

is a non-trivial element of a discrete group $\mathcal{O}$ extending the 1-form symmetry group $\Gamma^{(1)}$. Such line operators may themselves be charged under the 1-form symmetry $\Gamma^{(1)}$, which determines a homomorphism $\gamma : \mathcal{O} \to \widehat{\Gamma}^{(1)}$ and therefore a 't Hooft anomaly of the form

$$\mathcal{A}_5 = \exp\left( 2\pi i \int B_2 \cup \gamma\big(\text{Bock}(B_2)\big)\right).\tag{403}$$

Such anomalies arise when gauging a discrete subgroup of 1-form symmetry forming an extension.

**Example.** The pure $SO(N)$ gauge theory with $N = 4k+2$ is obtained from the $\text{Spin}(N)$ gauge theory by gauging a $\mathbb{Z}_2 \subset \mathbb{Z}_4$ subgroup of the 1-form center symmetry forming a short exact sequence of 1-form symmetries

$$0 \to \mathbb{Z}_2 \to \mathbb{Z}_4 \to \mathbb{Z}_2 \to 0.\tag{404}$$

See e.g. [2, 15] for a discussion of these symmetries and anomalies of these theories. The resulting 1-form symmetry of $SO(N)$ is $\Gamma^{(1)} = \mathbb{Z}_2 \times \mathbb{Z}_2$ with backgrounds $B_2 = (B_2^e, B_2^m)$ and mixed 't Hooft anomaly

$$\frac{2\pi i}{4} \int B_2^m \cup \text{Bock}(B_2^e),\tag{405}$$

involving the Bockstein homomorphism for the short exact sequence.

Let us see this anomaly directly from the perspective of solitonic defects. First, in the absence of a background for the electric 1-form symmetry the theory sums uniformly over $SO(N)$ bundles, which have obstruction to lifting to $\text{Spin}(N)$ bundles given by the second Stiefel-Whitney class $w_2^g$. However, turning on an electric 1-form background, the obstruction is no longer closed and the theory sums over $PSO(N)$ bundles with

$$\delta w_2^g = \text{Bock}(B_2^e).\tag{406}$$

Equivalently, the $SO(N)$ theory now sums over $PSO(N)$ bundles with obstruction $2B_2^e + \widetilde{w}_2^g$ to lifting to $\text{Spin}(N)$ bundles, where $\widetilde{w}_2^g$ is a $\mathbb{Z}_4$-valued co-chain lift of the second Stiefel-Whitney class.

Now consider 't Hooft lines in the $SO(N)$ theory in the equivalence class that carries non-trivial charge under the $\mathbb{Z}_2$ magnetic 1-form symmetry with background $B_2^m$. Such a line operator induces a magnetic gauge flux such that

$$\int_{S^2} w_2^g = 1 \bmod 2,\tag{407}$$

where $S^2$ is a small 2-sphere linking the line. In the presence of a background field for the electric 1-form symmetry, the relation (406) implies that such a line operator induces a background

$$\int_{D_3} \text{Bock}(B_2^e) = 1 \bmod 2,\tag{408}$$

where $D_3$ with $\partial D_3 = S^2$ is a small three-ball intersecting the 't Hooft line. This property of 't Hooft operators charged under the magnetic 1-form symmetry is equivalent to the 't Hooft anomaly. It is also possible to proceed in the other direction by showing that Wilson lines charged under the electric 1-form symmetry induce Bockstein flux for the magnetic 1-form background.

## 8.4 Solitonic defects in 5d

In 5d, we may consider codimension four line operators with the property that

$$\int_{D_4} \mathcal{P}(B_2), \tag{409}$$

is a non-trivial element of the universal quadratic group $Q(\Gamma^{(1)})$ of the 1-form symmetry. We may then consider two types of anomaly (see e.g. [21,25,57] for field theoretic and geometric derivations):

- If such line operators can end on local operators charged under an extension of the 0-form symmetry by $\mathcal{Z}$ then we obtain a homomorphism $\gamma : Q(\Gamma^{(1)}) \to \widehat{\mathcal{Z}}$ and a mixed 't Hooft anomaly

$$2\pi i \int w_2 \cup \gamma(\mathcal{P}(B_2)). \tag{410}$$

- If such line operators are themselves charged under the 1-form symmetry, this determines a homomorphism $\gamma : Q(\Gamma^{(1)}) \to \widehat{\Gamma^{(1)}}$ and cubic 't Hooft anomaly

$$2\pi i \int B_2 \cup \gamma(\mathcal{P}(B_2)). \tag{411}$$

Prominent examples of such anomalies arise in pure gauge theories in 5d, where the 0-form symmetry is an instanton symmetry and the 1-form symmetry is a center symmetry. An example is pure $SU(N)$ with CS-level 0, which has a $U(1)_I$ instanton symmetry with obstruction background $w_2 = c_1 \bmod N$ and 1-form center symmetry $\Gamma^{(1)} = \mathbb{Z}_N$. The 't Hooft anomaly is

$$2\pi i \frac{N-1}{2N} \int (c_1 \bmod N) \cup \mathcal{P}(B_2). \tag{412}$$

To see this anomaly directly, we consider codimension-four instanton-particle lines defined by a non-trivial integral instanton flux on a small ball $D_4$. Such configurations arise dynamically and are not line defects. In the presence of a background field $B_2$ for the 1-form symmetry the instanton number has fractional part

$$\frac{N-1}{2N} \int \mathcal{P}(B_2) \mod 1, \tag{413}$$

which cannot be screened by dynamical objects and therefore defines a non-trivial line defect that induces a background flux for the 1-form symmetry. These line defects may end on local instanton operators with fractional charge under the $U(1)_I$ instanton symmetry and we must pass to an extension by $\mathcal{Z} = \mathbb{Z}_N$ with obstruction class $w_2 = c_1 \bmod N$ and there is a mixed anomaly as above.

Now introduce a non-trivial even Chern-Simons level $k \in 2\mathbb{Z}$, which breaks the 1-form symmetry to $\mathbb{Z}_\ell$ with $\ell = \gcd(k, N)$. The fractional part of the instanton flux is now

$$\frac{p}{q} \int \mathcal{P}(B_2) \mod 1, \tag{414}$$

where

$$p = \frac{N(N-1)}{\gcd(2\ell^2, N(N-1))}, \qquad q = \frac{2\ell^2}{\gcd(2\ell^2, N(N-1))}, \tag{415}$$

are co-prime integers. This means we must pass to an extension of the instanton symmetry by $\mathcal{Z} = \mathbb{Z}_q$ and there is a mixed 't Hooft anomaly

$$2\pi i \, \frac{p}{q} \int (c_1 \bmod q) \cup \mathcal{P}(B_2). \tag{416}$$

In addition there can be a cubic $B^3$ 't Hooft anomaly for the electric 1-form symmetry. For example, for $SU(N)_k$ in 5d the 1-form symmetry $\Gamma^{(1)} = \mathbb{Z}_\ell$ has a cubic anomaly

$$\frac{kN(N-1)(N-2)}{6\ell^3} \int B_2^3. \tag{417}$$

This arises because the instanton-particle line defects may now be charged under the $\mathbb{Z}_\ell$ 1-form symmetry. One way to see this is that, due to the Chern-Simons interaction, the local instanton operators on which instanton-particle lines end now transform under the gauge symmetry and therefore must form junctions with Wilson lines. The latter are charged under the electric 1-form symmetry.

Clearly the implications for solitonic defects and their charges are numerous once known 't Hooft anomalies are viewed in this context.

## Acknowledgments

We thank Clay Cordova, Ingo Runkel and Yuji Tachikawa for illuminating discussions. MB would like to thank colleagues in the Department of Mathematical Sciences at Durham University for discussions and feedback when presenting this work in a series of lectures in the generalized symmetries journal club.

**Funding information** The work of MB is supported by the EPSRC Early Career Fellowship EP/T004746/1 "Supersymmetric Gauge Theory and Enumerative Geometry", STFC Research Grant ST/T000708/1 "Particles, Fields and Spacetime", and the Simons collaboration on Global Categorical Symmetries. This work is supported by the European Union's Horizon 2020 Framework through the ERC grants 682608 (LB and SSN) and 787185 (LB). SSN is supported in part by the "Simons collaboration on Special Holonomy in Geometry, Analysis and Physics".

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
