# Peer review of "Anomalies of Generalized Symmetries from Solitonic Defects"

_SciPost Physics, doi:SciPost Phys. 16, 087 (2024)_

## Round 1 · Referee Report · Anonymous (Referee 1) · 2023-7-28

Report
This manuscript presents a comprehensive study (with main focus on 3d theories but including many discussions valid for arbitrary dimensions) of 't Hooft anomalies of global symmetries from the perspective of properties (notably charges) of certain kinds of defects, called "solitonic defects" by the authors. The work contains solid and interesting results with many worked out examples. Modulo a small list of suggestions given below, it is considered that the work meets the expectations and criteria for the journal, and I recommend it for publication.
Here is the list of suggestions for the improvement and the authors are invited to address them.
-
The meaning or definition of "solitonic defect" is not clear enough and I found that it may lead to "struggles" for readers, especially for those who are not already experts on the subjects. The only properties listed at the beginning of section 1.1 is (i) they source a background for the generalized symmetry and (ii) they need not be "topological". Can authors clarify? For instance, on page 6 when they summarize the content of section 4, it is mentioned that some solitonic defects get converted into non-solitonic defects and what would this mean in terms of (i) and (ii)? It seems important to make the meaning more precise, without needing to wait until very technical discussion given in later sections.
-
Unclear or possibly wrong notations/typos. In equation (1.3), $c_{d-1} (\mathcal{B})$ is defined as some characteristic class made of background fields $\mathcal{B}$. I believe this notation means "degree $(d-1)$" object. However, in equations (1.7)-(1.10), this notation does not seem to be consistent with those equations. For e.g. in (1.7) and (1.9), do you mean $M_{d+1}$? In (1.8), $c_1$ is instead used to mean the first Chern class. Given that this is the introduction of the paper, this level of confusion may discourage further reading and so I suggest cleaning up.
-
It may be useful to discuss the relation between "non-genuine" (local) operators and projective representation. The latter is used for instance in a related recent work by Brennan, Cordova, and Dumitrescu "Line Defect Quantum Numbers $\&$ Anomalies".
-
When you define a vortex defect, can you comment on the relation of your definition (at least part of) to the more broadly used definition in terms of $\pi_1 (G/H)$ when $G$ is broken to $H$? Around (3.2), why the winding should be just one, and not other integer values? What is the physics intuition behind this?
-
For the 0-form flavor symmetry, notations $F$ and $\mathcal{F}$ are chosen so that the relation is given by $\mathcal{F} = F / \mathcal{Z}$. However, the gauged part discussed in section 4, they are flipped as $G = \mathcal{G}/\mathcal{Z}_g$ and I found it not the best choice, or even confusing. Also, it is not clear if it is really necessary to introduce more than one notations for the same objects, e.g. $Z(\mathcal{G})$ and $Z_{\mathcal{G}}$ both to mean the center of $\mathcal{G}$.
-
In (4.15), is the denominator $\mathbb{Z}_{2q}$ correct? Is it really $\mathcal{E}_r$ generated by the central element $(e^{i \pi /q}, e^{i\pi} 1_2)$? If so, can you explain, at least so that I can understand?
-
Can you elaborate more on the discussion below (4.27) as to how $\Gamma^{(1)}$ and $\Gamma_g^{(1)}$ are combined to result in $\Gamma_r^{(1)}$. I found it an important discussion and yet a bit too quick. Some physical insight (in addition to mathematical explanation) might be very helpful.
-
Throughout the draft, I spot several places where words are repeated e.g. "defects defects" on p32 above (5.11). I suggest you correct these typos.
-
One general (possibly very basic and even dumb) question: you are considering solitonic defects which source background for generalized symmetries. And yet, the existence of 't Hooft anomalies are closely related to the fact that they carry "charges" under those symmetries. The question is: is the ability to turn on background really conceptually separate from them being charged? Or is it a special kind of (e.g. fractional) charge which may not be captured by the spectrum of the background that is the key here. Basically, I am looking for more physics explanations of your findings and wonder if authors have a good insight.

---

## Round 1 · Referee Report · Anonymous (Referee 2) · 2023-11-1

Report
The manuscript systematically discusses diagnosis of anomalies from a large class of ``solitonic'' objects. Here are some comments to be addressed before I can recommend publication:
- p4. There is restriction on that kind of class you can use for nontrivial defects. In particular if the charge admits topological boundary condition the object is trivial at long distance.
See e.g. https://arxiv.org/pdf/1910.04962.pdf and various literature about condensation defects.
There are also various relations between characteristic classes and such definition of solitons contains redundancy.
-p5. If a ``solitonic'' object can end, the flux carried by the solitonic object is necessarily trivial and the symmetry is no longer A. Can the author clarify the discussion there?
- the manuscript mostly discussed ``solitonic'' defect of background field. On the other hand, an important source of anomalies are solitonic defects that carry fluxes of dynamical gauge fields. For instance, the monopoles in QCDs with massless fermions can carry nontrivial quantum number and contribute to anomalies due to fermion zero modes (e.g. https://arxiv.org/abs/1810.00844 https://arxiv.org/abs/2108.05369 and the references therein). Can the authors comment on the applications to these situations?

---

## Round 3 · Referee Report · Anonymous (Referee 1) · 2024-2-6

Report

Dear Editor,

The authors addressed most of my comments and the draft looks improved.
As I mentioned in my previous report, the content of this work is solid and interesting and I recommend its publication.

---

## Round 3 · Author Response

We thank very much the referees for the helpful comments. We reply to these comments below:

Report 2: To the points on page 4 and 5: our formalism consistently implements the fact that endable defects are uncharged under certain global symmetries. It is true that there are relations between characteristic classes, but this redundancy does not affect our analysis. To the point on dynamical gauge field: we discuss the fate of solitonic defects and anomalies after gauging global symmetries in section 4

Report 1: 1. The definition is given at the beginning of the paper (introduction): solitonic defects are by definition defects that induce background fields for global symmetries. When a symmetry is gauged, by definition a solitonic defect that was only sourcing that symmetry may stop being a solitonic defect (if it does not happen to source background fields for another global symmetry obtained after gauging) 2. We corrected M_{d-1} to M_{d+1}, and changed the general class c_{d-1} to q_{d-1} not to confuse it with a Chern class. 3. The fact that representations of central extensions of a group are projective representations of the group has been well-known since the birth of quantum mechanics. The paper of Brennan et al. came after ours, and we not feel we need to emphasise this point further. 4. The referee may refer to dynamical vortex defects with some vacuum at infinity that breaks the gauge symmetry. The unit winding is in some sense a choice: given any vortex defect there is a co-character $\phi$ whose respective embedded U(1) has a unit vortex, and this is the one we choose to characterise the vortex by. We rephrased a little the discussion above 3.2 to make this clear 5. The notation \cG and \cF is used to refer to physical gauge and flavor groups, while G and F are reserved for mathematically useful groups related to these physical groups. We have tried to replace \cZ(\cG) with \cZ_\cG whenever confusing 6. This is an arbitrary example, the denominator is that one by definition. The discussion we provide is consistent (the order of the generator is correct). 7. This is a consequence of our definitions. We provide an example on that page. 8. We removed all double words (regular expression) 9. No comment

---

## Round 3 · List of Changes

Minor changes have been made according to the reply to the above reply to the referees. These include notational changes and correction of typos as per comment 2, Referee 2, and notational change as per comment 5, referee 2.

---

## Editorial Decision

published